# Overestimation and Adjustment of Antarctic Ice Flow Velocity Fields Reconstructed from Historical Satellite Imagery

Rongxing Li[1, 2*], Yuan Cheng[1, 2*], Haotian Cui[1, 2], Menglian Xia[1, 2], Xiaohan Yuan[1, 2], Zhen Li[1, 2], Shulei Luo[1, 2], Gang Qiao[1, 2]

[1]Center for Spatial Information Science and Sustainable Development Applications, Tongji University, Shanghai, China
[2]College of Surveying and Geo-Informatics, Tongji University, Shanghai, China

*Correspondence to*: Rongxing Li (ronli_282@hotmail.com) and Yuan Cheng (chengyuan_1994@tongji.edu.cn)

**Abstract.** Antarctic ice velocity maps describe the ice flow dynamics of the ice sheet and are one of the primary components used to estimate the Antarctic mass balance and contribution to global sea level changes. In comparison to velocity maps derived from recent satellite images of monthly to weekly time spans, historical maps, from before the 1990s, generally cover longer time spans, e.g., over 10 years, due to the scarce spatial and temporal coverage of earlier satellite image data. We found velocity overestimations (OEs) in such long-span maps that can be mainly attributed to velocity gradients and time span of the images used. In general, they are less significant in slow-flowing grounded regions with low spatial accelerations. Instead, they take effects in places of high ice dynamics, for example, near grounding lines and often in ice shelf fronts. Velocities in these areas are important for estimating ice sheet mass balance and analyzing ice shelf instability. We propose an innovative Lagrangian velocity-based method for OE correction without the use of field observations or additional image data. The method is validated by using a set of "ground truth" velocity maps for the Totten Glacier and Pine Island Glacier which are produced from high-quality Landsat 8 images from 2013 to 2020. Subsequently, the validated method is applied to a historical velocity map of the David Glacier region from images from 1972–1989 acquired during Landsat 1, 4 and 5 satellite missions. It is demonstrated that velocity overestimations of up to 39 m a$^{-1}$ for David Glacier and 195 m a$^{-1}$ for Pine Island Glacier can be effectively corrected. Furthermore, temporal acceleration information, e.g., on basal melting and calving activities, is preserved in the corrected velocity maps and can be used for long-term ice flow dynamics analysis. Our experiment results in PIG show that OEs of a 15-year span can reach up to 1,300 m a$^{-1}$ along the grounding line and cause an overestimated GL flux of 11.5 Gt a$^{-1}$ if not corrected. The magnitudes of the OEs contained in both velocity and mass balance estimates are significant. When used alongside recent velocity maps of 1990s – 2010s, they may lead to underestimated long-term changes for assessment and forecast modeling of the global climate change impact on the Antarctic ice sheet. Therefore, the OEs in the long span historical maps must be seriously examined and corrected. We recommend that overestimations of more than the velocity mapping uncertainty ($1\sigma$) be corrected. This velocity overestimation correction method can be applied to the production of regional and ice sheet-wide historical velocity maps from long-term satellite images.

# 1 Introduction

Ice flow velocity fields on the Antarctic ice sheet (AIS) have been mapped by using SAR and optical satellite images to study ice sheet-wide ice flow dynamics and AIS responses to global climate changes (Rignot et al., 2011a; Gardner et al., 2018; Shen et al., 2018; Greene et al., 2020a). An important direct use of such velocity fields is to estimate the ice discharge from the AIS to the Southern Ocean and perform mass balance analyses (IMBIE, 2012; Gardner et al., 2018; IMBIE, 2018; Shen et al., 2018; Rignot et al., 2019). Any errors in the reconstructed velocity fields can cause uncertainties in the estimated AIS mass balance and associated contribution to global sea level (GSL) change (Rignot et al., 2011b; Church et al., 2013; DeConto & Pollard, 2016; IPCC, 2019).

The state of the reconstructed velocity fields can be represented by a series of velocity maps that are derived from periodical satellite images. The variations of the available satellite images in both spatial and temporal coverage determine the extent and time span of the velocity maps. While ice sheet-wide annual velocity maps, e.g., in the Inter-mission Time Series of Land Ice Velocity and Elevation (ITS_LIVE) project (Gardner et al., 2018, 2019), have been produced from recent satellite images (Storey et al., 2014; Lillesand, 2015; Li & Roy, 2017; Sabins Jr & James, 2020), regional velocity maps at a seasonal or monthly scale have been generated from optical and SAR images (e.g., Landsat and Sentinel; Frezzotti et al., 1998, 2000; Nakamura et al., 2010; Zhou et al., 2014; Greene et al., 2017, 2018, 2020a; Moon et al., 2021). Furthermore, a weekly ice velocity mapping scheme based on multi-mission satellite images was proposed in Altena & Kääb (2017). However, owing to low image quality, large geolocation errors, and low temporal and spatial coverage, satellite images prior to 1990 are generally less available; appropriate images for ice sheet-wide or large regional velocity mapping with a shorter time span (e.g., seasonal or annual), especially from 1960s and 1970s, are scarce. Therefore, earlier velocity maps have been produced with a limited extent and a longer time span. For example, combinations of first-generation film – based ARGON images of 1963 (Ruffner, 1995; Kim, 2004), early Landsat MSS images of 1970s and TM images of 1980s (Chander et al., 2009) have been used to create regional velocity maps with a time span ranging from 1 to 23 years (Bindschadler & Scambos, 1991; Bindschadler et al., 1996; Wang et al., 2016; Cheng et al., 2019; Rignot et al., 2019), although the unique case of a 1963 velocity map of the Rayner Glacier from two ARGON stereo image pairs has been presented (Li et al., 2017). Such a long time span is problematic when we use the feature matching technique for velocity mapping. For example, at *time1* a feature, with an initial velocity $v_0$ at the first location, is taken in the first image. The same feature is tracked in the second image taken at *time2* after traveling at the velocity $v_0$ and an acceleration $a$ for a time span of $\Delta t$ *(time2-time1)*. Thus, the velocity $v=v_0+a\Delta t$ increases along with the time span $\Delta t$ if acceleration $a$ exists. Given a constant acceleration, the velocity can be overestimated if the time span is long. Or the velocity overestimation is proportional to the time span. With a 10-year span, an overestimation of ~70 m a$^{-1}$ (or ~10%) was estimated for an area near the grounding line of Totten Glacier (Figure A1) (Greene, 2020b). Berthier et al. (2003) compensated the overestimations in a Mertz Glacier mapping study with an image span of 11 years by assigning the overestimated velocities to middle points of travelled segments. This technique should have corrected a large portion of the overestimation given the relatively weak spatial velocity gradient along the main trunk of the glacier. Although it has not been

brought to further attention in publications, given its nature and magnitude, this velocity overestimation issue should be fully understood and a comprehensive correction method should be developed so that corrected historical velocity maps can be analyzed alongside modern maps to create a long record of cohesive ice flow dynamics. Furthermore, this capability of building a long record of AIS ice flow dynamics is important for estimation of long-term AIS mass balance and prediction of the future GSL contribution (Rignot et al., 2019).

This study is a part of our efforts to develop an ice velocity map of East Antarctica for 1963 to 1989 (Li et al., 2017; Ye et al., 2017; Cheng et al., 2019). In this paper we prove the existence of ice velocity overestimation in long-term historical velocity maps and present an innovative correction method. The proposed correction method is based on the Lagrangian velocity that can be calculated from the overestimated velocity map itself, and thus does not require any field observations or additional satellite images. We used a set of "ground truth" velocity maps with time spans of 1 to 7 years derived from recent Landsat 8

images of 2013 to 2020 from the Totten Glacier (TG), East Antarctica, to validate the correction method. We then applied the validated method to a historical velocity map in the David Glacier region, East Antarctica, which was produced from images from 1972 to 1989 acquired during satellite missions of Landsat 1, 4 and 5. We show that the 17-year velocity overestimation can be successfully corrected to within the uncertainty ($1\sigma$) of the 1-year map. Another experiment at the Pine Island Glacier (PIG), one of the most dynamic glaciers in Antarctica, was carried out to demonstrate that the overestimation correction method

can effectively adjust the overestimated velocities as large as 195 m a$^{-1}$, while preserving the velocity change signature caused by temporal accelerations due to basal melting and calving activities for long-term ice flow dynamics analysis.

## 2 Methods

### 2.1 The velocity overestimation issue

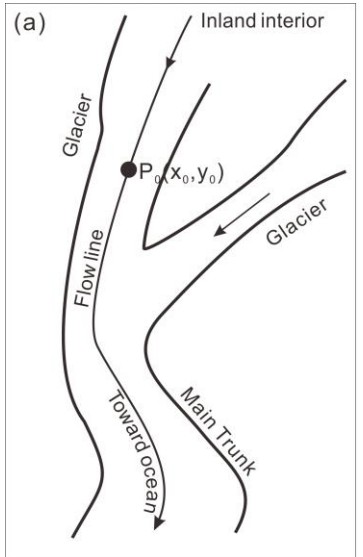 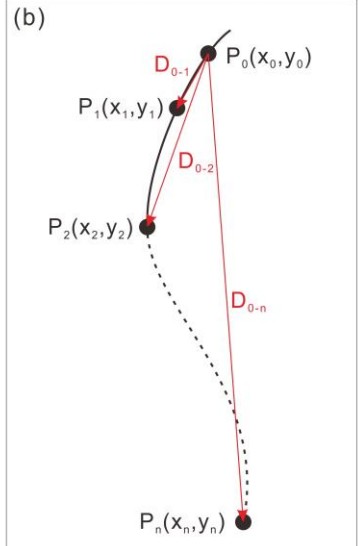 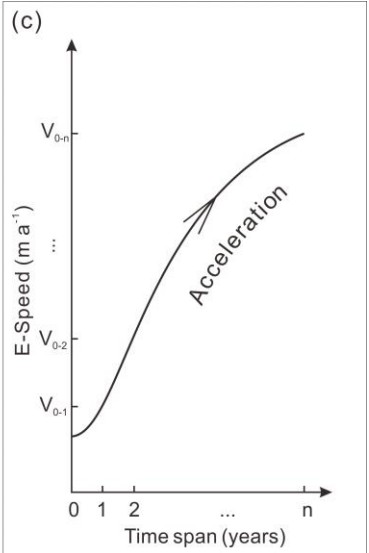

 **Figure 1: Illustration of acceleration - induced overestimation in a velocity map derived from a satellite image pair with a long time span: (a) schematic scene of accelerated ice flow in a glacier–ice shelf system in AIS; (b) calculation of E-velocities at $P_o(x_o, y_o)$ with an equal increment in time span (1 year); and (c) increase of E-velocity at the same point $P_o(x_o, y_o)$ as the time span increases – the cause of the acceleration - induced velocity overestimation.**

We describe an acceleration - induced overestimation using a typical scenario in AIS (Fig. 1a) where ice flow accelerates over a long slope from several glaciers originated from the inland interior, running through the main trunk, and discharging to the ocean (Bamber et al., 2000; Cuffey & Paterson, 2010; Rignot et al., 2011a). In order to quantify the velocity overestimation, for any point $P_o(x_o, y_o)$ on a glacier (Fig. 1a), we define its velocity in two different frameworks, Eulerian and Lagrangian (Chu & Fan, 2014; Chenillat et al., 2015; Altena & Kääb, 2017). First, the velocity in the Eulerian framework (E-velocity) is defined as $V^E = D / dt$, the *straight-line* distance ($D$) divided by the time span ($dt$). Note that for simplicity in equation derivation and discussion, we interchange a vector with its scalar; thus, velocity and speed are not strictly distinguished in this paper. Hence a velocity field described by a velocity map is also defined in the Eulerian framework. In reality, the start point of $D$ is measured on the first image and end point on the second image; the two images are taken with a time span of $dt$ apart (Li, 1998; McGlone, 2013). The reconstructed velocity map is stored as velocity components ($V^E_x$, $V^E_y$) in the $x$ and $y$ directions, from which the velocity field can be reconstructed. The velocity, represented by a velocity map, may indicate an "instantaneous" or average velocity depending on the time span $dt$ (Cuffey & Paterson, 2010; Rignot et al., 2011a). Second, the velocity in the Lagrangian framework (L-velocity) is defined as $V^L = S / dt$, the *curved distance traversed along the flow line* ($S$) divided by the time span ($dt$; Altena & Kääb, 2017). Given a time span and an initial point $P_o(x_o, y_o)$ (Fig. 1a), its L-velocity can be determined by tracking the point using a velocity map (Chu & Fan, 2014; Chenillat et al., 2015). Operations in the Lagrangian framework are often performed to estimate advected ice features (Altena & Kääb, 2017) or forecast future events in earth science applications, e.g., mud and debris flow of landslides (Debella-Gilo & Kääb, 2013; Feng et al., 2016), ocean currents at different depths (Glenn et al., 2016; van Sebille et al., 2018), and storm center motion of a typhoon or hurricane (Cram et al., 2007; Euler et al., 2019).

Assume that we use a set of $n+1$ images ($Image_i$, $i=0, 1, ... n$) that are taken with a time interval of 1 year to produce $n$ velocity maps, $V_{0-i}$, each derived from an image pair ($Image_0$, $Image_i$). The time span of the maps increases from 1 to $n$ years. These maps are defined in the Eulerian framework (Fig. 1b). For $P_o(x_o, y_o)$ its location after $i$ years, $P_i(x_i, y_i)$, is determined in $Image_i$; its Eulerian distance $D_{0-i}$ is measured using both $Image_0$ and $Image_i$ ($i=1, 2, ...n$). Consequently, the E-velocity of an $i$-year span ($\Delta t_{o-i}$) at $P_o(x_o, y_o)$ is

$$V^E_{0-i} = \frac{D_{0-i}}{\Delta t_{0-i}} . \tag{1}$$

As the time span increases at a fixed rate of 1 year, the traversed straight-line distance $D_{0-i}$ (red lines in Figure 1b), correspondingly E-velocity $V^E_{0-i}$, increases rapidly because of the acceleration over the traverse (Fig. 1c). In principle, every $V^E_{0-i}$ ($i=1, 2, ...n$) value represents the velocity at the same point $P_o(x_o, y_o)$ (Figure 1a) in these $n$ velocity maps. In the cases where $Image_i$ were not available and thus the maps $V_{0-i}$ ($i=1, ... n-1$) were not produced, we only had the map $V_{0-n}$ with the

longest span of $n$ years. It is obvious that at $P_o(x_o, y_o)$ its $n$-year velocity $V_{0-n}^E$ is significantly larger than the 1-year velocity $V_{0-1}^E$ (Figure 1c). In general, we define the velocity overestimation of an $i$-year E-velocity as

$\qquad OE_{0-i} = V_{0-i}^E - V_{0-1}^E.$ (2)

Here we use a velocity map of a 1-year span as a baseline ("overestimation free") throughout the paper for simplicity, which can be changed for glacier regions of different ice flow dynamics (spatial acceleration, mainly caused by bed topography and slopes). For example the baseline span is one year for TG in *Experiment 1* and three months for PIG in *Experiment 3*. We require that the overestimation of the baseline map is negligible, or smaller than σ (velocity mapping uncertainty).

## 2.2 Overestimation correction based on Lagrangian velocity

We propose a method for correction of overestimation in a long-term velocity map based on its Lagrangian velocities. The velocity field described by the baseline map is used to calculate trajectories of ice mass and L-velocities. The objective is to calculate the overestimation $OE_{0-i}$ in $V_{0-i}^E$ ($i=2, \dots n$) so that these velocities of longer spans are corrected using Equation (2) and describe the same velocity field as the 1-year $V_{0-1}^E$.

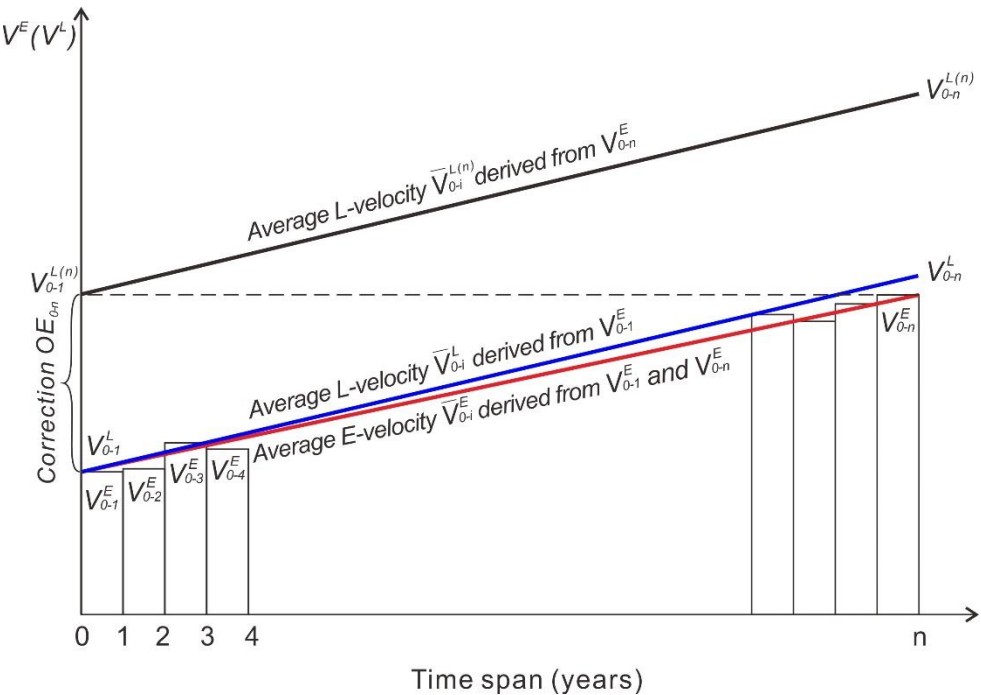

**Figure 2: Derivation of equation for overestimation correction using L-velocity. Eulerian velocities $V_{0-i}^E$ ($i=1, 2, \dots n$) are represented as bars. The red line is the average Eulerian velocity $\overline{V}_{0-i}^E$ of $V_{0-1}^E$ and $V_{0-n}^E$ . The blue line is the average Lagrangian velocity $\overline{V}_{0-i}^L$ of $V_{0-1}^L$ and $V_{0-n}^L$ derived from $V_{0-1}^E$. The black line is the average Lagrangian velocity $\overline{V}_{0-i}^{L(n)}$ of $V_{0-1}^{L(n)}$ and $V_{0-n}^{L(n)}$ derived from $V_{0-n}^E$.**

For point $P_o(x_o, y_o)$ in Fig. 1a, the E-velocity $V_{0-i}^E$ ($i=1, 2, \dots n$) is presented by individual bars in Fig. 2. Correspondingly its

L-velocity of $i$-year span can be calculated as (Halliday et al., 2013)

$$V_{0-i}^L = \frac{S_{0-i}^L}{\Delta t_{0-i}}, \tag{3}$$

where $S_{0-i}$ is the Lagrangian trajectory (L-trajectory) distance ($i=1, 2, \dots n$):

$$S_{0-i}^L = \int_{t_0}^{t_i} V_{0-1}^E(t)\, dt . \tag{4}$$

Within a baseline time span (e.g., 1 year or shorter), the difference between the E-distance and L-distance may be considered within the mapping uncertainty (1 $\sigma$): $V_{0-1}^E \approx V_{0-1}^L$. Beyond that span, the curved trajectory distance is longer than the straight distance ($S_{0-i} \geq D_{0-i}$, see Fig. 1b); thus, we have $V_{0-i}^L \geq V_{0-i}^E$ ($i=2, 3, \dots n$). An approximation of the increased difference between $V_{0-i}^L$ and $V_{0-i}^E$ is presented by the trends of L-velocity (blue line) and E-velocity (red line), which start from the same velocity at $V_{0-1}^L$ (or $V_{0-1}^E$) and reach the maximum difference between $V_{0-n}^L$ and $V_{0-n}^E$ at the end (Fig. 2).

**Premise I:** Within a baseline time span (e.g., 1 year or shorter) each segment (from $P_{i-1}$ to $P_i$ in Figure 1b) is relatively short and the E- and L-velocity difference is smaller than $\sigma$. Furthermore, over the map span of n years (e.g., 5-10 years or longer) the accumulated E- and L-distances along the entire trajectory do not deviate significantly from each other, so that the maximum velocity difference in the Lagrangian and Eulerian frameworks (end points of blue and red lines in Fig. 2) is limited within a threshold ($V_{0-n}^L - V_{0-n}^E \leq k\,\sigma$), where $k$ is a constant and $\sigma$ is the velocity mapping uncertainty.

In reality, the available historical images may only allow us to produce an E-velocity map with the longest time span, i.e. $V_{0-n}$, which leads to the maximum overestimation as defined in Equation (2). Based on this map $V_{0-n}$ of n-year span, the i-year span L-velocity (black line in Figure 2) is defined as follow:

$$V_{0-i}^{L(n)} = \frac{S_{0-i}^{L(n)}}{\Delta t_{0-i}}, \text{ and} \tag{5}$$

$$S_{0-i}^{L(n)} = \int_{t_0}^{t_i} V_{0-n}^E(t_i)\, dt . \tag{6}$$

Consequently, $V_{0-1}^{L(n)}$ (start point of black line in Figure 2) is set to be equal to $V_{0-n}^E$.

**Premise II:** Within the time span of $n$ years (e.g., over 5-10 years), the velocity field described by $V_{0-1}^E$ and $V_{0-n}^E$ does not change significantly, so that the line between $V_{0-n}^{L(n)}$ and $V_{0-1}^{L(n)}$ (black line in Fig. 2) and that between $V_{0-n}^L$ and $V_{0-1}^L$ (blue line in Fig. 2) are approximately parallel to each other. Accordingly, the difference between their simple averaged accelerations is within a threshold ($\left|\bar{a}(V^{L(n)}) - \bar{a}(V^L)\right| \leq k' \frac{\sigma}{\Delta t_n}$), where $k'$ is a constant and $\sigma$ is the velocity mapping uncertainty and

$$\bar{a}(V^{L(n)}) = \frac{V_{0-n}^{L(n)} - V_{0-1}^{L(n)}}{\Delta t_n}, \quad \bar{a}(V^L) = \frac{V_{0-n}^L - V_{0-1}^L}{\Delta t_n} . \tag{7}$$

Given the long span velocity $V_{0-n}^E$ we can calculate the $n$-year L-velocity $V_{0-n}^{L(n)}$. *Premise II* measures the degree of spatial acceleration invariance over the n-year span (parallelity between black and blue lines in Fig. 2). Thus, it is a *necessary condition*

for using the difference of $V_{0-n}^{L(n)} - V_{0-1}^{L(n)}$, or $\sim (V_{0-n}^{L(n)} - V_{0-n}^{E})$, from the n-year span map to substitute for the difference of $V_{0-n}^{L} - V_{0-1}^{E}$ from the 1-year span map; furthermore, based on *Premise I* the overestimation can be computed as

$$OE_{0-n} = V_{0-n}^{E} - V_{0-1}^{E} = V_{0-n}^{L} - V_{0-1}^{E} = V_{0-n}^{L(n)} - V_{0-n}^{E}. \tag{8}$$

Consequently, using the map with the longest time span of n years, we can effectively go back to $V_{0-1}^{E}$ using a correction term defined as *Correction*$=V_{0-n}^{E} - V_{0-n}^{L(n)}$.

In principle, *Premise I* presents a sufficient condition, $V_{0-n}^{L} - V_{0-n}^{E} \leq k\,\sigma$. In case *Premise I* does not hold, for example, because of temporal accelerations induced by calving and basal melting activities (*Experiments 1* and *2*), we are still able to correct the OE portion induced by spatial acceleration and preserve the uncorrected OE portion induced by temporal acceleration in the
170 residual $\varepsilon$ for ice flow dynamics change analysis (Fig. A2).

***Overestimation Correction Theorem***: Assume that the necessary condition in *Premise II* is met, spatial acceleration - induced overestimations in long time span velocities $V_{0-n}^{E}$ can be corrected or reduced using the following *Correction* term, regardless of temporal acceleration:

$$V_{0-1}^{E} = V_{0-n}^{E} + Correction + \varepsilon, \tag{9}$$

$$Correction = V_{0-n}^{E} - V_{0-n}^{L(n)}. \tag{10}$$

If *Premise I* holds, *Correction* $\approx -OE_{0-n}$; otherwise, $|Correction| < |OE_{0-n}|$, preserving the velocity increases induced by temporal accelerations in the residual term $\varepsilon$ (Fig. A2).

Additionally, given $n$ discrete E-velocities $V_{0-i}^{E}$ ($i=1, 2, \ldots n$; Fig. 2), the acceleration can be estimated using a linear regression model. Its least squares (LS) estimation is (Montgomery et al., 2021)

$$a(V^{E}) = \frac{n \sum i V_{0-i}^{E} - \sum i \sum V_{0-i}^{E}}{n \sum i^2 - (\sum i)^2}. \tag{11}$$

On the other hand, this acceleration can also be approximated by an averaged acceleration using the initial and end velocities:

$$\overline{a}(V^{E}) = \frac{V_{0-n}^{E} - V_{0-1}^{E}}{\Delta t_n}. \tag{12}$$

The two acceleration estimates can be compared to determine if the averaged acceleration be used in the case that the intermediate velocities $V_{0-i}^{E}$ ($i=2, \ldots n-1$) are not available. The red line is then used instead of $V_{0-i}^{E}$ ($i=1, 2, \ldots n$; Fig. 2).

**2.3 Implementation aspects**

**Trajectory and L-velocity computation:** The computation of the L-trajectory distances $S_{0-i}^{L}$ and $S_{0-i}^{L(n)}$ involves the numerical implementation of the integral of the velocity field from $P_o(x_o, y_o)$ to $P_i(x_i, y_i)$ along a flow line on the maps of $V_{0-1}$ and $V_{0-n}$ using Equations (4) and (6), respectively. We select an area of the velocity map of 2013 of Pine Island Glacier, West Antarctica

(Figs. 3a and b; Gardner et al., 2018), to show the details of integral implementation. The velocity map has a time span of 1 year and a resolution of 240 m. At each grid cell the velocity is stored according to its components ($V_x^E$, $V_y^E$). We preprocess each component separately by using a moving window smoothing filter to reduce noise. A 10-year trajectory goes through an ice flow profile from $P_0$ at 2560 m a$^{-1}$ to $P_{10}$ at 3935 m a$^{-1}$ with an average acceleration of 138 m a$^{-2}$. As shown in the enlarged area for the annual trajectory from $P_2$ to $P_3$ (Fig. 3c), the integral is carried out by accumulation of sub-trajectories (between white dots) with a monthly increment. The sub-grid positions of the monthly segments are interpolated for integration of the overall L-distance. Finally, we have the result of $V_{0-1}^E = 2560$ m a$^{-1}$, $S_{0-10}^L = 34960$ m and $V_{0-10}^L = 3496$ m a$^{-1}$, compared to $V_{0-10}^E = 3491$ m a$^{-1}$. The 10-year overestimation ($V_{0-10}^E - V_{0-1}^E$) at $P_0$ is approximately 931 m a$^{-1}$ (36%). Examples of overestimation correction in PIG are given in *Experiment 2* and Discussion.

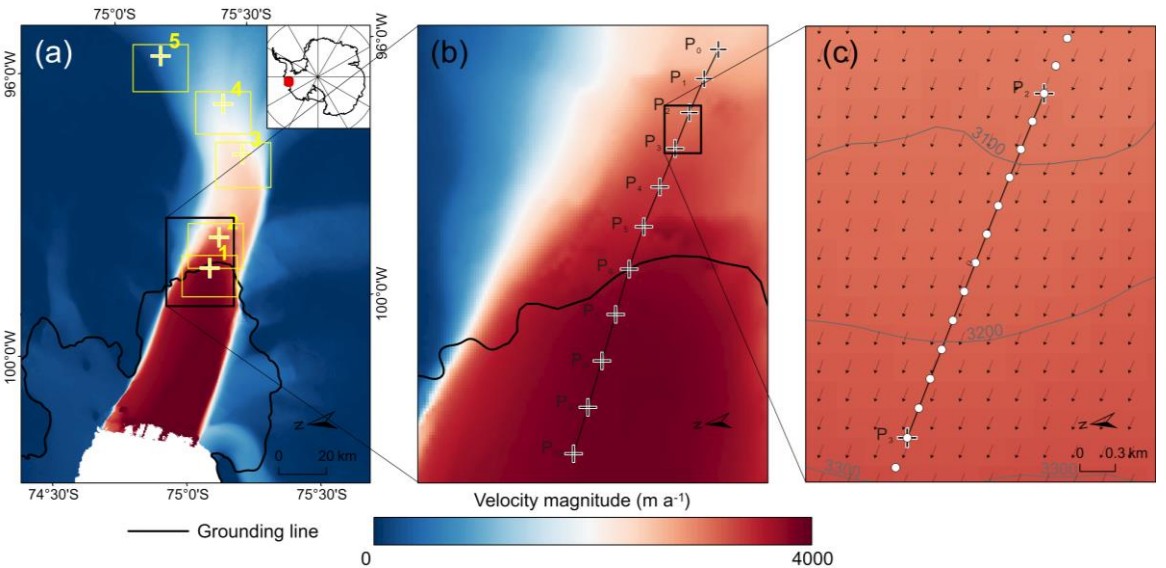

**Figure 3: Implementation of numerical integral and L-velocity calculation in Pine Island Glacier, West Antarctica: (a)** black rectangle indicates the area of 10-year trajectory in (b); numbered yellow rectangles with crosses are five selected areas in *Experiment 3* (Fig. A3); background is 1-year velocity map (Gardner et al., 2018); **(b)** enlarged trajectory area with annual points from $P_0$ to $P_{10}$; and **(c)** numerical integral of the third-year sub-trajectory from $P_2$ to $P_3$ with a monthly increment.

**Denoising and map quality:** the quality of the reconstructed velocity field is important for L-trajectory calculation and subsequent overestimation correction. Compared to the above Landsat 8 velocity map, velocity maps derived from historical images may have higher uncertainties due to the low image quality that is inherent in the satellite images of early missions (Kim, 2004; Ye et al., 2017). In some cases, there may be even gaps in the velocity map because image features may not be tracked by using the optical flow or feature matching techniques when ice features disappear through glaciological processes or large advection motion in ice shelf fronts (Scambos et al., 1992; Schenk, 1999; McGlone, 2013). Therefore, preprocessing of the velocity maps should be performed to eliminate outliers; then interpolation may be applied to fill small gaps so that the integral of L-trajectories and the calculation of L-velocities can be realized. Despite the sub-pixel accuracy of the

orthorectification of historical images and the sub-pixel precision of the cross-correlation-based feature matching method, the reconstructed velocity maps may still be subject to a systematic bias called pixel-locking, which results in matched features moving close to integer pixel positions (Shimizu & Okutomi, 2005). This pixel level bias may introduce additional uncertainties to areas of large velocity gradients. The pixel-locking effect can be reduced by using a coarse-to-fine hierarchical feature matching method as demonstrated in the Antarctic and planetary environments (Debella-Gilo & Kääb, 2011; Li et al., 2011; Heid & Kääb, 2012; Li et al., 2017).

## 3 Results

### 3.1 Experiment 1: Validation of the correction method at Totten Glacier, East Antarctica

*Preparation of validation velocity maps*

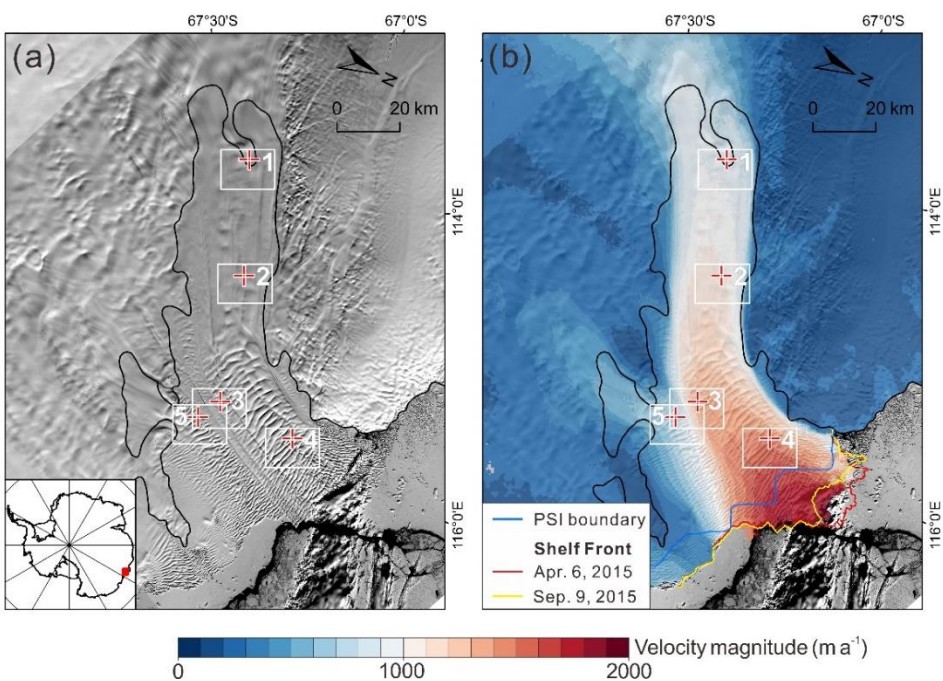

**Figure 4: Totten Glacier as an example for the validation of the velocity overestimation correction method: (a) the Totten Glacier region and five areas (white rectangles) selected in different parts of the ice shelf; background is Landsat 8 image of December 3, 2013; black line is grounding line from Rignot et al. (2011c); and (b) velocity map of Totten Glacier of 2013 (Gardner et al., 2018); red and yellow lines are the shelf front of April 6 and September 9, 2015, respectively; light blue line is the PSI boundary (Fürst et al., 2016).**

This experiment is designed to validate the proposed velocity overestimation correction method using the velocity data of Totten Glacier (TG) (Fig. 4a), which is one of the most active glaciers in East Antarctica and has been experiencing significant mass loss since 1989 (Li et al., 2017; Gardner et al., 2018; Shen et al., 2018). Figure 4b shows an ice flow velocity map of TG

from 2013 from ITS_LIVE (Gardner et al., 2019). This 240 m resolution velocity map was derived from the 15 m resolution

Landsat 8 images with a 1-year span (January to December, 2013). Under the primary forces of gravity, margin shear, basal drag, and others (Bamber et al., 2000; Cuffey & Paterson, 2010; Rignot et al., 2011a), the ice flow accelerates over a path of 130 km along the main trunk at a velocity of ~600 m a$^{-1}$ upstream of the grounding line to ~2,300 m a$^{-1}$ at the ice shelf terminus.

In order to validate the velocity overestimation correction method, we need to determine if *Premises I and II* are fulfilled. This further requires us to have E-velocity from the map series of $V_{0-i}$ (*i=1, 2, ... n*), which are not readily available except the 1-

235 year map of 2013 in Fig. 4b. To avoid lower quality historical velocity maps that may influence the effectiveness of the validation, we use the earliest available high-quality Landsat 8 images from 2013 to 2020 to produce velocity maps $V_{2013-i}$ (*i=2014, ... 2020*). The validation should also consider velocities of areas with different ice flow dynamics. Thus, we select five areas (white rectangles in Fig. 4) to produce smaller maps (~10 km × ~20 km), which contain 7-year trajectories and represent different ice flow dynamic characteristics of the ice shelf, i.e., *Area 1* close to the grounding line, *Areas 2 and 4*

along the main trunk, *Area 5* on the tributary flow, and *Area 3* near a suture zone between the main and tributary glaciers. Using the matching method adapted for ice surface features (Li et al., 2011, 2017), we produced 35 smaller-sized velocity maps $V_{2013-i}$ (*i=2014, ... 2020*) for seven time spans and five areas. Information about the Landsat 8 images used and the acquisition dates are given in Table A1. The "OE-free" velocity maps of 1-year $V_{2013-2014}$ of all 5 areas are illustrated in Panels 1a–5a of Fig. 5. The corresponding overestimated maps of 7-year span $V_{2013-2020}$ are shown in Panels 1b–5b.

The uncertainty of the maps is estimated from two error sources: the geolocation error of the satellite images $\sigma_{Ref}$ and the error of feature matching $\sigma_{Match}$:

$$\sigma_{Velocity} = \frac{1}{\Delta t} \sqrt{\sigma_{Ref}^2 + \sigma_{Match}^2}, \tag{13}$$

where $\sigma_{Ref}$ for Landsat 8 images is 18 m (Storey et al., 2014) and $\sigma_{Match}$ is set to 8 m, ~0.5 pixel (Li et al., 1998 and 2017). Thus, the uncertainty of the measured E-distance is 20 m. Consequently, the velocity uncertainty $\sigma_{Velocity}$ becomes 20 m a$^{-1}$

and 3 m a$^{-1}$ for a 1-year map and a 7-year map, respectively.

In each area, we select an ice feature in the upper stream part in 2013. Its locations in the following 7 years are determined from the satellite images of 2014–2020 using the feature matching method and shown as red triangles in the maps (Fig. 5). Correspondingly, we track its annual trajectory locations from the 1-year and 7-year maps using Equations (4) and (6) and plotted them as blue dots and black crosses, respectively, on the maps. Using these trajectories in the Eulerian and Lagrangian

frameworks we calculate $V_{2013-i}^E$, $V_{2013-i}^L$, and $V_{2013-i}^{L(7)}$ and plot them as red, blue, and black curves in Panels 1c–5c, respectively.

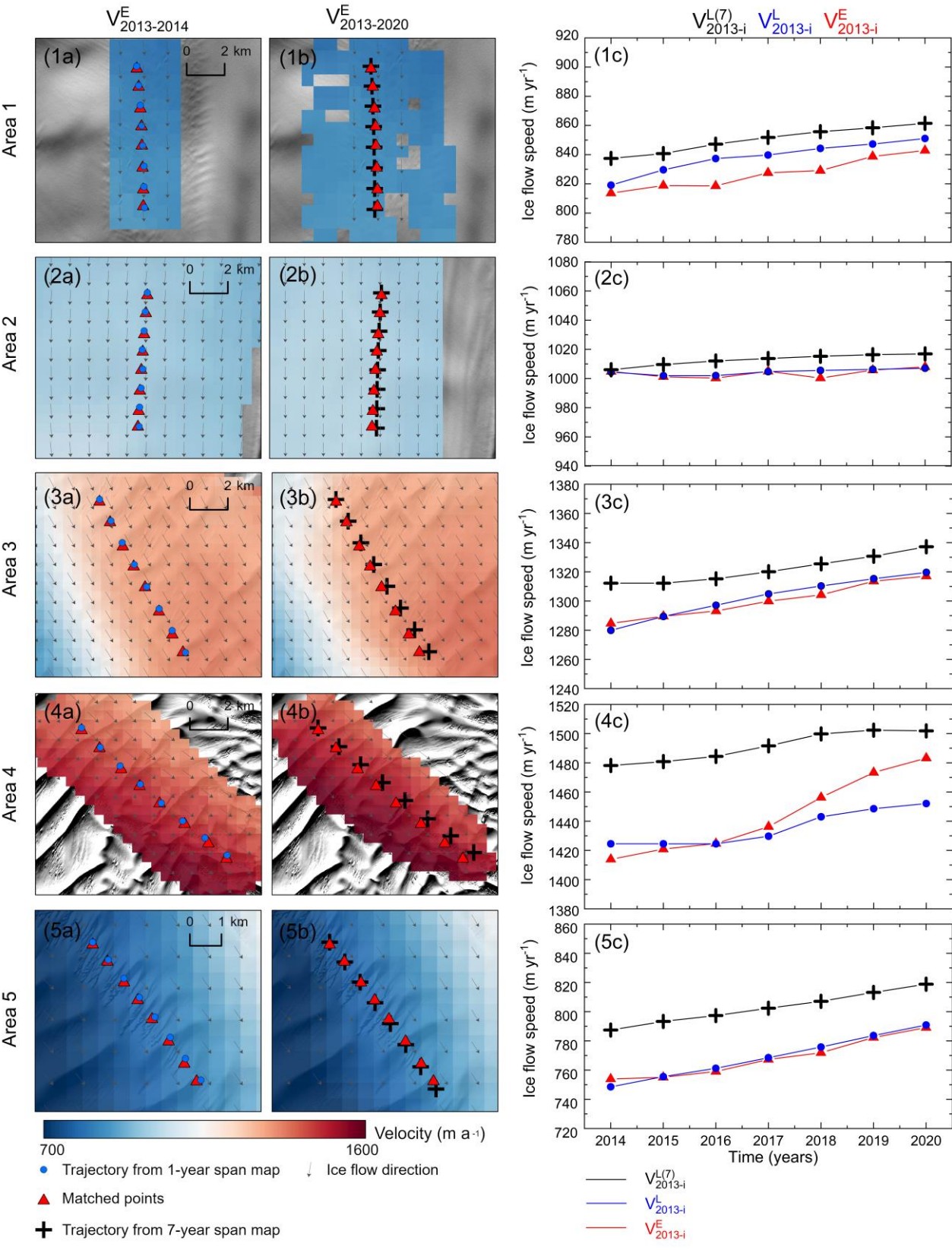

**Figure 5: Velocities in five areas (rectangles in Fig. 4) of the Totten Glacier are used to validate *Premises I* and *II*. (a)** Panels 1a–5a show the reconstructed 1-year velocity maps $V_{2013\text{-}2014}$ in the areas; matched points (red triangles) are used to map E-velocity $V^E_{2013-i}$ (*i=2014, …, 2020*) (red lines in Panels 1c-5c); points along the flow line (blue dots) are tracked from the 1-year maps and used to calculate L-velocity $V^L_{2013-i}$ (blue lines in Panels 1c-5c). **(b)** Similarly, Panels 1b–5b illustrate the reconstructed 7-year velocity maps $V_{2013\text{-}2020}$ in the areas with the matched points (red triangles) for E-velocity $V^{E(7)}_{2013-i}$; points along the flow line (black crosses) are tracked from the 7-year span velocity map and used to calculate L-velocity $V^{L(7)}_{2013-i}$ (black lines in Panels 1c-5c). **(c)** In each area (Panels 1c–5c) the difference between red and blue lines increases with time, but is limited within k σ at the end (*Premise I*), except a large k in *Area 4* because of effect of a calving event; the black line is above blue line due to the spatial acceleration - induced OE over 7-year span; but they are relatively parallel (*Premise II*) and thus, a correction can be estimated.

### *Validation of Premises I and II*

In general, the L-velocities from the 1-year map (blue curves in Fig. 5) are mostly faster than the corresponding E-velocities (red curves). To validate *Premise I* we further examine the L-velocity $V^L_{2013-2020}$ and E-velocity $V^E_{2013-2020}$ of the 7-year span, i.e., differences between the end points of blue and red curves in all five areas (Panels 1c–5c of Fig. 5). They are less than or equal to 8 m a$^{-1}$ (($V^L - V^E)_{2013-2020}$ in Table 1), except -31 m a$^{-1}$ in *Area 4*, which was influenced by a calving event in 2015 as mentioned in the Discussion section. The averaged difference of all five areas is -4 ± 16 m a$^{-1}$, that is, within 20 m a$^{-1}$, the uncertainty (*1 σ*) of the 1-year velocity map (2013–2014) that is considered "OE free". Consequently, the conditions in *Premise I* are met.

For each area we calculated a least squares acceleration estimate $\boldsymbol{a}$ ($V^{L(7)}_{2013-i}$) of the 7-year L-velocity  (black curves in Panels 1c–5c of Fig. 5) over all years (*i=2013, 2014, … 2020*) using Equations (11); then we calculated an average acceleration $\overline{\boldsymbol{a}}\left(V^{L(7)}_{2013-2020}\right)$ using Equation (12) based only on the velocities of the beginning and end years (2013 and 2020). In all five areas their differences (Table A2) are less than 1.3 m a$^{-2}$ which is less than $\sigma_a$ (3 m a$^{-2}$), the acceleration uncertainty estimated from the velocity uncertainty of 20 m a$^{-1}$ (*1 σ*)). That means that within the uncertainty of the 1-year velocity map we can substitute the more rigid LS acceleration $\boldsymbol{a}$ represented by a black curve in Fig. 5 with the average acceleration $\overline{\boldsymbol{a}}$ estimated from the straight line between the beginning and end velocities. Similarly, the differences between accelerations $\boldsymbol{a}$ and $\overline{\boldsymbol{a}}$ of the 1-year L-velocity $V^L_{2013-i}$ in all five areas (Table A2) are also less than 1.5 m a$^{-2}$. Within the uncertainty of the 1-year velocity map we can then substitute $\boldsymbol{a}$ (blue curve in Fig. 5) with $\overline{\boldsymbol{a}}$ (straight line) for the 1-year L-velocity $V^L_{2013-i}$.

Validation of *Premise II* requires us to examine the accelerations represented by the 7-year L-velocity $V^{L(7)}_{2013-i}$ (black curves in Fig. 5) and the 1-year L-velocity $V^L_{2013-i}$ (blue curves). The relatively well-maintained parallelity of the black and blue curves in all five areas shows that the velocity trend in the Lagrangian framework has not changed significantly over the 7 years. This is further quantified by the differences between the averaged 7-year and 1-year accelerations $\overline{\boldsymbol{a}}\left(V^{L(7)}\right) - \overline{\boldsymbol{a}}\left(V^L\right)$ in Table 1. The differences are within ± 2.1 m a$^{-2}$ in all five areas, with an overall difference of -0.8 ± 1.3 m a$^{-2}$, which are all within $\sigma_a$ (3 m a$^{-2}$). Thus, the conditions in *Premise II* are also fulfilled.

**Table 1. Velocity and acceleration in Eulerian and Lagrangian frameworks used for validation of the overestimation correction method.** *"Actual E-velocity and OE"* lists actually mapped 1-year and 7-year E-velocities and their differences as overestimations in all five areas. *"Premise I"* contains 7-year L-velocities computed from the 1-year velocity map and corresponding L- and E-velocity differences, which are used for validating *Premise I*. *"Premise II"* illustrates averaged L-accelerations computed from the 7-year and 1-year velocity maps, respectively, as well as their differences, which are used for validating *Premise II*. *"Overestimation correction"* presents 7-year L-velocities computed from the 7-year map, overestimation corrections, and E-velocities and residuals (or errors) after correction.

| Area | *Actual E-velocity and OE* | | | *Premise I* | | *Premise II* | | | *Overestimation correction* | | | |
|---|---|---|---|---|---|---|---|---|---|---|---|---|
| | $V^E_{2013-14}$ (m a$^{-1}$) | $V^E_{2013-20}$ (m a$^{-1}$) | $OE_{Actual}$ (m a$^{-1}$) | $V^L_{2013-20}$ (m a$^{-1}$) | $(V^L - V^E)_{2013-20}$ (m a$^{-1}$) | $\bar{a}(V^{L(7)})$ (m a$^{-2}$) | $\bar{a}(V^L)$ (m a$^{-2}$) | $\bar{a}(V^{L(7)}) - \bar{a}(V^L)$ (m a$^{-2}$) | $V^{L(7)}_{2013-20}$ (m a$^{-1}$) | Corr. (m a$^{-1}$) | $V^E_{Corr.}$ (m a$^{-1}$) | $\varepsilon$ (m a$^{-1}$) |
| 1 | 814 | 843 | 29 | 851 | 8 | 3.4 | 4.6 | -1.2 | 861 | -18 | 825 | 11 |
| 2 | 1005 | 1008 | 3 | 1007 | -1 | 1.6 | 0.4 | 1.2 | 1017 | -9 | 999 | -6 |
| 3 | 1285 | 1317 | 32 | 1320 | 3 | 3.6 | 5.7 | -2.1 | 1337 | -20 | 1297 | 12 |
| 4 | 1414 | 1483 | 69 | 1452 | -31 | 3.4 | 3.9 | -0.5 | 1502 | -19 | 1464 | 50 |
| 5 | 754 | 789 | 35 | 791 | 2 | 4.5 | 6.0 | -1.5 | 819 | -30 | 759 | 5 |
| MEAN | 1054 | 1088 | 34 | 1084 | -4 | 3.3 | 4.1 | -0.8 | 1107 | -19 | 1069 | 14 |
| STD | 289 | 302 | 24 | 290 | 16 | 1.1 | 2.3 | 1.3 | 300 | 7 | 304 | 21 |

### *Overestimation correction*

As shown in the first part of Table 1, we compare the overestimated E-velocity $V^E_{2013-2020}$ in the 7-year map $V_{2013-2020}$ with $V^E_{2013-2014}$ in the 1-year map $V_{2013-2014}$. Their differences make the actual overestimation $OE_{Actual}$ in five areas. The mean overestimation of 34 m a$^{-1}$ with a range from 3 to 69 m a$^{-1}$ is significant in comparison to σ (20 m a$^{-1}$). Thus, the overestimation should be corrected. It should be noted that a higher overestimation is expected for historical velocity maps of longer time spans, e.g., 10–15 years.

Using our correction method in Equations (9-10), we estimated the *Correction* term ($V^E_{2013-2020} - V^{L(7)}_{2013-2020}$) and applied it to the overestimated $V^E_{2013-2020}$ (Table 1). After the overestimation correction, the corrected velocity $V^E_{Corr.}$ from the 7-year span $V^E_{2013-2020}$ is agreeable to the 1-year span $V^E_{2013-2014}$ within an average difference of 14 m a$^{-1}$ ($\varepsilon$, residual), which is less than σ (20 m a$^{-1}$). Overall, the overestimations are effectively corrected.

### 3.2 Experiment 2: Velocity overestimation correction in the David Glacier region, East Antarctica

*Experiment 2* in the David Glacier region on the Scott Coast, East Antarctica, demonstrates the applicability of the introduced method to correct the overestimation in historical multi-year span velocity maps derived from long-term images from 1972 to 1989. Velocities in this region from 1988 to 1992 were mapped by using GPS and image feature tracking techniques (Frezzotti et al., 1998, 2000). A new GPS campaign was carried out in the region during 2005-2006 (Danesi et al., 2008). Velocity changes from 2016 to 2020 were detected using Sentinel-1A SAR images (Moon et al., 2021). In this experiment we produced a velocity map of the region from 64 Landsat images collected from 1972 to 1989 (Table A3). The image pairs used for velocity mapping have mainly four time spans, namely 1, 15, 16 and 17 years, with their footprints illustrated in Fig. 6a. The Landsat

images were first orthorectified using ground control points (GCPs), which were selected at outcrops, blue ice, and ice rises (Ye et al., 2017). Then the velocity field was reconstructed in three steps: a) measurement of a set of manually selected seed

points for representing the initial structural velocity information in stationary, low velocity, and dynamic flow areas; b) velocity point generation by the ice surface feature-based matching method controlled by a triangular irregular network (TIN) model and initiated by the seed points; and c) dense grid matching for generation of the gridded ice velocity field with a mixed time span from 1972 to 1989. This feature and grid image matching method has been developed for surface mapping from optical satellite images in challenging planetary and polar environments, and successfully applied to the reconstruction of the ice

velocity fields in the Rayner and Fimbul ice shelves in East Antarctica from historical optical satellite images of 1963–1987 (Li et al., 2011, 2017). The produced velocity map has a grid spacing of 500 m after a Kriging interpolation from all matched feature and grid points (Fig 6b). The uncertainty ($1\sigma$) of the velocity points is 4 m a$^{-1}$, 31 m a$^{-1}$ and 62 m a$^{-1}$ for time spans of 15–17 years, 1 year (TM), and 1 year (MSS), respectively.

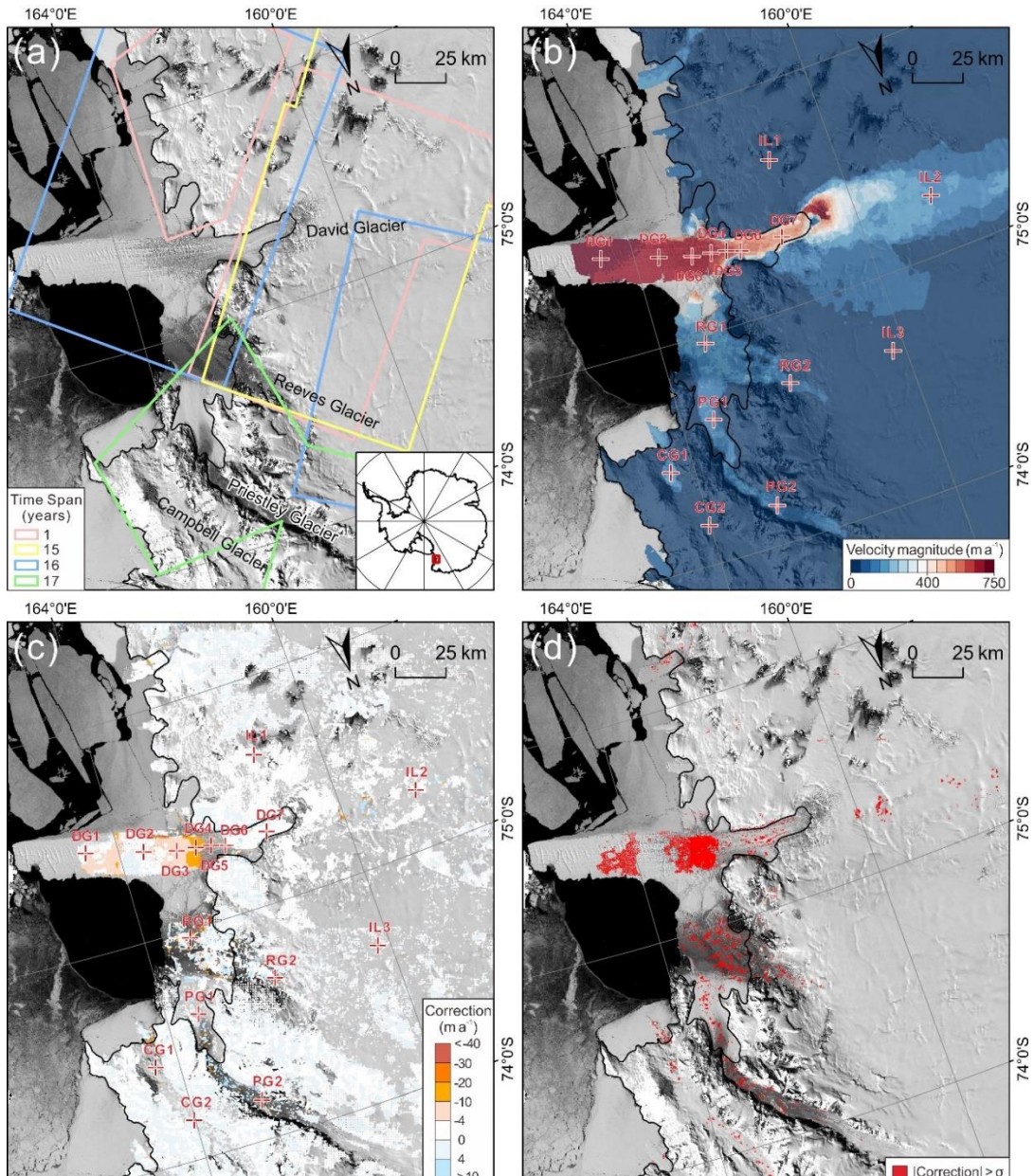

**Figure 6: Application of the overestimation correction method to a historical velocity map of 1972–1989 of the David Glacier region, East Antarctica. (a) Footprints and time spans of the Landsat image pairs used for velocity mapping with a background of Landsat Image Mosaic of Antarctica (Bindschadler et al., 2008), (b) produced velocity map of 1972–1989 and selected points for examination of velocity overestimation and correction, (c) overestimation values of all velocity points generated using the image feature matching technique, and (d) velocity points with OE ≥ 1σ (red points).**

At all velocity points, including 8,564 matched features from the feature matching process and 59,783 grid points from the dense grid matching process, we calculated the velocity overestimation values (Fig. 6c). The OEs are generally low on the grounded ice, mostly within ±4 m a⁻¹, which are less than the mapping uncertainties (1σ) of different time spans (Fig. 6d and

Table 2). The higher OEs ranging from ~4 m a⁻¹ to ~50 m a⁻¹ are on the ice tongue, ice shelves, and glaciers. We selected a total of 16 points from four glaciers and the inland region to examine detailed OEs in different areas of ice flow dynamics (Figs. 6b and 6c). The OEs at three points in the inland region (IL1–IL3) are negligible (Table 2). Points DG1–DG6 along the main trunk of the David Glacier present an increasing trend of OEs from 1 to 4 m a⁻¹ on the ice tongue up to 36 m a⁻¹ at the ice shelf outlet, as the average Lagrangian acceleration $\bar{a}\,(v^{L(n)})$ increases from 0.1 m a⁻² to 2.2 m a⁻². However, DG7 is located in a relatively steady velocity area, and thus has negligible overestimation. Furthermore, the OE values at the selected points on the smaller glaciers, Reeves Glacier (RG1 and RG2), Priestley Glacier (PG1 and PG2), and Campbell Glacier (CG1 and CG2), range from 0 m a⁻¹ to 12 m a⁻¹ and are generally greater than those on the grounded ice. At the velocity points where the OEs are greater than or equal to the velocity mapping uncertainties (1σ), the corrected velocities are calculated by applying the overestimation corrections.

**Table 2: Ice flow dynamics parameters, computed OEs, and corrected velocities at 16 selected points**

| ID | Time span (years) | $V_{Original}^{E}$ (m a⁻¹) | $1\,\sigma$ (m a⁻¹) | $\bar{a}\,(V^{L(n)})$ (m a⁻²) | Correction (-OE) (m a⁻¹) | $V_{Corr.}^{E}$ (m a⁻¹) |
|---|---|---|---|---|---|---|
| DG1 | 15.95 | 717 | 4 | 0.2 | -4 | 713 |
| DG2 | 15.95 | 692 | 4 | 0.1 | -1 | 692 |
| DG3 | 15.95 | 679 | 4 | 0.3 | -5 | 674 |
| DG4 | 15.95 | 665 | 4 | 0.8 | -12 | 653 |
| DG5 | 15.95 | 634 | 4 | 1.4 | -22 | 612 |
| DG6 | 15.95 | 572 | 4 | 2.2 | -36 | 536 |
| DG7 | 1.04 | 524 | 62 | 0.0 | 0 | 524 |
| RG1 | 16.96 | 125 | 4 | 0.7 | -12 | 113 |
| RG2 | 15.18 | 88 | 4 | 0.5 | -7 | 81 |
| PG1 | 16.96 | 110 | 4 | 0.4 | -7 | 103 |
| PG2 | 16.96 | 100 | 4 | 0.3 | -6 | 94 |
| CG1 | 16.96 | 143 | 4 | 0.3 | -5 | 138 |
| CG2 | 16.96 | 36 | 4 | 0.0 | 0 | 36 |
| IL1 | 1.00 | 8 | 31 | 0.0 | 0 | 8 |
| IL2 | 1.04 | 168 | 62 | 0.0 | 0 | 168 |
| IL3 | 16.16 | 15 | 4 | 0.0 | 0 | 15 |
| **MEAN** | **13.43** | **330** | **13** | **0.5** | **-7** | **323** |
| **STD** | **6.18** | **289** | **20** | **0.6** | **10** | **286** |

Overall, for a historical velocity map of 1972–1989 with time spans from 1 to 17 years in the David Glacier region, the overestimations and corrections mainly occur on the David Glacier and smaller glaciers where accelerations in the spatial

domain (not the temporal domain) exist, while those on the grounded ice are non-significant. We recommend that the overestimations that are greater than or equal to the mapping uncertainties ($1\sigma$) be corrected.

### 3.3 Experiment 3: Velocity overestimation correction at Pine Island Glacier, West Antarctica

We further apply the OE correction method at the Pine Island Glacier, one of the most dynamic glaciers in Antarctica. Four areas (1, 2, 3 and 4) along the main trunk and one area (5) in the margin in PIG are selected (Fig. 3a). The longest time span of 7 years (2013-2020) is determined, restricted by the very high velocity (up to ~4500 m a$^{-1}$) on the ice shelf, untraceable surface features over long spans, and availability of the high-quality Landsat 8 images. Hence, most of the ice shelf cannot be covered by a 7-year span velocity map and the five selected areas are near and upstream from the grounding line. The five

areas were all mapped with a baseline span of 3-month (Panels 1a-5a in Fig. A3), but different longest spans (n years, n = 6, 7, 7, 7, 3 for the five areas) (Panels 1b-5b in Fig. A3) because of difficulties in image feature tracking. Accordingly, in each area one surface feature was selected and tracked for n years to estimate E-velocity $V_{2013-i}^{E}$; its L-velocities of $V_{2013-i}^{L}$ and $V_{2013-i}^{L(n)}$ were computed from the three month-span and n-year span maps, respectively. Consequently, the resulted E-velocity (red lines), L-velocity from the three month-span maps (blue lines), and L-velocity from the n-span maps (black lines) are

illustrated in Panels 1c-5c in Fig. A3.

The mapped ground truth data of three-month and 7-year span velocities in five selected areas showed actual OE values in PIG from 18 m a$^{-1}$ in *Area 5* of the inland interior to 626 m a$^{-1}$ in *Area 2* near grounding line at glacier outlet (Table 3). Much of these OEs are attributed to the temporal acceleration patterns in Panels 1c-5c in Fig. A3 where the tracked E-velocities $V_{2013-i}^{E}$ of red lines are, in the later years, consistently above the L-velocities $V_{2013-i}^{L}$ of blue lines from the three-month span maps of

2013. These temporal accelerations were mainly caused by the drastic calving activities in and after 2017 in PIG and continuous basal melting in the Amundsen Sea sector during the study period (Joughin et al., 2020, 2021). Furthermore, the black and blue lines are approximately parallel (Panels 1c-5c in Fig. A3). Hence, the proposed method corrected in average 97 m a$^{-1}$ of the spatial acceleration - induced OE, which is ~40% of the overall OE (245 m a$^{-1}$) in five areas (Table 3). The average residual of 148 m a$^{-1}$, accounting for another 60% of the overall OE, represents the OE portion induced by the temporal acceleration in

Equation (9).


**Table 3. Application of the OE correction method in PIG. "Actual E-velocity and OE" includes E-velocities of 3-month and n-year spans and their differences as OEs. "Overestimation correction" presents n-year L-velocities from n-year span map, OE corrections, corrected E-velocities, and residuals (or errors) after correction.**

| Area ID (span) | Actual E-velocity and OE | | | Overestimation correction | | | |
| --- | --- | --- | --- | --- | --- | --- | --- |
| | $V_{3\,months}^{E}$ (m a$^{-1}$) | $V_n^E$ (m a$^{-1}$) | $OE_{Actual}$ (m a$^{-1}$) | $V_n^{L(n)}$ (m a$^{-1}$) | Correction (m a$^{-1}$) | $V_{Corr.}^E$ (m a$^{-1}$) | $\varepsilon$ (m a$^{-1}$) |
| **1** (6) | 3935 | 4054 | 119 | 4090 | -36 | **4018** | 83 |
| **2** (7) | 3278 | 3904 | 626 | 4099 | -195 | **3709** | 431 |
| **3** (7) | 2341 | 2532 | 191 | 2599 | -67 | **2465** | 124 |
| **4** (7) | 1651 | 1922 | 271 | 2103 | -181 | **1741** | 90 |
| **5** (3) | 392 | 410 | 18 | 416 | -6 | **404** | 12 |
| **MEAN** | 2319 | 2564 | 245 | 2661 | -97 | **2467** | 148 |
| **STD** | 1387 | 1506 | 232 | 1538 | 86 | **1477** | 163 |

In summary, OEs exist in historical long span velocity maps of Antarctica, from smaller glaciers, such as David Glacier, to the fast-flowing glacier of PIG, where spatial accelerations are produced by, e.g., bed topography and slopes. In general, they are less significant in slow-flowing grounded regions with low spatial accelerations. Instead, they take effects in places of high ice dynamics, for example, near grounding lines and often in ice shelf fronts. Velocities in these areas are important for estimating ice sheet mass balance and contribution to global sea level changes, and for analyzing ice shelf instability. For instance, in the

David Glacier the large OE corrections (up to 36 m a$^{-1}$) occur on the ice shelf (Fig. 6). The OEs of a 7-year span, up to 69 m a$^{-1}$ (Table 1), would be about ~50% of the velocity increase detected during 1989 – 2015 in the Totten Glacier (Li et al., 2016). The PIG experiment showed an extreme case in Antarctica where the OEs of a 7-year span can go as large as 626 m a$^{-1}$ (~20%) near the grounding zone (Table 3); furthermore, the OEs of a 15-year span can reach up to 1,300 m a$^{-1}$ along the grounding line and cause an overestimated GL flux of 11.5 Gt a$^{-1}$ if not corrected (Fig. A4). Therefore, the magnitude of the OEs contained

in the long span historical velocity maps is significant. When overestimated historical maps of 1960s – 1980s are used alongside recent maps of 1990s – 2010s for assessment of the long-term global climate change impact on the Antarctic ice sheet and for forecast modeling, the overestimated historical states may lead to underestimated long-term changes. Furthermore, compromised forecasting may be resulted. Thus, the OEs in the long span historical maps must be seriously examined and corrected.

**4. Discussion**

In this section we discuss the applicability of the proposed method in terms of overestimation-free time span, influence of complex glacier geometry, overestimation in fast flowing glaciers, and comparison with the "Midpoint" method.

### 4.1 Threshold of the overestimation-free time span for trajectory segments

The choice of a short time span $\Delta t_{Ref}$ (e.g., a few months to one year) for the "OE-free" segments along a trajectory in *Premise I* makes sure that the difference between the E-and L-velocities within the span is negligible, or less than σ (velocity mapping uncertainty). It is also the baseline time span of the initial "OE-free" velocity map. Determination of this threshold has an implication on validation of *Premises I* and *II*, as well as the integration period of the trajectory segments from $P_i$ to $P_{i+1}$ (Fig. 3b). Estimation of $\Delta t_{Ref}$ can be performed in each glacier by a linear regression between the E- or L-velocity $V$ and time span $\Delta t$, $V = K \Delta t + b$. Given $b$, $K$ and σ (20 m a$^{-1}$), $\Delta t_{Ref}$ can be calculated as $\Delta t_{Ref} = \frac{\sigma - b}{K}$. Our experiment results show $\Delta t_{Ref}$ as 3.2 years and 3.0 months for TG and PIG, respectively. Thus, the estimated OE-free time spans appear to be related to ice flow dynamics of the glaciers. In *Experiments 1 and 2*, we used 1 year for TG and 3 months for PIG, respectively. We suggest that a linear regression for $\Delta t_{Ref}$ estimation be performed before extensive historical velocity mapping would be carried out.

### 4.2 Glaciers with complex geometry

Within a long time span (e.g., over 5 - 10 years) in *Premise I* the difference between L- and E-velocities accumulated over all segments along the entire trajectory, i.e., the end-point deviation between the red and blue lines in Fig. 2, is measured with a more tolerable threshold: $(V_{0-n}^L - V_{0-n}^E) \leq k\ \sigma$. Although the OEs of the trajectory segments are controlled by $\Delta t_{Ref}$, ice mass moving along a curved flow line over this long span may result in an accumulative discrepancy. In *Experiment 1* we showed that the average difference $(V_{0-n}^L - V_{0-n}^E)$ in TG is within 1σ. Our further experiments in five smaller Antarctic glaciers with complex geometry, including the George VI, Abbott, Dotson, Crosson, and Getz glaciers, resulted in $(V_{0-n}^L - V_{0-n}^E)$ values that are negligible (smaller than *1*σ) in all five glaciers. We further investigated *Area 2* of PIG (Fig. 3a), a section with a high velocity of 3,278 m a$^{-1}$ and the highest curvature along the main trunk (7%, namely, a 1,305 m cross flow deviation from the 19,720 m long straight line). Based on the feature tracking result using eight annual Landsat 8 images, this curvature – induced E- and L-velocity difference is calculated as 206 m a$^{-1}$ over a 7-year span, among which 195 m a$^{-1}$ (95%) was corrected (*Corr.* for *Area 2* in Table 2). The results in these glaciers suggest that the velocity overestimation in glaciers with complex geometry can be mostly corrected and should not affect the applicability of the proposed method.

### 4.3 OE correction in fast-flowing glaciers

The Totten Glacier and Pine Island Glacier are among the most dynamic glaciers in Antarctica. The accelerated ice mass loss in these two fast-flowing glaciers has been realized by ice shelf basal melting and front calving (Li et al., 2015; Joughin et al., 2020, 2021). In *Experiments 1 and 3* we show that such temporal accelerations do not alter the necessary condition in *Premise II* significantly, so that $\left| \bar{a}(V^{L(n)}) - \bar{a}(V^L) \right| \leq k'\ \frac{\sigma}{\Delta t_n}$ holds with $k'$ = 1 and 3 in average for all experiment areas in TG (Table 1) and PIG (Table A4), respectively. The spatial acceleration – induced OEs can be corrected. On the other hand, the impact of temporal accelerations is represented in the sufficient condition in *Premise I*, $(V_{0-n}^L - V_{0-n}^E) \leq k\ \sigma$. For example, *Area 4* of

TG is located in a dynamic area of a velocity of ~1400 m a⁻¹, about 30 km from the ice shelf front (Fig. 4). It's E-velocity (red curve in Panel 4c in Fig. 5) exceeded the L-velocity (blue curve) in 2016 and showed a significant acceleration thereafter. This resulted in the largest actual overestimation $OE_{Actual}$ of ~69 m a⁻¹ (Table 1), which may be attributed to large shelf front calving occurring during the period between April 6 and September 20, 2015, with an area loss of 136 km² (2%; Fig. 4b). The L-velocities $V^L_{2013-i}$ (blue curve in Panel 4c of Fig. 5) are derived from the one-year span map $V_{2013-2014}$ before 2015 and thus, do not reflect this temporal acceleration; furthermore, in L-velocities $V^{L(7)}_{2013-i}$ (black curve) derived from the 7-year span map $V_{2013-2020}$ the effect of the calving induced acceleration is averaged out over the 7-year span. Thus, the back and blue lines are approximately parallel with an average accelerations difference of only -0.5 m a⁻² (Table 1). Consequently,, it appears that the calving event did not significantly reduce the buttressing potential of the ice shelf because the ice shelf front retreat occurred outside of the passive shelf-ice (PSI) boundary (light blue line in Fig. 4b; Fürst et al., 2016); the other four areas are located farther away, inland, and away from the PSI boundary and thus were not influenced as much as in *Area 4*. Therefore, the calving - induced temporal acceleration only occurred in *Area 4* ($k$ = ~1.5 in *Premise I*), but not in the four other areas ($k \leq$ 0.5). Hence, the calculated correction of -19 m a⁻¹ in *Area 4* is at a similar level to that in the four other areas and has only adjusted the OE caused by the spatial acceleration along the trajectory. The velocity increase due to the temporal acceleration by the calving event remains in the adjusted map as a large "residual" of 50 m a⁻¹, which can serve as a signature to study the relationship between calving activities and ice flow dynamics.

With $k$ = *1, 2, 2, 10, 5* in *Premise I* for the five selected areas from inland interior to grounding zone in PIG (Table A4), the temporal accelerations caused by climate warming impact are indicated by the high values of $k$ (10 and 5) near grounding line at the glacier outlet. Overall, the proposed method was able to correct ~40% of the total OEs, leaving another ~60% in residuals as the signature of the velocity changes induced by the temporal accelerations.

In addition, based on an annual E-velocity map of 2013 in PIG from ITS_LIVE (Fig. A4a), L-velocities along grounding line (GL; Gardner et al., 2018) with time spans of 1 to 15 years (Fig. A4b) and the associated ice discharge was computed. The actual flux gates were set with nodes separated every 240 m, which were located up to 13 km upstream the GL to reduce the uncertainty of ice thickness data (Gardner et al., 2019) from BedMachine (Morlighem et al., 2020). The results show that the velocity OEs of the 15-year span can reach up to ~1,300 m a⁻¹ in the GL region. Such OEs in velocity can further cause an overestimation in ice discharge cross flux gates upstream the GL, which increases rapidly by ~6.3 Gt a⁻¹ within the first 4-year span (Fig. A4c); thereafter the flux overestimation slows down until a maximum of 11.5 Gt a⁻¹ is reached at the 15-year span. This suggests that in fast-flowing glaciers like PIG, OEs in velocity maps with a time span of greater than 0.5 – 2 years should be corrected. The fact that the OE effect in PIG appears to level off after a couple of years is mainly because the velocity in the majority of the ice shelf is approximately leveled at ~4000 km a⁻¹ and the acceleration is thus very small. This makes the average L-velocities along the GL to increase in a much-reduced pace where an integral of E-velocities is performed in the leveled velocity area a few years after crossing the GL (Fig. A4d). Consequently, the flux OE indicated in Fig. A4c showed a similar trend. Extension of this exercise to the entire continent involves a complex situation, including deceleration caused by

ice rises, mapping errors near ice shelf shear margins, GL data points that would calve into the ocean within n-year span, and other potential factors. It should be noted that given a velocity range in flux computation, the ice flow angle between a flux gate and ice flow controls the flux magnitude. In principle, at the same gate this angle changes with velocities of different time spans and flow line patters, resulting in an overestimation or underestimation in flux for individual glaciers. A systematic and

470 in-depth investigation should be carried out to handle the overestimation issue in GL flux of the entire Antarctica.

**4.4 Comparison with the "Midpoint" method**

The "Midpoint" method presented in Berthier et al. (2003) compensates overestimations by assigning the overestimated velocities to middle points of the trajectories. We use the velocity measurements in *Experiment 1* to compare the performances of these two OE correction methods. Since *Area 4* was affected by a calving event during the time span, here we use other four

areas (*Areas 1, 2, 3,* and *5,* Fig. 4a). We estimated the E-velocities of 7 years (2013-2020) $V^E_{2013-2020}$ and assigned them to the midpoints of the trajectories in the four areas (Table A5). They were then compared to the one-year span velocity $V^E_{2013-2014}$ at the midpoints to calculate the bias $\varepsilon_M$. Similarly, the same overestimated 7-year span E-velocities $V^E_{2013-2020}$ were corrected using the OE correction method of this paper and assigned to the start points of the trajectories as $V_{corrected}$ (Table A5). They were then compared to the one-year span $V^E_{2013-2014}$ also, but at the start points to calculate another set of bias $\varepsilon_S$. As shown

in Table A5, the proposed OE correction method achieved a higher overall accuracy of $4 \pm 10$ m a$^{-1}$, compared to $12 \pm 14$ m a$^{-1}$ of the "midpoint" method.

In summary, our validation and application experiment results demonstrated that the proposed OE correction method can be applied to different types of glaciers in Antarctica, regardless of their ice flow dynamics. While the OEs are effectively corrected, temporal velocity increases caused by global warming – induced activities can be preserved for

long-term change studies. The implication is that, when using newer velocity maps of short spans along with historical maps of long spans produced in previous studies over the past few decades, the overestimation of historical velocities could have caused an underestimation in the long-term acceleration magnitude. On the other hand, new efforts in historical velocity mapping at an ice sheet – wide or large regional scale should be made with a full consideration of OE corrections. The applicability of the method to glaciers in Greenland should be further investigated in future research.

**5 Conclusions**

Velocity overestimation exists in Antarctic ice flow velocity maps produced from optical satellite images of long time spans. Such overestimations are inevitable for historical velocity maps due to the poor availability of earlier satellite images in Antarctica, especially before 1990. The results in this study show that the overestimations can reach up to ~69 m a$^{-1}$ in the Totten Glacier, East Antarctica, over a 7-year span and ~931 m a$^{-1}$ in the Pine Island Glacier, West Antarctica, over a 10-year

span. The overestimated historical velocity maps should be adjusted before they are combined with recent velocity maps to

build a long-term record for monitoring and forecasting the Antarctic ice flow dynamics and impact of global climate changes on the ice sheet. We used a set of "ground truth" velocity maps in the Totten Glacier produced from Landsat 8 images of 2013 to 2020 to validate the proposed innovative method for velocity overestimation correction. Based on the validated method, we successfully corrected the overestimations of a velocity map of 1972–1989 with time spans from 1 to 17 years in the David Glacier region, East Antarctica. Another experiment at the Pine Island Glacier (PIG) was carried out to demonstrate that the method can effectively adjust the overestimated velocities as large as 195 m a$^{-1}$, while preserving the velocity change signature for long-term ice flow dynamics analysis. In summary we draw the following conclusions.

1) The proposed Lagrangian velocity-based method is effective and easy to implement because the overestimation corrections are calculated by using the long-term velocity map itself only, without field observations or additional image data.

2) The premises of the correction method are validated by using a set of "ground truth" velocity maps developed from high-quality Landsat 8 images from 2013 to 2020 to show the rigidity of the method. The velocity overestimations of up to 7-year span are proven to be corrected effectively to within the uncertainty ($1\sigma$) of the 1-year map.

3) The validated correction method is then successfully applied to correct overestimations in a historical velocity map of 1972–1989 with time spans from 1 to 17 years.

4) It is proven that ice velocity change information of temporal acceleration events, e.g., caused by shelf front calving and basal melting in the Totten Glacier and Pine Island Glacier, is preserved after the correction and can be used for long-term ice flow dynamics analysis.

The magnitude of the OEs contained in the long span historical velocity maps are significant. When overestimated historical maps are used alongside recent maps for assessment of the long-term global climate change impact on the Antarctic ice sheet and for forecast modeling, the underestimated long-term changes and compromised forecasting may be resulted. Thus, the OEs in the long span historical maps must be seriously examined and corrected. It is recommended that OE values be computed for long-term historical velocity maps and corrections be made when OEs are more than the velocity uncertainty ($1\sigma$). This velocity overestimation correction method can be applied to adjust velocity fields for the production of regional and ice sheet-wide historical velocity maps from long-term satellite images before 1990s.

*Author contributions.* RL led the study and developed the overestimation correction model. YC, MX, XY, ZL and SL carried out velocity mapping. HC performed programing. RL, YC and GQ were involved in data analysis and presentation.

*Competing interests.* The authors declare that they have no conflict of interest.

*Acknowledgements.* We thank the editor and two reviewers for their constructive comments and suggestions. We also thank the United States Geological Survey (USGS) for the Landsat images.

*Financial support.* This research has been supported by the National Natural Science Foundation of China (41730102), the National Key Research and Development Program of China (2017YFA0603100), Chinese Arctic and Antarctic Administration (CXPT2020017), and the Fundamental Research Funds for the Central Universities.

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

690

**Appendix**

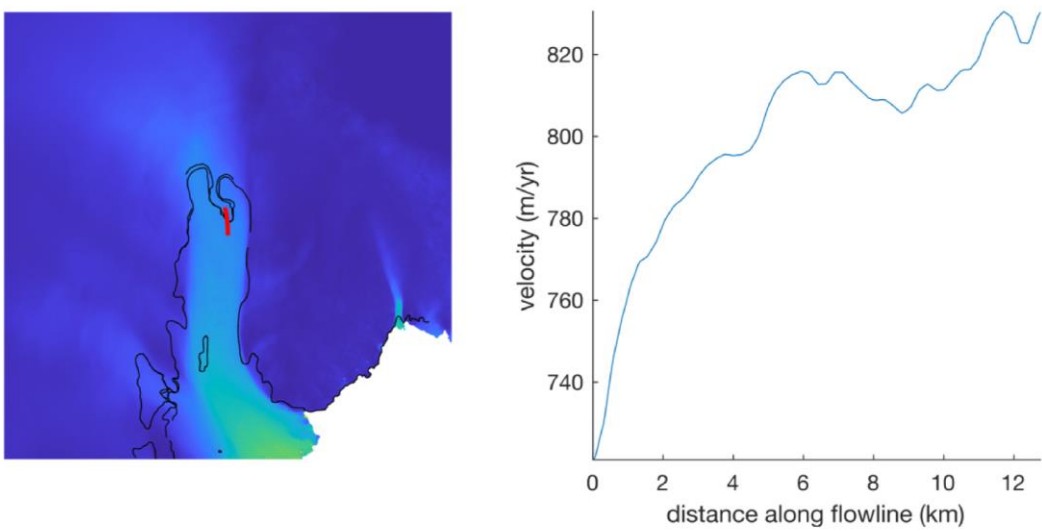

**Figure A1. Velocity map of the Totten Glacier from ITS_LIVE (left) used to explain the concept of velocity overestimation caused by acceleration (right):** "*Over 16 years, that parcel of ice travels about 13 km downstream (red path). It begins at a velocity of about 720 m/yr, and in the first 8 months it travels at an average rate very close to 720 m/yr. But then the ice picks up speed as it moves downstream, so in the first 10 years it does not travel just 7200 m—it actually travels about 7900 m, or an average speed of 790 m/yr......*" **(Greene, 2020b).**

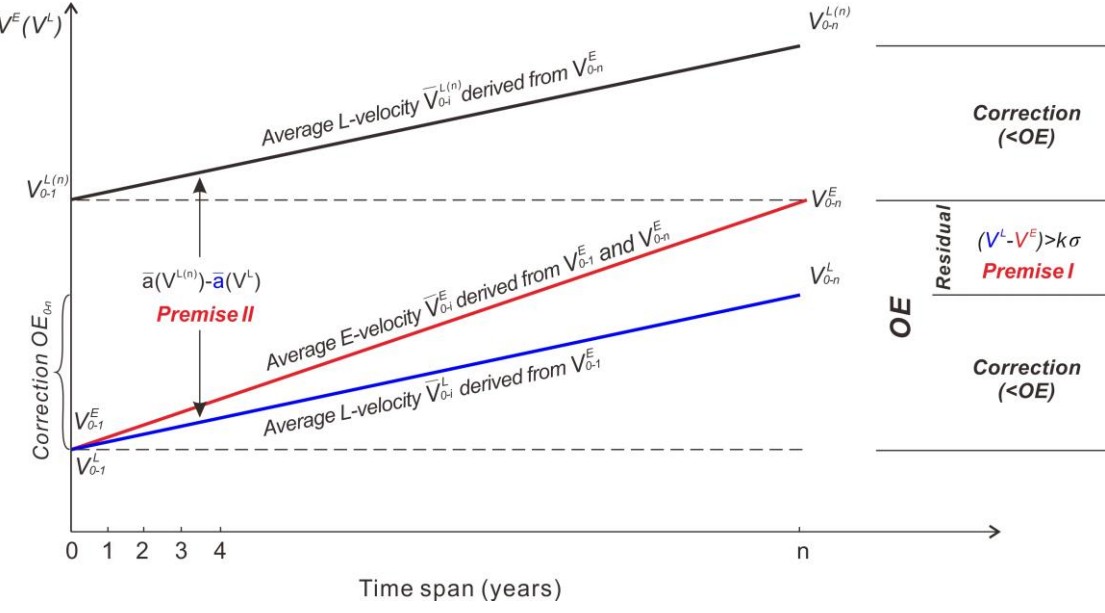

**Figure A2. Overestimation correction in case of temporal accelerations.**

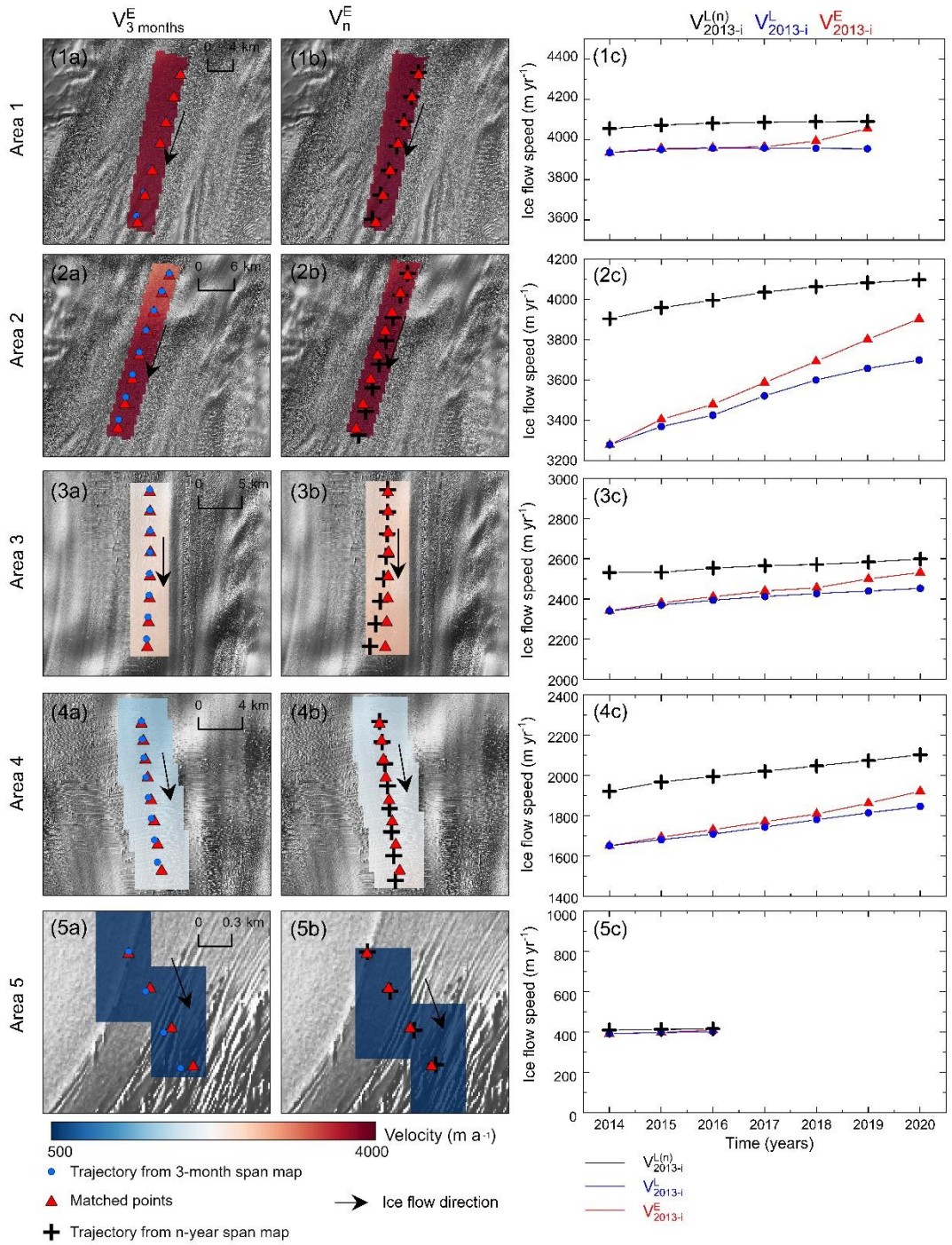

**Figure A3. Velocities in five areas (numbered yellow rectangles in Fig. 3) of PIG are used to demonstrate application of the OE correction method. (a) Panels 1a–5a show the reconstructed 3-month velocity maps $V_{2013-2014}$ in the areas; matched points (red**

triangles) are used to map E-velocity $V_{2013-i}^{E}$ ($i=2014$ (3 months), 2015, …, n) (red lines in Panels 1c-5c); points along the flow line (blue dots) are tracked from the 3-month maps and used to calculate L-velocity $V_{2013-i}^{L}$ (blue lines in Panels 1c-5c). (b) Similarly, Panels 1b–5b illustrate the reconstructed n-year velocity maps $V_{2013-n}$ in the areas with the matched points (red triangles) for E-velocity $V_{2013-i}^{E(n)}$; points along the flow line (black crosses) are tracked from the n-year span velocity map and used to calculate L-velocity $V_{2013-i}^{L(n)}$ (black lines in Panels 1c-5c). (c) In each area (Panels 1c–5c) the difference between red and blue lines increases with time; the black line is above blue line due to the spatial acceleration - induced OE over n-year span; but they are relatively parallel and thus, a correction can be estimated. Background in Panels a and b is Landsat 8 image of February 12, 2014.

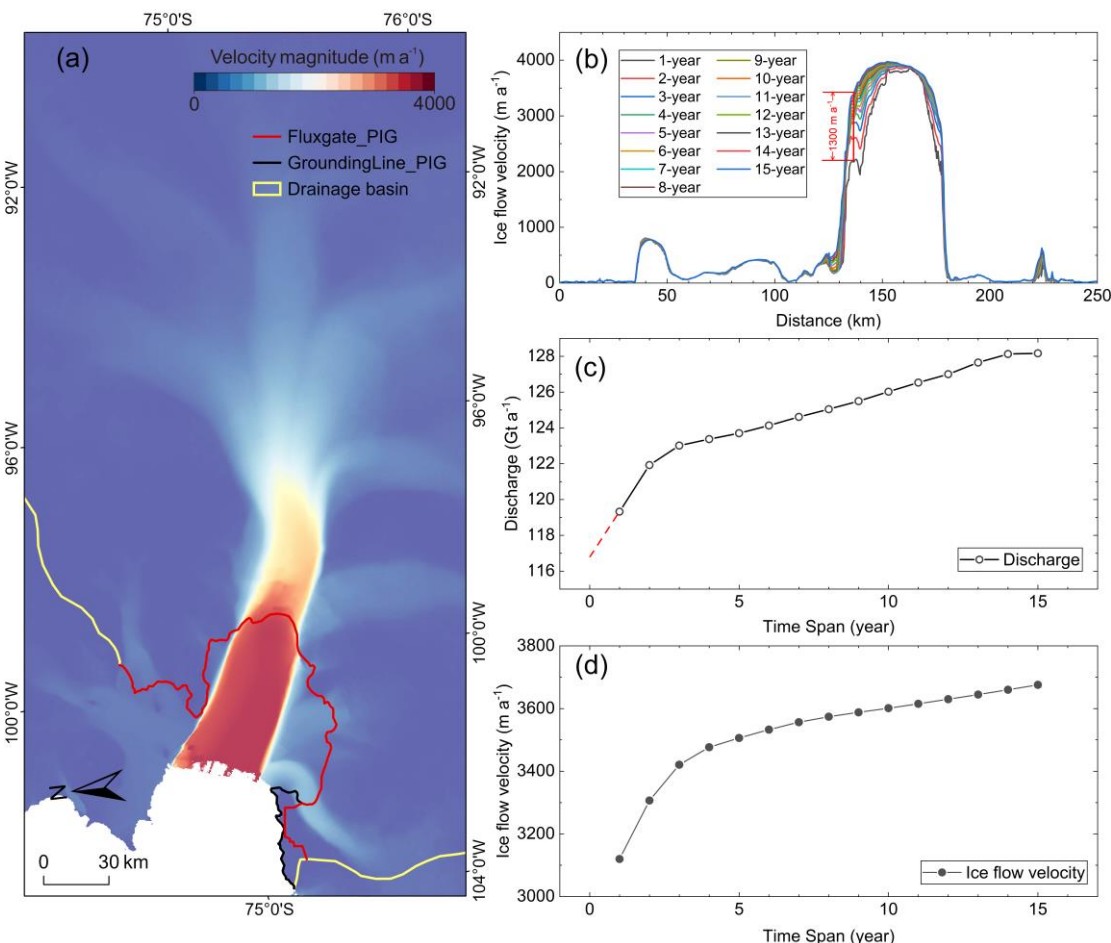

Figure A4. (a) One year span ice velocity map of PIG (2013) from ITS_LIVE (Gardner et al., 2019), (b) L-velocities of 1- to 15-year span $V_{0-i}^{L}$ ($i=1, 2, … 15$) along GL calculated from the 2003 velocity map, (c) Ice discharge cross flux gates upstream grounding line estimated with time spans from 1 to 15 years, and (d) L-velocities $\bar{V}_{0-i}^{L}$ (averaged over the GL portion across main trunk) vs. time span.

**Table A1.** Information for the Landsat 8 images used for velocity mapping in *Experiment 1*.

| Acquisition date (UTC) | Scene ID |
|---|---|
| 2013/12/03 | LC08_L1GT_102107_20131203_20170428_01_T2_B8 |
| 2014/11/13 | LC08_L1GT_101107_20141113_20170417_01_T2_B8 |
| 2015/12/25 | LC08_L1GT_102107_20151225_20170331_01_T2_B8 |
| 2016/11/25 | LC08_L1GT_102107_20161125_20170317_01_T2_B8 |
| 2017/11/05 | LC08_L1GT_101107_20171105_20171120_01_T2_B8 |
| 2018/12/17 | LC08_L1GT_102107_20181217_20181227_01_T2_B8 |
| 2019/12/29 | LC08_L1GT_101107_20191229_20200111_01_T2_B8 |
| 2020/10/19 | LC08_L1GT_102107_20201019_20201019_01_RT_B8 |

**Table A2.** Acceleration estimates using least squares regression $a$ and simple average $\overline{a}$ applied to L-velocities of 7-year $V^{L(n)}$ and 1-year $V^{L}$.

| Area | $a\,(V_{2013-i}^{L(n)})$ (m a$^{-2}$) | $\overline{a}\,\left(V_{2013,2020}^{L(n)}\right)$ (m a$^{-2}$) | Diff$(a-\overline{a})$ (m a$^{-2}$) | $a\,(V_{2013-i}^{L})$ (m a$^{-2}$) | $\overline{a}\,\left(V_{2013,2020}^{L}\right)$ (m a$^{-2}$) | Diff$(a-\overline{a})$ (m a$^{-2}$) |
|---|---|---|---|---|---|---|
| 1 | 4.1 | 3.4 | 0.7 | 4.9 | 4.6 | 0.3 |
| 2 | 1.8 | 1.6 | 0.2 | 0.7 | 0.4 | 0.3 |
| 3 | 4.4 | 3.6 | 0.8 | 6.6 | 5.7 | 0.9 |
| 4 | 4.6 | 3.4 | 1.2 | 5.3 | 3.9 | 1.4 |
| 5 | 5.1 | 4.5 | 0.6 | 7.1 | 6.0 | 1.1 |
| MEAN | 4.0 | 3.3 | 0.7 | 4.9 | 4.1 | 0.8 |
| STD | 1.3 | 1.1 | 0.4 | 2.5 | 2.2 | 0.5 |

725

**Table A3.** Information for the Landsat images used for velocity mapping in *Experiment 2*.

| Satellite | Acquisition date (UTC) | Scene ID |
|---|---|---|
| Landsat 1 (MSS) | 1972/11/28 | LM10641121972333AAA04 |
| | 1972/11/28 | LM10641131972333AAA04 |
| | 1972/11/28 | LM10651121972333FAK04 |
| | 1972/12/1 | LM10671121972336AAA04 |
| | 1972/12/1 | LM10671131972336AAA04 |
| | 1972/12/2 | LM10681121972337AAA04 |

| | |
|---|---|
| 1972/12/5 | LM10711121972340AAA02 |
| 1972/12/11 | LM10781131972346FAK03 |
| 1973/1/2 | LM10641141973002AAA04 |
| 1973/1/3 | LM10831131973003AAA04 |
| 1973/1/16 | LM10611151973016AAA04 |
| 1973/1/17 | LM10781141973017XXX01 |
| 1973/1/25 | LM10681131973025FAK03 |
| 1973/1/31 | LM10741121973031FAK03 |
| 1973/1/31 | LM10741131973031FAK03 |
| 1973/1/31 | LM10741151973031FAK03 |
| 1973/2/6 | LM10801151973037AAA05 |
| 1973/2/8 | LM10641151973039AAA02 |
| 1973/2/9 | LM10831141973040XXX01 |
| 1973/2/11 | LM10671161973042FAK07 |
| 1973/2/12 | LM10861121973043FAK03 |
| 1973/2/14 | LM10701121973045AAA05 |
| 1973/2/14 | LM10701131973045AAA05 |
| 1973/2/20 | LM10761121973051FAK03 |
| 1973/2/28 | LM10661141973059AAA04 |
| 1973/2/28 | LM10661161973059AAA04 |
| 1973/3/6 | LM10721131973065XXX01 |
| 1973/10/29 | LM10751121973302AAA02 |
| 1973/10/29 | LM10751131973302AAA02 |
| 1973/11/4 | LM10811131973308FAK03 |
| 1973/11/4 | LM10811141973308FAK03 |
| 1973/11/7 | LM10841141973311FAK03 |
| 1973/11/21 | LM10801131973325FAK03 |
| 1973/11/24 | LM10641161973328AAA04 |
| 1973/12/20 | LM10731151973354FAK03 |
| 1973/12/25 | LM10601161973359AAA04 |
| 1973/12/31 | LM10661141973365AAA04 |
| 1973/12/31 | LM10661161973365FAK02 |
| 1973/12/31 | LM10671141973365FAK02 |
| 1973/12/31 | LM10671151973365FAK01 |
| 1974/1/5 | LM10711121974005AAA04 |
| 1974/1/5 | LM10711131974005AAA04 |
| 1974/1/12 | LM10781141974012AAA04 |
| 1974/1/12 | LM10781151974012AAA04 |
| 1974/1/14 | LM10621161974014AAA04 |
| 1974/1/16 | LM10641141974016AAA02 |
| 1974/1/18 | LM10661151974018AAA04 |
| 1974/1/20 | LM10861121974020AAA04 |

| | 1974/1/25 | LM10731121974025AAA05 |
|---|---|---|
| | 1988/12/15 | LT40601141988350XXX04 |
| | 1988/12/15 | LT40601151988350XXX03 |
| | 1989/1/29 | LT40551161989029XXX04 |
| | 1989/3/24 | LT40651131989083XXX01 |
| | 1989/3/24 | LT40651141989083XXX11 |
| | 1989/3/24 | LT40651151989083XXX01 |
| Landsat 4 (TM) | 1989/11/12 | LT40641151989316XXX01 |
| | 1989/11/14 | LT40621121989318XXX02 |
| | 1989/11/14 | LT40621131989318XXX02 |
| | 1989/11/26 | LT40661121989330XXX01 |
| | 1989/11/30 | LT40621151989334XXX01 |
| | 1989/12/5 | LT40651121989339XXX02 |
| | 1989/12/16 | LT40621141989350XXX02 |
| Landsat 5 (TM) | 1986/1/4 | LT50561161986004XXX04 |
| | 1986/12/13 | LT50571161986347XXX05 |

**Table A4. Application of the OE correction method in five selected areas in PIG. "*Premise I*" contains n-year tracked E-velocities and L-velocities computed from the 3-month velocity map, and their differences. "*Premise II*" illustrates averaged L-accelerations computed from the n-year and 3-month velocity maps, respectively, as well as their differences.**

| Area ID | Time span (years) | Premise I | | | Premise II | | |
|---|---|---|---|---|---|---|---|
| | | $V_n^E$ (m a$^{-1}$) | $V_n^L$ (m a$^{-1}$) | $(V^L - V^E)_n$ (m a$^{-1}$) | $\overline{a}\,(V^{L(n)})$ (m a$^{-2}$) | $\overline{a}\,(V^L)$ (m a$^{-2}$) | $\overline{a}\,(V^{L(n)}) - \overline{a}\,(V^L)$ (m a$^{-2}$) |
| 1 | 6 | 4054 | 3954 | **-100** | 6.0 | 3.2 | **2.8** |
| 2 | 7 | 3904 | 3699 | **-205** | 28.2 | 60.8 | **-32.6** |
| 3 | 7 | 2532 | 2453 | **-79** | 9.5 | 15.9 | **-6.4** |
| 4 | 7 | 1922 | 1847 | **-75** | 25.6 | 27.8 | **-2.2** |
| 5 | 3 | 410 | 402 | **-8** | 2.2 | 3.6 | **-1.4** |
| MEAN | 6 | 2564 | 2471 | **-93** | 14.3 | 22.3 | **-8.0[*]** |
| STD | 2 | 1506 | 1447 | **71** | 11.8 | 23.8 | **14.2** |

[*]With an average span of 6 years for all areas and $\sigma = 20$ m a$^{-1}$, the mean of $\overline{a}\,(V^{L(n)}) - \overline{a}\,(V^L)$ in "*Premise II*" is 8 m a$^{-2}$ ($\leq 3\frac{\sigma}{\Delta t_n}$).

**Table A5. Comparison of the proposed OE correction method with the "Midpoint" method in Berthier et al. (2003)**

| Area ID | OE velocity assigned to midpoint<br>$V^E_{2013-2020}$<br>($m\ a^{-1}$) | 1-year map at midpoint<br>$V^E_{2013-2014}$<br>($m\ a^{-1}$) | Bias<br>$\varepsilon_M$<br>($m\ a^{-1}$) | OE-corrected velocity assigned to start point<br>$V_{corrected}$<br>($m\ a^{-1}$) | 1-year map at start point<br>$V^E_{2013-2014}$<br>($m\ a^{-1}$) | Bias<br>$\varepsilon_S$<br>($m\ a^{-1}$) |
|---|---|---|---|---|---|---|
| 1 | 843 | 860 | 17 | 824 | 824 | 0 |
| 2 | 1008 | 1007 | -1 | 999 | 1007 | 8 |
| 3 | 1317 | 1332 | 15 | 1297 | 1280 | -17 |
| 5 | 789 | 807 | 18 | 759 | 754 | -5 |
| MEAN | 989 | 1001 | 12 | 970 | 966 | -4 |
| RMSE | | | 14 | | | 10 |