# Peer review of "Overestimation and Adjustment of Antarctic Ice Flow Velocity Fields Reconstructed from Historical Satellite Imagery"

_The Cryosphere, 2021_

## Referee Comment (RC2)

**Overestimation and Adjustment of Antarctic Ice Flow Velocity Fields**
**Reconstructed from Historical Satellite Imagery**
by Rongxing Li et al.

*Review by Chad A. Greene, NASA/JPL.*

This paper identifies three key shortcomings of a common velocity measurement technique, and provides a solution that addresses all three. At issue are 1. the true location of a feature-tracked velocity measurement, 2. the acceleration that a parcel of ice may experience between image acquisition times, and 3. the fact that ice does not always move in a perfectly straight line between image acquisition times. The problem and solution are described well in this paper, and the authors demonstrate that they have a good handle on the data and how velocities are interpreted from a glaciological standpoint.

This work will be of value to the community, both to raise awareness of the overestimation issue, and to provide a solution to it. I recommend publication, with just a few suggestions that may help readers understand the impact of overestimation and how it should affect our interpretation of previous studies.

**Main issues**

The paper does a good job of describing the problem and solution from a technical standpoint, and anyone who has written feature-tracking algorithms will benefit from reading the paper. However, there are many readers who don't write their own algorithms, but will nonetheless want to understand how overestimation might affect their scientific results. Some work could be done in this paper to better communicate the overall impact of how overestimation impacts long-term studies.

Here's a type of analysis that I would find much more insightful than the stats for PIG, Totten, and David Gl that are currently presented in the abstract: I would like to see a figure showing Eulerian grounding line flux calculations as a function of dt, where dt might range from a day to 20 years. This would provide readers with some intuition for a threshold value of dt, beyond which Eulerian measurements produce significantly different estimates of ice flux. It's possible that the percentage reduction in GL flux as a function of dt might vary regionally, and that diversity could be interesting to show as well.

In addition to a figure showing how dt affects GL flux in Eulerian measurements, I'd like some clear guidance in the abstract for when the Eulerian approximation is sufficient or insufficient.

In the abstract and/or discussion, I suggest flipping the logic/wording around at least once to make it clear that the overestimation of historical velocities could mean that previous papers have *underestimated* the magnitude of glacier acceleration over the past few decades. It's only

a minor change in wording, but I think it's an important take-home message of this paper that should be stated directly.

**Minor comments**

**Abstract:** The case studies of PIG, Totten, and David Glacier provide decent testing grounds for the methods presented in this paper, but the details of these studies feel somewhat anecdotal and very specific to the exact images that were used in these particular cases. I recommend generalizing the results in the abstract to give readers a better overview of the problem. Only after discussing the overall impact of the overestimation, then it may be helpful to mention a specific case of PIG, Totten, or David to as a tangible example.

**L29:** This line mentions "the input-output method" and some good references are provided for it, but some readers may be unfamiliar with the term. If the term is necessary for some point that's being made, then I recommend briefly describing what is meant by "the input-output method" here. If the term is not important for this paper, then consider removing it.

**L69:** Recommend changing "It is proven that…" to "We show that…" to make it clear that the correction is original work that is presented in this paper.

**L80:** I'm not entirely sure what "descending passages" means. Consider rewording.

**L160**: "At each grid…" I think this should be "At each grid cell…" or "At each pixel…"

**Figure 5** is very compelling, and I want to make sure I understand it. Unfortunately, the labels and caption are somewhat cryptic, so I'm not sure if I'm even getting the main message right. The caption contains a list of the data labels that are mostly redundant with labels that are presented directly in the figure. What's missing is physical interpretation or any direct take-home message. For example, the variables U, U', and V are labeled in the figure and in the caption, but there's no physical definition of what U, U', or V mean. Help readers by providing a sentence or two in the caption that directly states the main point and any secondary point(s) that may be worth noticing. The main point, I assume, is that the black line is consistently higher than the red and blue curves. State that in the caption, in terms of what it means physically. What causes the red and blue curves to cluster together or spread apart from each other? Mention the underlying mechanism in the caption. Most of these points are described in detail on page 11, but most readers will appreciate having the main points stated directly in the figure caption.

---

## Author Comment (AC1)

We appreciate the constructive comments from two reviewers and the community. Our manuscript will be much improved by their input. We have made changes to our manuscript. In the following responses, we use "**bold**" text for comments, "non-bold" text for our responses, and "*italic*" for changed text in the manuscript.

**Referee #1**

**General comments**

**The paper by Li et al., 2021 presents an innovative method that aims at correcting glacier velocity overestimation, that are due to accelerations, when using long timespan. The paper is well presented with a clear structure, well written and the Figures are relatively clear.**

**A large share of the paper is dedicated to the description of the method, which is simple in principle, but that could be ambiguous to understand clearly. Consequently, I have few comments that I hope, will help to make the paper more understandable.**

**Among those comments, the definition of the different Premises needs a bit of clarification, and particularly on their area of validity (see below)……**

Response:
In *Premises I and II* we clarified the correspondence between the short span (months to year) and "OE-free" ($< \sigma$) trajectory segments, as well as that between the longer span (5 to over 10 years) and difference of E- and L-velocities ($< k\,\sigma$) at the end of a trajectory.

*"**Premise I:** Within a period of n short time spans (e.g., 1 year or shorter) each segment (from $P_{i\text{-}1}$ to $P_i$ in Figure 1b) along a flow line is relatively short and straight, so that their accumulated curved L-distance $S_{0\text{-}i}$ over a longer span (e.g., over 5-10 years) is not significantly different from the corresponding straight E-distance $D_{0\text{-}i}$; furthermore, their averaged velocity trends of $V_{0-i}^L$ in the Lagrangian framework and $V_{0-i}^E$ in the Eulerian framework (blue and red lines in Fig. 2) do not deviate significantly from each other, and their maximum difference is limited within a threshold ($V_{0-n}^L - V_{0-n}^E \leq k\,\sigma$), where k is a constant and $\sigma$ is the velocity mapping uncertainty."*

*"**Premise II:** Within the time span of n years (e.g., over 5-10 years), the velocity field described by $V_{0-1}^E$ and $V_{0-n}^E$ does not change significantly, so that the line between $V_{0-n}^{L(n)}$ and $V_{0-1}^{L(n)}$ (black line in Fig. 2) and that between $V_{0-n}^L$ and $V_{0-1}^L$ (blue line in Fig. 2) are approximately parallel to each other. Accordingly, the difference between their simple averaged accelerations is within a threshold ($\left|\overline{a}\left(V^{L(10)}\right) - \overline{a}(V^L)\right| \leq k'\,\frac{\sigma}{\Delta t_n}$), where k' is a constant and $\sigma$ is the velocity mapping uncertainty and*

$$\overline{a}\left(V^{L(10)}\right) = \frac{V_{0-n}^{L(n)} - V_{0-1}^{L(n)}}{\Delta t_n}, \quad \overline{a}(V^L) = \frac{V_{0-n}^L - V_{0-1}^L}{\Delta t_n}.\text{''}$$

Their validity in different types of glaciers is introduced in the Discussion section (see following responses).

**…… The discussion also needs to be supplemented with an overview of the method applicability to different glacier types, and specifically fast flowing glaciers (e.g Jakobshavn Isbrae), or with a more complex geometry (e.g Zachariae Isstrøm ice shelf, Getz Ice Shelf or George VI ice shelf). Another interesting point of discussion is the impact of the glacier seasonal variability. Are the corrections significant with respect to the natural variability of glacier flow? While, seasonal signals are not really pronounced in Antarctica, variability can be much greater in Greenland (cf. Joughine et al., 2020).**

**For example, using a 1-year velocity reference for a glacier like Jakobshavn Isbrae, might not be ideal, as the glacier is flowing at more than 15 km/yr (which increases the chances of acceleration along a flowline). Similarly does the premises still holds, for glaciers that are changing directions and not flowing in a straight line (for example the ice shelf of Zachariae Isttrøm before 2000)?**

Response:
The Discussion section is restructured. At the beginning we added an overview statement: "*In this section we discuss the applicability of the proposed method in terms of overestimation-free time span, influence of complex glacier geometry, overestimation in fast flowing glaciers, and preserve of historical glacier change signature.*"

Then we added three new subsections to make a strong Discussion section according to the comments:

*"**4.1 Threshold of the overestimation-free time span for trajectory segments**

*The choice of a short time span $\Delta t_{Ref}$ for the "overestimation-free" segments along a trajectory in Premises I makes sure that the difference between the E-and L-velocities within the span (e.g., a few months to one year) is negligible, or less than $\sigma$ (velocity mapping uncertainty). It is also the time span of the initial "OE-free" E-velocity map that is used for ice mass tracking and L-velocity computation in premise validation. Determination of this threshold has an implication on validation of Premise I, as well as the integration period of the trajectory segments from $P_i$ to $P_{i+1}$ (Fig. 3b). Estimation of the OE-free time span $\Delta t_{Ref}$ can be performed by a linear regression between the E- or L-velocity V and time span $\Delta t$, $V = K \Delta t + b$. Given $\sigma$ (20 m a$^{-1}$), $\Delta t_{Ref}$ can be calculated as $\Delta t_{Ref} = \frac{\sigma - b}{K}$. Our experiment results show $\Delta t_{Ref}$ as 3.2 years, 3.0 months, and 1.4 months for TG, PIG, and Jakobshavn Isbrae (JI), Greenland, respectively. Thus, the estimated OE-free time spans appear to be related to ice flow dynamics of the glaciers. In Experiments 1 and 3, we used 1 year for TG*

*and 3 months for PIG, respectively. We suggest that a linear regression for $\Delta t_{Ref}$ estimation be performed before extensive historical velocity mapping would be carried out.*

**4.2 Glaciers with complex geometry and E- and L-velocity difference along trajectory**

*Furthermore, within a longer time span (e.g., over 5 - 10 years) in Premise I the difference between E- and L-velocities accumulated over all segments along the entire trajectory $\Delta V^{L-E}$, i.e., the end-point deviation between the red and blue lines in Fig. 2, is measured with a more tolerable threshold of $k$ times of $\sigma$ ($k$ $\sigma$). Although the OEs of the trajectory segments are controlled by $\Delta t_{Ref}$, ice mass moving along a curved flow line over this long span may result in an accumulative discrepancy. In Experiment 1 we showed that the E- and L-velocity difference $\Delta V^{L-E}$ in TG are within $2\sigma$. Our further experiment in five smaller Antarctic glaciers with complex geometry, including the George VI, Abbott, Dotson, Crosson, and Getz glaciers, resulted in $\Delta V^{L-E}$ values that are negligible (smaller than $1\sigma$) in all five glaciers. Thus, if Premise I is met (e.g., $k \leq 2$) there are mainly spatial acceleration-induced OEs, which can be effectively corrected. We found that in places, such as PIG in Experiment 3 and Area 4 in TG in Experiment 1, where temporal accelerations caused by basal melting and calving activities (Li et al., 2015; Joughin et al., 2020, 2021) exist the threshold $k$ exceeded 2. However, the computed OE corrections can still remove the spatial acceleration-induced OE portion and leave the temporal acceleration-induced OE portion as a signature for long-term ice dynamics studies. Hence, Premise I is not a necessary condition, but a sufficient condition. Furthermore, the largest curvature-induced OE in PIG is 206 m a$^{-1}$ over a 7-year span trajectory, among which 195 m a$^{-1}$ (95%) was corrected.*

**4.3 OE estimation in fast flowing glaciers**

*We estimate OE corrections assuming that within a longer time span in Premise II the acceleration trend would not change significantly during the time span, $\left|\bar{a}\left(V^{L(n)}\right) - \bar{a}\left(V^L\right)\right| \leq k' \frac{\sigma}{\Delta t_n}$. As shown in Experiments 1 and 2, the acceleration trend difference is under $1 \frac{\sigma}{\Delta t_n}$ for both David Glacier and Totten Glacier that is one of the most dynamic glaciers in East Antarctica. Since the velocity requirements in Premise I and acceleration requirement in Premise II were met properly, we were able to correct in average 88% of the OEs in TG (up to 69 m a$^{-1}$, Table 1). Furthermore, in the fast-flowing PIG the acceleration trend differences $\left|\bar{a}\left(V^{L(n)}\right) - \bar{a}\left(V^L\right)\right|$ in all 5 areas are in average less than $3 \frac{\sigma}{\Delta t_n}$. Correspondingly, the black and blue lines of all 5 areas appear parallel (Panels 1c-5c in Fig. R1-8), indicating that the acceleration condition in Premise II is properly met. Consequently, out of the total average OE of 245 m a$^{-1}$ the proposed method effectively corrected the spatial acceleration-induced*

*portion of 97 m a⁻¹ (~40%), leaving the temporal acceleration-induced portion of 148 m a⁻¹ (60%) in residuals. The uncorrected portion of OEs represent the velocity change signature over the time span caused by the continuous basal melting and drastic calving activities in and after 2017 in PIG (Joughin et al., 2021). Finally, our experiment results show that in Jakobshavn Isbrae the grounding zone and floating ice part cannot be covered by multi-year span maps (over 2-3 years) because of the relatively short main trunk (~60 km) and extremely high velocity (~15000 m a⁻¹), i.e., lost opportunities for historical velocity recovery. A comprehensive study is needed to investigate the influence of the extremely high ice flow dynamics on the proposed OE correction method in Greenland."*
* * *
The above revision of the manuscript is supported by the results of a number of new experiments and data analysis. In the following we provide a detailed version:

4.1 Threshold of the overestimation-free time span for trajectory segments

The choice of a short time span $\Delta t_{Ref}$ (e.g., a few months to a year) for the "overestimation-free" segments along a trajectory in *Premises I* makes sure that the difference between E-and L-velocities within the span is negligible, or less than σ (velocity mapping uncertainty, σ=20 m a⁻¹ in this study). It is also the time span of the initial "OE-free" E-velocity map that is used for ice mass tracking and L-velocity computation in premise validation. Determination of this threshold has an implication on validation of *Premise I*, as well as the integration period of the trajectory segments from $P_i$ to $P_{i+1}$ (Fig. 3b). Estimation of $\Delta t_{Ref}$ can be performed in a systematic way. An area of the highest acceleration in a glacier should be selected. Within the area a multi-span E-velocity series $V_{0-i}^E$ (*i=1, 2, ... n*) can be used to establish a linear relationship between the E-velocity $V^E$ and time span $\Delta t$, $V^E = K \Delta t + b$, by a linear regression (red line in Fig. 2). With the known parameters of *b* and *K*, $\Delta t_{Ref}$ can be calculated as $\Delta t_{Ref} = \frac{\sigma - b}{K}$. For example, *Area 5* in TG in *Experiment 1* has the highest acceleration (Table 1, Fig. 5c). After a regression using the E-velocities in *Area 5*, $\Delta t_{Ref}$ is calculated as 3.2 years ($R^2$=0.96, Fig. R1-1a). Thus, the selected $\Delta t_{Ref}$ of one year for TG in *Experiment 1* is justified.

Alternatively, if the E-velocities are not available, multi-span L-velocities along a profile on the main trunk of a glacier may be established from an available short span E-velocity map (e.g., 1-year map in PIG, Fig. R1-2a). Along the profile the highest acceleration (location "A" in Fig. R1-2a) is localized where a multi-span L-velocity series can be computed (Fig. R1-2b). Using this L-velocity series and the above regression method, the "OE-free" time span $\Delta t_{Ref}$ can also be estimated. For example, given σ=20 m a⁻¹ and a series of computed multi-span L-

velocities from 1- to 10-years at location "A" in PIG (Fig. R1-2) and "B" in Jakobshavn Isbrae (JI, Fig. R1-6), we estimated $\Delta t_{Ref}$ as ~3.0 months ($R^2$=0.99, Fig. R1-1b) and ~1.4 months ($R^2$=0.86, Fig. R1-1c) for PIG and JI, respectively.

Based on the above analysis results of $\Delta t_{Ref}$ in TG (3.2 years), PIG (3.0 months) and JI (1.4 months), it appears that the threshold of an "OE-free" time span is strongly related to the ice flow dynamics of the glaciers. Given a known σ, shorter spans of $\Delta t_{Ref}$ should be selected for trajectory segments in validation of *Premise I* and L-velocity integration in faster flow glaciers. We suggest that an analysis of multi-span L-velocities and a regression for $\Delta t_{Ref}$ be performed before extensive historical velocity mapping would be carried out.

[Figure]

Figure R1-1. Linear regression of multi-span velocities vs. time span is performed to estimate the "OE-free" time span. (a) Totten Glacier (TG): 7-years of E-velocities in *Area 5* (Fig. 4), $\Delta t_{Ref}$=3.2 years; (b) Pine Island Glacier (PIG): 10-years of L-velocities at location "A" near grounding line (Fig. R1-2), $\Delta t_{Ref}$=3.0 months; and (c) Jakobshavn Isbrae (JI): 10-years of L-velocities at location "B" along the main trunk profile (Fig. R1-6), $\Delta t_{Ref}$=1.4 months.

[Figure]

Figure R1-2. (a) Velocity, grounding line, and profile position along the main trunk of PIG; (b) E-velocity of 1-year span along the profile from ITS_LIVE, and L-velocities of 2-, 3-, and 10-year spans calculated from the 1-year span E-velocity map; (c) L-acceleration of the corresponding time spans along the profile, and (d) Estimated OEs caused by the L-velocities of different time spans. Please note that the L-velocities are only used for simulation estimation of time span $\Delta t_{Ref}$. In validation of *Premises I* and *II*, we used actual measurements of multi-year span E-velocities.

4.2 Glaciers with complex geometry and E- and L-velocity difference along trajectory

Furthermore, within a longer time span (e.g., over 5 - 10 years) in *Premise I* the difference between the E- and L-velocities accumulated over all segments along the entire trajectory, i.e. the end-point deviation between the red and blue lines in Fig. 2, is measured with a more tolerable threshold of $k$ times of $\sigma$ ($k$ $\sigma$). Although the OEs of the trajectory segments are controlled by $\Delta t_{Ref}$, ice mass moving along a curved flow line over this long span may result in an additional discrepancy along the entire trajectory.

As suggested, we examined the complex geometry issue in 5 Antarctic glaciers, including George VI, Abbott, Dotson, Crosson and Getz (Fig. R1-3a) where the velocity ranges from ~100 m a$^{-1}$ to 1,000 m a$^{-1}$. We used one-year velocity maps of 2013 of 5 glaciers from ITS_LIVE to derive 20-year span L-velocities along trajectories in the significantly curved sections of the glaciers (Fig. R1-4). The computed straight and curved lengths vs. time span in

the 5 glaciers are illustrated in Fig. R1-3b. The statistics of the corresponding E- and L-velocities are given in Table R1-1.

[Figure]

Figure R1-3. (a) Locations of 5 glaciers with complex geometry, including George VI, Abbott, Dotson, Crosson, and Getz. (b) Computed straight (displacement) and curved (trajectory) lengths vs. time span in 5 glaciers.

[Figure]

Figure R1-4. (a-e) locations of trajectories in 5 glaciers with complex geometry, including

George VI, Abbott, Dotson, Crosson, and Getz. (f-j) 20-year span trajectories in 5 glaciers.

It is shown in Table R1-1 that in these 5 glaciers three glaciers (Abbott, Dotson and Getz) have very small OEs in the 20-year span L-velocities ($< \sigma = 20$ m a$^{-1}$), and their "curvature" induced differences $\Delta V_{20\,y}^{L-E}$ are 0 m a$^{-1}$. The other two glaciers (George VI and Crosson) have OEs of 114 m a$^{-1}$ and 143 m a$^{-1}$, respectively. However, the "curvature"-induced differences are only 11 m a$^{-1}$ and 6 m a$^{-1}$, both of which are smaller than $\sigma$ and thus, negligible. So, the OEs are mainly caused by spatial acceleration here, not the curvature.

Table R1-1. Overestimations and "curvature" induced velocity differences in 5 glaciers

| ID | Name | $V_{1\,y}^{E}$ (m a$^{-1}$) | $V_{20\,y}^{L}$ (m a$^{-1}$) | OE (m a$^{-1}$) | $\Delta t$ (year) | E-dist. (m) | L-dist. (m) | $\Delta V_{20\,y}^{L-E}$ (m a$^{-1}$) |
|---|---|---|---|---|---|---|---|---|
| 1 | George VI | 507 | 621 | -114 | 20 | 12412 | 12633 | 11 |
| 2 | Abbott | 110 | 115 | -6 | 20 | 2306 | 2308 | 0 |
| 3 | Crosson | 1054 | 1197 | -143 | 20 | 23942 | 24064 | 6 |
| 4 | Dotson | 280 | 290 | -10 | 20 | 5803 | 5809 | 0 |
| 5 | Getz | 315 | 324 | -9 | 20 | 6481 | 6482 | 0 |

In the Totten Glacier flow lines are less curved and velocity is higher (up to ~1,400 m a$^{-1}$). The E- and L-velocity differences of the 7-year trajectories $\Delta V_{7\,years}^{L-E}$ in all five areas ($V^{L} - V^{E}$ in Table 1) are within $2\sigma$ (40 m a$^{-1}$, $< 2\%$ of their velocities). Thus, the flow line curvature does not cause a significant E- and L-velocity difference, and the velocity requirement in *Premise I* is well met.

Moving to PIG (Fig. R1-7), the most dynamic glacier in Antarctica (velocity up to ~4,000 m a$^{-1}$), there exist a high level of temporal accelerations caused by basal melting and calving activities (Joughin et al., 2020, 2021), similar to what happened in *Area 4* of TG in *Experiment 1* (Li et al., 2015). In this case the threshold $k$ for the average E- and L-velocity difference is 5 ($> 2$). However, the computed OE corrections can still remove the spatial acceleration-induced OE portion and leave the temporal acceleration-induced OE portion as a signature for long-term ice dynamics studies. Hence, *Premise I* is not a necessary condition, but a sufficient condition.

We further performed an in-depth experiment in *Area 2* of PIG (in Fig. R1-7), which is located in the most curved section along the main trunk of PIG (Fig. R1-5). We tracked point *P* in 2013 to point *P'* in 2020 consecutively using 7 Landsat-8 images with a 1-year interval, resulting in 7 trajectory segments. The straight distance *PP'* is 19,720 m and the curved distance is 21,161 m. Accordingly, the E-and L-velocities of the 7-year span are 2,817 m a$^{-1}$ and 3,023 m a$^{-1}$, respectively. That means that the curvature of the 7-year trajectory (a deviation $b$ of 1,305 m from the straight line) created a difference $\Delta V_{7\,years}^{L-E}$ of 206 m a$^{-1}$ at the trajectory end, among which 195 m a$^{-1}$ (95%) was corrected by our method (Table R1-2).

[Figure]

Figure R1-5. Illustration of difference between E- and L-velocities along a 7-year trajectory from $P$ to $P'$ in the most curved section in PIG. The background is the one-year span velocity map of 2013 from ITS_LIVE.

**4.3 OE estimation in fast flowing glaciers**

We estimate OE corrections assuming that within a longer time span (e.g., over 5 - 10 years) in *Premise II* the acceleration trend would not change significantly during the time span, $\left|\overline{a}\left(V^{L(n)}\right) - \overline{a}\left(V^L\right)\right| \leq k' \frac{\sigma}{\Delta t_n}$. As shown in *Experiments 1*, this trend change is under $1\frac{\sigma}{\Delta t_n}$ (the acceleration equivalent of velocity mapping uncertainty $\sigma$) for the Totten Glacier, one of the fast-flowing glaciers in East Antarctica. Since the velocity requirements in *Premise I* and acceleration requirement in *Premise II* were met properly, we were able to correct in average 88% of the OEs.

The extremely high ice dynamics exists mostly in fast flowing glaciers in West Antarctica or Greenland due to impact of climate warming. Here we evaluate the influence of such high dynamics on OE corrections in PIG. The acceleration trend differences $\left|\overline{a}\left(V^{L(n)}\right) - \overline{a}\left(V^L\right)\right|$ in all 5 areas (Table R1-2) are in average less than $3\frac{\sigma}{\Delta t_n}$. Correspondingly, the black and blue lines of all 5 areas appear parallel (Panels 1c-5c in Fig. R1-8), indicating that the acceleration condition in *Premise II* is properly met. Consequently, the proposed method corrected in average 97 m a⁻¹ (~40%) of the total OE (245 m a⁻¹), leaving the residuals (60%) in the adjusted velocities. The residuals represent the velocity change signature over the time span caused by the continuous basal melting and drastic calving activities in and after 2017 in PIG (*Experiment 3*).

We performed an experiment for Jakobshavn Isbrae (JI), Greenland. We used a baseline velocity map of 2013 (one year span) from ITS_LIVE, based on which we calculated L-velocities of 2-, 3-, and 10-year time spans along the centerline of the main trunk. Subsequently, we estimate the L-accelerations and OEs (Fig. R1-6).

[Figure]

Figure R1-6. (a) Velocity, grounding line, and profile position along the main trunk of Jakobshavn Isbrae; (b) E-velocity of 1-year span along the profile from ITS_LIVE, and L-velocities of 2-, 3-, and 10-year spans calculated from the 1-year span E-velocity map; (c) L-accelerations of the corresponding time spans along the profile, and (d) Estimated OEs caused by the L-velocities of 3 time spans.

Jakobshavn Isbrae is about ~60 km from the inland interior to the marine terminus (Figs. R1-6a), over which the ice mass picks up velocity from ~1000 m a$^{-1}$ to over ~15000 m a$^{-1}$. The maximum velocity is ~3 times higher, while the main trunk is ~4 times shorter than in PIG. Consequently, the ice shelf part can only be covered by the 1-year span E-velocity map (Fig. R1-6b); similarly, only 16 km long inland interior along the profile is covered by the 10-year span L-velocity. This makes it difficult to map velocities in the grounding zone and ice shelf region using images of over 1-year span, i.e., lost opportunities for historical velocity

recovery. The estimated OEs of three spans reached ~1,500 m a$^{-1}$ (up to 38% of the 10-year span L-velocity, Figs. R1-6b and d). This is significantly higher than 19% in PIG (Table R1-2).

In comparison to the reported velocity changes of ~125 m a$^{-1}$ from 2001-2008 in TG (Li et al., 2015) and ~500 m a$^{-1}$ from 2018-2020 in PIG (Joughin et al., 2021), the estimated average OE corrections of up to ~20 m a$^{-1}$ in TG and DG (Tables 1 and 2), and ~97 m a$^{-1}$ in PIG (Table R1-2) are not significant with respect to the natural variability of the glacier flow. In addition, the proposed method is applied for longer span velocities (a few years to over 5-10 years), seasonal variations should be averaged out. Therefore, the applicability of this method should not be affected by seasonal velocity changes and natural variability of glacier flow in Antarctica.

The estimated OEs in Jakobshavn Isbrae reached ~1,500 m a$^{-1}$ (up to 38% of the 10-year span L-velocity, Figs. R1-6b and d). The reported seasonal change can go as high as ~5,000 m a$^{-1}$ (50%) in last decade (Joughin et al., 2020). We believe that more comprehensive studies are needed in applicability of our method in Jakobshavn Isbrae and other fast flowing glaciers in Greenland.

Li, X., E. Rignot., M. Morlighem., J. Mouginot., & B. Scheuchl. (2015). Grounding line retreat of Totten Glacier, East Antarctica, 1996 to 2013. Geophysical Research Letters, 42(19), 8049-407. http://doi.org/10.1002/2015GL065701.

Joughin, I., Shean, D. E., Smith, B. E., & Floricioiu, D. (2020). A decade of variability on Jakobshavn Isbræ: ocean temperatures pace speed through influence on mélange rigidity. The Cryosphere, 14(1), 211-227.

Joughin, I., D. Shapero, B. Smith, P. Dutrieux, M. Barham (2021). Ice-shelf retreat drives recent Pine Island Glacier speedup. Sci. Adv. 7, eabg3080.

**Finally, while the authors are discussing the large overestimation error on Pine Island glacier (36%), they are presenting a first application of the method on Totten glacier. Hence, I think that it would increase the paper's logic and readability to keep this example for the application part (Totten could be put in the supplementary material). With such a high overestimation, I expect the results to be spectacular.**

Response:

As suggested, we carried out an experiment in PIG, *Experiment 3*. We present the detailed results in the following.

Experiment 3. Velocity overestimation correction in the Pine Island Glacier, West Antarctica

Five areas along the centerline and in the margin in PIG (Fig. R1-7) are selected to show the applicability of the OE correction method. Limited by the fast velocity, lost image features over long time spans, and availability of images, the maximum span is 7 years and most of the ice shelf cannot be covered. Since TG has a complete coverage of 7-year span and more

systematic results, its results in *Experiment 1* are presented in the main text of the paper. And the PIG results in *Experiment 3* will be presented in Appendix.

[Figure]

Figure R1-7. Application of the OE correction method in PIG. Five areas are selected to map E-velocities and calculate L-velocities. The OEs are estimated and adjusted. Their effect on grounding line flus estimation is also analyzed. The background is the 1-year velocity map of PIG 2003 from ITS_LIVE.

We mapped five areas with E-velocities of 3-month (Panels 1a-5a in Fig. R1-8) and 7-year (Panels 1b-5b in Fig. R1-8) spans and used them for calculation of L-velocities (blue and black lines, Panels 1c-5c in Fig. R1-8). The multi-span E-velocities (3 months, 2, 3, … 7 years) are mapped by using image tracking to form the red lines. Overall, in *Premise II* the average acceleration trend difference is 8 m a$^{-2}$, less than $3\frac{\sigma}{\Delta t_n}$ (Avg. $\Delta t_n$=6 years, σ=20 m a$^{-1}$); the black lines and blue lines are approximately parallel (Panels 1c-5c in Fig. R1-8). Since there is a high level of basal melting and calving activities (Li et al., 2015; Joughin et al., 2020, 2021) in PIG over the 7 year period, the temporal acceleration-induced OE portion contributed to the average E- and L-velocity difference along the trajectories in *Premise I*, less than 5σ (4% of the average velocity). However, the computed OE corrections can still remove the spatial acceleration-induced OE portion and leave the temporal acceleration-induced OE portion as a signature for long-term ice dynamics studies. Hence, Premise I is not a necessary condition, but a sufficient condition.

[Figure]

Figure R1-8. Velocities in five areas of PIG and OE corrections. (a) Panels 1a–5a show the reconstructed 3-month velocity maps $V_{3\ months}$ in the areas; matched points (red triangles) are used to map E-velocity $V^E_{2013-i}$ ($i=2014, …, 2020$); points along the flow line (blue dots) are tracked from the 3-month maps and used to calculate L-velocity $V^L_{2013-i}$; they are presented as the red and blue lines in Panels 1c–5c. (b) Similarly, Panels 1b–5b illustrate the reconstructed 3, 6 and 7-year velocity maps $V_{2013-n}$ in the areas with the matched points (red triangles) for E-velocity $V^{E(n)}_{2013-i}$; points along the flow line (blue dots) are tracked from the n-year velocity map and used to calculate L-velocity $V^{L(n)}_{2013-i}$, which are presented as the black

lines in Panels 1c–5c. (c) Panels 1c–5c show E-velocity $V_{2013-i}^{E}$ (red line), L-velocity $V_{2013-i}^{L}$ from 3-month E-velocity map (blue line), and L-velocity $V_{2013-i}^{L(n)}$ from n-year E-velocity map (black line) in each area.

The OE in *Area 5* of the low flowing interior margin is 18 m a$^{-1}$, under $1\sigma$ (Fig. R1-7, Table R1-2). From *Area 1* to *Area 4* along the main trunk the OEs are higher, ranging from 119 m a$^{-1}$ (3%) to 626 m a$^{-1}$ (19%). Correspondingly, there is an increase in actual E-velocities over the entire span (2013-2020), making the E-velocities (red lines) mostly above the L-velocities from the 3-month span map (blue line) (Panels 1c-5c in Fig. R1-8). This temporal acceleration correlates to long-term basal melting (Joughin et al., 2021). More specifically, the OE in *Area 2* close to grounding line reached the maximum of 626 m a$^{-1}$, with majority of the increase occurred in and after 2017 (Panel 2c in Fig. R1-8), which may be attributed to the drastic calving activities in and after 2017 as reported in Joughin et al. (2021). Overall, the average OE of five areas in PIG is 245 m a$^{-1}$ (11% of the average velocity 2319 m a$^{-1}$, Table R1-2), among which 97 m a$^{-1}$ (40%) of the spatial acceleration induced portion is effectively corrected, leaving 148 m a$^{-1}$ (60%) of the uncorrected and temporal acceleration induced portion. The latter portion is preserved in residuals as a significant signature of "historical" changes caused by the climate warming.

Table R1-2. Application of the OE correction method in PIG. "*Actual E-velocity and OE*" includes E-velocities of 3-month and n-year spans and their differences as OEs. "*Overestimation correction*" presents n-year span L-velocities from n-year span map, OE corrections, corrected E-velocities, and residuals (or errors) after correction.

| Area ID (span) | Actual E-velocity and OE | | | Overestimation correction | | | |
|---|---|---|---|---|---|---|---|
| | $V_{3\ months}^{E}$ (m a$^{-1}$) | $V_{n}^{E}$ (m a$^{-1}$) | $OE_{Actual}$ (m a$^{-1}$) | $V_{n}^{L(n)}$ (m a$^{-1}$) | Corr. (m a$^{-1}$) | $V_{Corr.}^{E}$ (m a$^{-1}$) | $\varepsilon$ (m a$^{-1}$) |
| **1** (6) | 3935 | 4054 | 119 | 4090 | -36 | **4018** | 83 |
| **2** (7) | 3278 | 3904 | 626 | 4099 | -195 | **3709** | 431 |
| **3** (7) | 2341 | 2532 | 191 | 2599 | -67 | **2465** | 124 |
| **4** (7) | 1651 | 1922 | 271 | 2103 | -181 | **1741** | 90 |
| **5** (3) | 392 | 410 | 18 | 416 | -6 | **404** | 12 |
| **MEAN** | 2319 | 2564 | 245 | 2661 | -97 | **2467** | 148 |
| **STD** | 1387 | 1506 | 232 | 1538 | 86 | **1477** | 163 |

Finally, based on an annual E-velocity map of 2013 in PIG from ITS_LIVE, L-velocities along GL with time spans of 1-15 years and the associated GL ice flux were computed. The results show that the OEs of the 15-year span can reach up to ~1300 m a$^{-1}$ in the GL region. These OEs in velocity can further cause an overestimation in GL flux, which rapidly increases by ~6.3 Gt a$^{-1}$ within a 4-year span; thereafter it slows down until the 15-year span, resulting in a total flux OE of 11.5 Gt a$^{-1}$. Consequently, the results indicate that a velocity map of a

time span within 3 years would be "flux OE-free" ($< \sigma_{Flux}$) in PIG. We used $\sigma_{Flux}$=5.8 Gt a$^{-1}$ that is the flux uncertainty in PIG reported by Rignot et al. (2019). Overall, the influence of the OEs on the GL flux appears not very significant, with the OE of a 15-year span map (11.5 Gt a$^{-1}$) less than 2 $\sigma_{Flux}$ (11.6 Gt a$^{-1}$). The details of this GL flux analysis is presented in responses to Referee #2's comments.

**Comments**

**L45. This is a citation for the Landsat-8 program. Not appropriate here.**
Response:
Agreed. We replaced Wulder et al. 2019 with Chander et al. (2009) that is more relevant to the historical Landsat programs (Landsat MSS, TM, etc.). It is also added in references.

Chander, G., Markham, B. L., & Helder, D. L. (2009). Summary of current radiometric calibration coefficients for Landsat MSS, TM, ETM+, and EO-1 ALI sensors. Remote sensing of environment, 113(5), 893-903.

**L46. "3 to 15 years", is not accurate. Bindschadler and Scambos., 1991 used a cross-correlation algorithm on two images separated by roughly 1 year. Similarly, Bindschadler et al., 1996 also uses 1 year image-pairs (see Table 1 of their paper). Wulder et al does not contain ice velocity maps prior to the 1990s.**
Response:
We revised the text to "*... ... have been used to create regional velocity maps with a time span ranging from 1 to 23 years (Bindschadler & Scambos, 1991; Bindschadler et al., 1996; Wang et al., 2016; Cheng et al., 2019; Rignot et al., 2019)*". Here we deleted Wulder et al. 2019.
The time spans for the maps in the cited papers are: ~ 1 year in Bindschadler & Scambos (1991), 1 to 7 years in Bindschadler et al. (1996), 2 to 23 years in Wang et al. (2016), 1 to 15 years in Cheng et al. (2019), and 1 to 15 years in Rignot et al. (2019).

**L46. Can you define after which time span the overestimation is significant? (2 yr, 3yr ?). I found the use of images acquired more than 2 years apart quite rare, or limited to few points (large rifts for example).**
Response:
In Discussion we added: "*……Our experiment results show $\Delta t_{Ref}$ as 3.2 years, 3.0 months, and 1.4 months for TG, PIG, and Jakobshavn Isbrae (JI), Greenland, respectively. Thus, the estimated OE-free time spans appear to be related to ice flow dynamics of the glaciers. In Experiments 1 and 3, we used 1 year for TG and 3 months for PIG. We suggest that a linear*

*regression for $\Delta t_{Ref}$ estimation be performed before extensive historical velocity mapping would be carried out.*" The detailed reasoning is given in the responses to General Comments (4.1).

**L49-51. From these lines it is a bit difficult to understand the overestimation issue. Please, extend a bit this description with more details, and split the sentence in two or three parts.**

Response:

We explained it in a more mathematical or physics way. "*For example, at time1 a feature, with an initial velocity $v_0$ at the first location, is taken in the first image. The same feature is tracked in the second image taken at time2 after traveling at the velocity $v_0$ and an acceleration a for a time span of $\Delta t$ (time2-time1). Thus, the velocity $v=v_0+a\Delta t$ increases along with the time span $\Delta t$ if acceleration a exists. Given a constant acceleration, the velocity can be overestimated if the time span is long. Or the velocity overestimation is proportional to the time span.*"

**L52. Greene et al., 2020b; the reference list just says (Personal communication, comments on a manuscript), which I found a bit weak for a reference of a concept that is the base of this paper.**

Response:

We understand the concern. Chad Greene is now Referee #2 who volunteered to make the referee information open. We added Greene's Figures in Appendix as Figure A1 and quoted his text.

[Figure]

Figure R1-9. Velocity map of the Totten Glacier from ITS_LIVE (left) used to explain the concept of velocity overestimation caused by acceleration (right): "*Over 16 years, that parcel of ice travels about 13 km downstream (red path). It begins at a velocity of about 720 m/yr, and in the first 8 months it travels at an average rate very close to 720 m/yr. But then the ice picks up speed as it moves downstream, so in the first 10 years it does not travel just 7200*

*m—it actually travels about 7900 m, or an average speed of 790 m/yr……*" (Greene, 2020b).

**L53. The overestimation calculation over Pine Island Glacier is derived later in the manuscript, hence remove this part of the sentence.**

Response:

The "Pine Island" part of the sentence is deleted.

**L54. I would like to see a complete comparison of the simple method from Berthier et al., 2003, with the approach proposed here in the discussion section.**

Response:

Here we first have an analytical proof and then an experiment of TG in *Experiment 1*.

Assume that a tracked feature flows from A to B over a period of n years, with the middle point of AB denoted as M (arriving in m years); $v_0$ is the initial velocity at A; $a$ is acceleration that is constant both spatially and temporally (not a requirement in this paper). The following figure is a simplified situation (straight flow line, otherwise Lagrangian path and L-velocity have to be used).

[Figure]

Figure R1-10. Analytical description of the "midpoint" method for OE correction.

The one year (or shorter) "true" velocity at A (year 0 - 1) is $V_{0-1} = \frac{D_{0-1}}{1\ year}$; the one year (or shorter) "true" velocity at M (year (m-1) – m) is $V_{(m-1)-m)} = \frac{D_{(m-1)-m}}{1\ year}$; the overestimated velocity at A using the n year segment AB (year 0 - n) is $V_{0-n} = \frac{D_{0-n}}{n\ year}$. The math or physics problem is

$$V_{(m-1)-m)} = \frac{D_{(m-1)-m}}{1\ year} \ =? \ V_{0-n} = \frac{D_{0-n}}{n\ year} \tag{R1-1}$$

We further calculate the one year "true" velocity at M:

$$V_{(m-1)-m} = \frac{D_{(m-1)-m}}{1\ year} = \frac{V_{(m-1)} \cdot 1\ year + \frac{1}{2} \cdot a \cdot 1\ year^2}{1\ year} = V_{(m-1)} + \frac{1}{2} \cdot a \cdot 1\ year = V_0 + a \cdot$$

$$(m-1)\ years + \frac{1}{2} \cdot a \cdot 1\ year = V_0 + a \cdot \left(m - \frac{1}{2}\right)\ years.$$

On the other hand, the overestimated velocity is

$$V_{0-n} = \frac{D_{0-n}}{n\,years} = \frac{V_0 \cdot n\,years + \frac{1}{2} \cdot a \cdot (n\,years)^2}{n\,years} = V_0 + \frac{1}{2} \cdot a \cdot n\,years$$

Therefore, if Equation (R1-1) holds we must have

$$\left(m - \frac{1}{2}\right) = \frac{n}{2};\, m = \frac{n+1}{2}. \tag{R1-2}$$

However, given that it is a simplified uniformly accelerated motion, the tracked feature arrives at the halfway (M) in m years, which must be greater than half time $\frac{n}{2}$, namely

$$m > \frac{n}{2}. \tag{R1-3}$$

Therefore, we have

$$V_{(m-1)-m} = V_0 + a \cdot \left(m - \frac{1}{2}\right) years > V_0 + a \cdot \left(\frac{n}{2} - \frac{1}{2}\right) years = V_{0-n} - a\frac{1}{2}year.$$

We further have

$$V_{(m-1)-m} > V_{0-n} - a\frac{1}{2}year. \tag{R1-4}$$

If a = 0, $V_{(m-1)-m}$ and $V_{0-n}$ are the same. Otherwise, they may be different, depending on acceleration "a". Thus, for glaciers with low to median range acceleration (weak spatial gradient), the overestimation may have been corrected to a good percentage by "assigning velocities at middle points of segments". But if a≠0, the velocity at midpoint M is different from $V_{0-n}$.

In addition, in *Experiment 1* we estimated and corrected OEs in five areas of TG. Since *Area 4* was affected by a calving event during the time span, here we use other four areas (*Areas 1, 2, 3,* and *5*) to compare our OE correction method with the midpoint method. We estimated the E-velocities of 7 years (2013-2020) $V_{2013-2020}^{E}$ and assigned them to midpoints of the trajectories in the four areas (Table R1-3). They were then compared to the true velocity $V_{2013-2014}^{E}$ (one year span E-velocity) at the midpoints to calculate the bias. Similarly, the same overestimated 7-year span E-velocities $V_{2013-2020}^{E}$ were corrected using the method of this paper and assigned to the start points of the trajectories as *V_corrected* (Table R1-3). They were then compared to the true velocity $V_{2013-2014}^{E}$ also, but at the start points to calculate another set of bias. As shown in Table R1-3, the overall accuracy (Mean±RMSE) of the proposed method is 4±10 m a$^{-1}$ and that of the midpoint method is 12±14 m a$^{-1}$. Therefore, the proposed OE correction method should provide a more accurate velocity map than the "midpoint" method. We will add a statement in Discussion.

Table R1-3. Comparison of the proposed OE correction method with the "midpoint" method by Berthier et al. (2003)

| Area | Velocity assigned to the midpoint | 1-year map at the midpoint | Bias | Velocity assigned to the start point | 1-year map at the start point | Bias |
|------|------|------|------|------|------|------|
| | $V_{2013-2020}^{E}$ (m a$^{-1}$) | $V_{2013-2014}^{E}$ (m a$^{-1}$) | $\varepsilon_M$ (m a$^{-1}$) | $V_{corrected}$ (m a$^{-1}$) | $V_{2013-2014}^{E}$ (m a$^{-1}$) | $\varepsilon_S$ (m a$^{-1}$) |
| **1** | 843 | 860 | 17 | 824 | 824 | 0 |
| **2** | 1008 | 1007 | -1 | 999 | 1007 | 8 |
| **3** | 1317 | 1332 | 15 | 1297 | 1280 | -17 |
| **5** | 789 | 807 | 18 | 759 | 754 | -5 |
| **MEAN** | **989** | **1001** | **12** | **1069** | **1056** | **-4** |
| **RMSE** | **238** | **236** | **14** | **304** | **286** | **10** |

**L108. Please add reference to Figure 1a,b,c to help the reader's understanding of the whole concept.**

Response:

Wherever appropriate, we added references to these figures in the text for more clarity. "*As the time span increases at a fixed rate of 1 year, the traversed straight-line distance $D_{0-i}$ (red lines in Figure 1b), correspondingly E-velocity $V_{0-i}^{E}$, increases rapidly because of the acceleration over the traverse (Fig. 1c). In principle, every $V_{0-i}^{E}$ (i=1, 2, ...n) value represents the velocity at the same point $P_o(x_o, y_o)$ (Figure 1a) in these n velocity maps. In the cases where Image$_i$ were not available and thus the maps $V_{0-i}$ (i=1, ... n-1) were not produced, we only had the map $V_{0-n}$ with the longest span of n years. It is obvious that at $P_o(x_o, y_o)$ its n-year velocity $V_{0-n}^{E}$ is significantly larger than the 1-year velocity $V_{0-1}^{E}$ (Figure 1c).*

**L110. Here and in the remaining of the manuscript you use 1 year ice velocity as a reference map. But does your method still apply for very fast glaciers? For example Jakobshavn Isbrae (Greenland), or Penguin gl. (Patagonia) are flowing at speeds that are exceeding 10 km/yr, hence there is good chances of acceleration along flowlines within that year. Can you please discuss this point here? And better specify the use of a 1 year ice velocity map as a reference for your method.**

Response:

We agree with you that the time span of the reference (or baseline) velocity map may be different for different glaciers. We changed the sentences: "*Here we use a velocity map of a 1-year span as a baseline throughout the paper for simplicity, which can be changed for glacier regions of different ice flow dynamics as demonstrated in Experiment 3 (3 months for PIG). We require that the overestimation of the baseline map is negligible, or smaller than $\sigma$ (velocity mapping uncertainty, 20 m a$^{-1}$).*"

As explained in the response to the general comments (above), we also added a section in Discussion to introduce an analytical method for determining the threshold of an "OE-free" time span: "*The choice of a time span for defining an "overestimation-free" map in validation of Premises I and II can be performed in a more systematic way. An area of the highest acceleration in a glacier should be selected. Within the area the multi-span E-velocity series $V_{0-i}^E$ (i=1, 2, ... n) can be used to establish a linear relationship between the E-velocity and time span, $V^E = K \Delta t + b$, by a linear regression (red line in Fig. 2). Given an achievable velocity mapping uncertainty σ, the time span $\Delta t_{Ref}$ which would induce an OE that is smaller than σ and considered negligible can be calculated as $\Delta t_{Ref} = \frac{\sigma - b}{K}$. For example, Area 5 in Experiment 1 has the highest acceleration in TG (Table 1) and σ is 20 m a$^{-1}$. After a regression using the E-velocities in Area 5 (Fig. 5c) the "overestimation-free" time span is calculated as less than 3.2 years ($R^2$=0.96). Thus, the selected $\Delta t_{Ref}$ of one year in Experiment I is justified. Alternatively, if the multi-span E-velocities are not available, a velocity profile along the main trunk of a glacier may be established from an available short span E-velocity map. Along the profile the highest acceleration is located where a multi-span L-velocity series (blue line in Fig. 2) can be computed. Using this L-velocity series and the above regression method, the "OE-free" time span $\Delta t_{Ref}$ can also be estimated. Subsequently, we computed a series of multi-span L-velocities from 1- to 10-years and estimated an "OE-free" time span, 0.25 years (~3.0 months, $R^2$=0.99 and σ=20 m a$^{-1}$) for PIG and 0.11 years (~1.3 months, $R^2$=0.99 and σ=20 m a$^{-1}$) for Jakobshvn Isbrae (JI), Greenland, respectively. Therefore, in Experiment 3 we used the 3-month E-velocity as the "OE-free" $\Delta t_{Ref}$ for validation of Premises I and II in PIG (Appendix). In summary the estimated $\Delta t_{Ref}$ decreases from 3.2 years for TG, 3.0 months for PIG, and 1.3 months for JI. In general, the threshold of an "OE-free" time span for a specific glacier appears to depend on ice flow dynamics, along with others factors such as bed topography and availability of images. We suggest that a $\Delta t_{Ref}$ analysis using multi-span L-velocities be performed before an extensive historical velocity mapping would be carried out.*"

**L114-115. Does it depends on the speed of the glacier? i.e this assumption still hold for Jakobshavn Isbrae flowing at more than 15 km/yr ? Or Penguin gl. In Patagonia (12 km/yr) ?**

Response:

Yes, it does depend on speed/acceleration of the glaciers. Our new experimental results proved that also. In the earlier part of the paper we added: "*Here we use a velocity map of a 1-year span as a baseline throughout the paper for simplicity, which can be changed for glacier regions of different ice flow dynamics as demonstrated in Experiment 3 (3 months for PIG).*

*We require that the overestimation of the baseline map is negligible, or smaller than* $\sigma$ *(velocity mapping uncertainty, 20 m $a^{-1}$).*"

In Discussion we added: "...... *Our experiment results show* $\Delta t_{Ref}$ *as 3.2 years, 3.0 months, and 1.4 months for TG, PIG, and Jakobshavn Isbrae (JI), Greenland, respectively. Thus, the estimated OE-free time spans appear to be related to ice flow dynamics of the glaciers. In Experiments 1 and 3, we used 1 year for TG and 3 months for PIG. We suggest that a linear regression for* $\Delta t_{Ref}$ *estimation be performed before extensive historical velocity mapping would be carried out.*"

**Section 2.2. I am getting lost with the notation, between the U, S, Map, V.... What do you mean by Maps? Maps of ice velocity I guess, than why introducing Maps if you have later V? Why not just using V, and add E and L for Eulerian and Lagrangian as indice (VE and VL).**
Response:
We accepted your suggestion and used $V$ for map, $V^E$ for Eulerian and $V^L$ for Lagrangian velocity throughout the manuscript.

**Figure 2. Please give a more comprehensive caption of Figure2. This one is just not enough to understand what is there. What the difference between the two Lagrangian lines mean? See earlier comment on the writing of equation to simplify the text and improve the understanding the paper. I guess you have Map₀₋₁ to specify that the Lagrangian is only calculated with Map₀₋₁ ? This should be specified in the caption.**
Response:
The caption of Figure 2 is extended according to your suggestion. *"Figure 2: Derivation of equation for overestimation correction using L-velocity. Eulerian velocities $V_{0-i}^{E}$ (i=1, 2, ... n) are represented as bars. The red line is the average Eulerian velocity $\overline{V}_{0-i}^{E}$ of $V_{0-1}^{E}$ and $V_{0-n}^{E}$. The blue line is the average Lagrangian velocity $\overline{V}_{0-i}^{L}$ of $V_{0-1}^{L}$ and $V_{0-n}^{L}$ derived from $V_{0-1}^{E}$. The black line is the average Lagrangian velocity $\overline{V}_{0-i}^{L(n)}$ of $V_{0-1}^{L(n)}$ and $V_{0-n}^{L(n)}$ derived from $V_{0-n}^{E}$."*

**L125-155. I think that the choice of hyperscript and subscript in equations could be simplified for the seek of the reader's understanding. First, ice velocity maps are defined as Map₀₋ᵢ, V is used for Eulerian ice velocity and U is used for Lagrangian ice velocity. All of these are referring to ice velocities, so I suggest you switch to V for the velocity maps, V$_{E(0-i)}$ for the Eulerian speeds and V$_{L(0-i)}$ for the Lagrangian ones. You could also do V$^L_{0-I}$ and V$^E_{0-I}$, since I don't think that the use of J at line 240 is necessary for understanding (you could just say in the text that you calculate the overestimation of all sub-images).**

Response:

We accepted your suggestion and used *V* for map, $V^E_{0-i}$ for Eulerian and $V^L_{0-i}$ for Lagrangian velocity throughout the manuscript.

**L 133. What do you define as a "short" lagrangian trajectory? This should depend on the glacier speed (see earlier comments on fast flowing glaciers), hence the distance where this premise holds decreases when the glacier speed increases (which is in part linked to the local bedrock slope). Furthermore, this premise holds if you assume that the point moved on a straight line within this short time span.**

**What do you define as short time span? If I assume this is 1 year, this premise might be true for some ice shelves, but what happens if the flow changes direction? This might happen within 1 year for example for George VI, Abbott, Dotson/Crosson or the Getz ice shelves in Antarctica.**

Response:

We added a section in Discussion to introduce a linear regression method for determine the threshold of an "OE-free" time span $\Delta t_{Ref}$ (see responses to "General Comments" above). The result shows that $\Delta t_{Ref}$ is ~3.2 years ($R^2$=0.96) for TG. Thus, $\Delta t_{Ref}$=1 year would be appropriate for a large number of glaciers in Antarctica, including TG. However, a reduced time span of ~3.0 months ($R^2$=0.99) for PIG and ~1.4 months ($R^2$=0.86) for Jakobshan Isbrae in Greenland should be used. As shown in the proposed regression equation, this threshold changes with how "fast" the ice flows in a glacier.

*We changed the text for Premise I to: "**Premise I:** Within a period of n short time spans (e.g., 1 year or shorter) each segment (from $P_{i-1}$ to $P_i$ in Figure 1b) along a flow line is relatively short and straight, so that their accumulated curved L-distance $S_{0-i}$ over a longer span (e.g., 5-10 years) is not significantly different from the corresponding straight E-distance $D_{0-i}$; furthermore, their averaged velocity trends of $V^L_{0-i}$ in the Lagrangian framework and $V^E_{0-i}$ in the Eulerian framework (blue and red lines in Fig. 2) do not deviate significantly from each other, and their maximum difference is limited within a threshold ($V^L_{0-n} - V^E_{0-n} \leq k\ \sigma$), where k is a constant and $\sigma$ is the velocity mapping uncertainty."*

**L134-136. Here the use of the i=1,2,…,n is confusing. You are describing the case of a short time span, hence why not just using the $V_{0-1}$ and $U_{0-1}$ (as you just said in the previous lines)? Or $V_{i-i+1}$ ? What is a short time span on Figure 2 ? All of this Premise holds in what you define as a "limited time span" and "short L trajectory" (which should be straight). Please clarify these points.**

Response:

Agreed. We now use "*i to i+1*". Thus, we changed *Premise I*. See text above.

**The author choose to make a clear distinction between the theory vs the application,**

**which I think was a good idea, but here, it would help the readers to have some more self-explanatory examples, as it is done in section 2.3.**

Response:

We revised the text of *Premise I*. Figures 1b and 2 are used in the text to explain the concept. That way we link the theory of *Premise I* with "application" in these two figures. Hopefully, this will help the readers to better understand.

**L 137. This could be reformulated, for the more clarity, to "In reality, the available historical images only allow us to produce eulerian velocity maps with a long timespan, i.e $Map_{0-n}$ which leads to the maximum overestimation value as defined in equation 2"**

Response:

Accepted. The sentence is rewritten accordingly.

**L138. "As we can only use $Map_{0-n}$ , the lagrangian velocity, for a long time span, is defined as follow". Please also add a reference to the line in Figure 2.**

Response:

Yes, the sentence is changed to: "*As we can only use the map $V_{0-n}$, the L-velocity (black line in Figure 2), for a long time span, is defined as follow:*"

We added the text to the black line in Figure 2: "*Average L-velocity $\overline{V}_{0-i}^{L(n)}$ from $V_{0-n}^{E}$*".

**L 141. "Consequently, the 1 year L-velocity $U'_{0-1}$…"**

Response:

Thanks. It is so changed.

**L145. What do you define again as a limited time span? If you compare $Map_{0-1}$ and $Map_{0-n}$, then you are comparing the smallest and largest time span, hence the use of the term "short timespan" is a bit confusing**

Response:

It is changed to "*Within the time span of n years ……*"

**L145. I guess that the magnitude of the velocity $Map_{0-n}$ should be larger than Map0-1 , but the pattern is similar ? Can you provide a figure example with velocity direction to illustrate this point?**

Response:

We agree that the text along the three lines in Figure 2 is a bit confusing. The blue is the average L-velocity calculated from map $V_{0-1}$, and red line is the average E-velocity calculated from 1 year and n year E-velocities. Generally, the L-velocities are greater than E-velocities (see blue and red examples in Figure 5). We revised text in Figure 2 as follows:

"*Average L-velocity $\overline{V}_{0-i}^{L(n)}$ from $V_{0-n}^{E}$ (for black line)*

*Average L-velocity $\overline{V}_{0-i}^{L}$ from $V_{0-1}^{E}$ (for blue line)*

*Average E-velocity $\overline{V}_{0-i}^{E}$ from $V_{0-1}^{E}$ and $V_{0-n}^{E}$ (for red line)"*

**L148. The first part of the sentence can be removed since it has been described earlier, before Premise II. Then, you can just start with: "Hence, based on Premise II, we have…."**

Response:

Thanks. The text is revised accordingly.

**L150. Again, the $U_{0\text{-}n}=V_{0\text{-}n}$ is based on the fact that you are considering only short timespan. But is that the case if you use "0-n" ? (see earlier comment)**

Response:

Whether $V_{0-n}^{L} = V_{0-n}^{E}$ (or $U_{0\text{-}n}=V_{0\text{-}n}$) in *Premise I* does depend on the time span, given acceleration and geometric complexity of a glacier. We added two subsections in Discussion to show our new experimental results.

**L 151. I would reformulate this sentence to remind the reader about the aim of this paper : "Consequently, using the map with the longest timespan, we can go back to $V_{0\text{-}1}$ using a Correction term defined as Correction=$V_{0\text{-}n}$-U'$_{0\text{-}n}$".**

Response:

Thanks. The sentence is revised accordingly: "*Consequently, using the map with the longest time span, we can go back to $V_{0-1}^{E}$ using a correction term defined as Correction=$V_{0-n}^{E} - V_{0-n}^{L(n)}$.*"

**L 164. What does it mean to interpolate the positions to the sub grid-level ? Does it make any sense to interpolate the position at a higher level of resolution than the velocity field?**

Response:

The interpolation is not used to make a new higher resolution velocity map, but to determine the positions of the distance segments for L-distance integration. Thus, the intermediated sub-grid positions are used for a continuous distance integration. The sentence is changed to: "*The sub-grid positions of the monthly segments are interpolated for integration of the overall L-distance.*"

**Figure 3. Please add a general Figure of the entire Pine Island glacier, to check out where the location of your flowline is (similar as Figure 4).**

Response:

It is done. Thanks.

**L179-180. What about orthorectification errors in historical images ?**

Response:

We added it to the sentence: "*Despite the subpixel accuracy of the orthorectification of historical images and ……*"

**Acceleration computation: this has already been described L147. I would suggest to move this part earlier (or remove it).**

Response:

This section is now moved to the earlier part.

**Section 3.1. Since section 2.3 shows an example over Pine Island glacier, I don't know why the author didn't continue using this example. Since the overestimation is quite spectacular, I would strongly suggest to use Pine Island instead of Totten here.**

Response:

Yes, we accepted your suggestion and added the PIG results in *Experiment 3* (see responses to General Comments).

Limited by the fast velocity, lost image features over long time spans, and availability of images, the maximum span is 7 years and most of the ice shelf cannot be covered. Since TG has a complete coverage and more systematic results, its results in *Experiment 1* are presented in the main text of the paper. The PIG results in *Experiment 3* will be presented in Appendix.

**L213. Why did you choose a 7 year trajectory ?**

Response:

We found the earliest available high-quality Landsat 8 images in 2013 and latest ones in 2020 (7 years). We changed the sentence: "*To avoid lower quality historical velocity maps that may influence the effectiveness of the validation, we use the earliest available high-quality Landsat 8 images from 2013 to 2020 to produce velocity maps $V_{2013-i}$ (i=2014, ... 2020)……*"

**L217. See previous comment on the choice of symbols in equations.**

Response:

They are all fixed throughout the manuscript (see responses to previous comments)

**L215. Do you generate the map separately or over the entire glacier directly ?**

Response:

Because we have to generate maps for 7 time spans for validating *Premises I* and *II*, it is a lot of work to produce them all. Therefore, we only mapped areas of 5 trajectories, instead of the entire glacier directly.

**L225. I am surprised about the error estimation here. Millan et al., 2019; had some smaller number for 1 year map of ice velocity using Landsat-8. Can you discuss why is that? How does your map compare with available NSIDC data ? What is the difference with recent map assembled from sar interferometry? (see Mouginot et al., 2019)**

Response:

The highest accuracy of less than 1 m/year was achieved by using **InSAR** technique

(Mouginot et al., 2019), which used the InSAR phase information and the data requirement is generally high. An accuracy of 10 m/year was reported by Millan et al. (2019) for average annual velocities from **multiple individual velocities** derived from Sentinel-2 (10 m resolution) and Landsat-8 (15 m resolution) and other images. Similarly, Gardner et al. (2018) also achieved an accuracy of 10 m/year of annual velocities by averaging velocities derived from **multiple Lansat-7 and -8 image pairs** (15 m resolution). However, our velocity sub-maps (5 areas) were built from **only one Landsat-8 image pair** (15 m resolution) for each map. Given the accuracy of individual maps as $\sigma^2_i$, the accuracy of the averaged velocity is $= \frac{\sqrt{\Sigma \sigma_i^2}}{n}$. In general, the accuracy of the average annual velocity should be smaller than that of the velocity of an individual pair. Therefore, our accuracy of 20 m/year for 1-year (individual pair) and 3 m/year for 7-year (individual pair) velocities are reasonable values.

References

Mouginot, J., Rignot, E., & Scheuchl, B.(2019). Continent wide, interferometricSAR phase, mapping of Antarctic icevelocity. Geophysical Research Letters,46, 9710–9718. https://doi.org/10.1029/2019GL083826

Millan, R., Mouginot, J., Rabatel, A., Jeong, S., Cusicanqui, D., Derkacheva, A., & Chekki, M. (2019). Mapping surface flow velocity of glaciers at regional scale using a multiple sensors approach. Remote Sensing, 11(21). https://doi.org/10.3390/rs11212498

Gardner, A. S., Moholdt, G., Scambos, T., Fahnstock, M., Ligtenberg, S., van den Broeke, M., and Nilsson, J.: Increased West Antarctic and unchanged East Antarctic ice discharge over the last 7 years, The Cryosphere, 12, 521–547, https://doi.org/10.5194/tc-12-521-2018, 2018.

**L239. Is it "faster than" or "close" ?**
Response:
We deleted "close to".

**L240. I think that the use of "J" in exponent is adding to much complication (see earlier comment). You can just specify that you do the calculation for all sub-images.**
Response:
We accepted your suggestion and removed "J" throughout the manuscript.

**L257. The "apparent parellelity" is quite subjective I think. Is the premise II validated for case 2c and 2a?**
Response:
Cases 2c and 2a are ok (average difference of 1.2 m a$^{-2}$). Cases 3c and 3a are 2.1 m a$^{-2}$ (max) that is still within the allowable uncertainty of 3 m a$^{-2}$ determined based on the velocity mapping uncertainty. To make it objective, we changed "*apparent parellelity*" to "*relatively well-maintained parellelity*".

**Figure 5. Add the direction of the flow in the sub-images. You could also consider using different symbols for the 7 yr and 1 yr trajectory and use a color gradient for the position of the points that changes with the year. This "year" color could then also be used in the scatter plots.**

Response:

We revised Fig. 5 accordingly.

[Figure]

Figure 5.

**Table 1. Please add more details on the caption of the Table, ie, the content of each column.**

Response:

The caption is revised: "*Table 1. Velocity and acceleration in Eulerian and Lagrangian frameworks used for validation of the overestimation correction method. "Actual E-velocity and OE" lists actually mapped 1-year and 7-year E-velocities and their differences as overestimations in all five areas. "Premise I" contains 7-year L-velocities computed from the 1-year velocity map and corresponding L- and E-velocity differences, which are used for validating Premise I. "Premise II" illustrates averaged L-accelerations computed from the 7-year and 1-year velocity maps, respectively, as well as their differences, which are used for validating Premise II. "Overestimation correction" presents 7-year L-velocities computed from the 7-year map, overestimation corrections, and E-velocities and residuals (or errors) after correction.*"

**L302. The acronym OE has been defined before.**

Response:
We deleted "(OE)".

**Section 4 Discussion. Can you discuss the performance of your method, with the relatively simple approach defined by Berthier et al., 2003? I think that the section is missing some discussion on the applicability of the method to 1) fast glacier, 2) the glacier geometry, which can be much more complex than the glaciers that were used here to validate the method (not straight) (see earlier comment). An additional discussion about the significance of the correction, with respect to the seasonal variation in ice flow velocity of the glacier should also be discuss, if the method is expected to be applicable in Greenland. Specifically, does the magnitude of the correction could exceed the natural variability of the glacier? I guess that the amount of acceleration with a flowline would need to be significant in order to induce a correction that would exceed the variability of the seasonal signal?**

Response:

The comparison results with the midpoint method are presented in the response to comment L54.

The responses related to fast glaciers, complex geometry, seasonal variability, applicability in Greenland etc. are presented in responses to General Comments.

**Figure 5-6. Can you provide a figure of the corrected ice velocity? Maybe a difference map.**

Response:

We produced the map of DG (Fig. 6) after correction (Fig. R1-11b). It does not appear distinctly different from the map before correction (Fig. R1-11a). Thus, we will not include this corrected map in the main text. The map with the OEs (Fig. 6c) is actually the difference map. Similarly, the corrected submaps in PIG are also visually not distinct from those before correction. You may agree that we would not add the corrected or difference map to Figs. 5 and 6.

[Figure]

Figure R1-11. E-velocity in TG: (a) before OE correction, and (b) after OE correction.

---

## Author Comment (AC2)

We appreciate the constructive comments from two reviewers and the community. Our manuscript will be much improved by their input. We have made changes to our manuscript. In the following responses, we use "**bold**" text for comments, "non-bold" text for our responses, and "*italic*" for changed text in the manuscript.

**Referee #2 (Chad A. Greene):**

**This paper identifies three key shortcomings of a common velocity measurement technique, and provides a solution that addresses all three. At issue are 1. the true location of a feature tracked velocity measurement, 2. the acceleration that a parcel of ice may experience between image acquisition times, and 3. the fact that ice does not always move in a perfectly straight line between image acquisition times. The problem and solution are described well in this paper, and the authors demonstrate that they have a good handle on the data and how velocities are interpreted from a glaciological standpoint.**

**This work will be of value to the community, both to raise awareness of the overestimation issue, and to provide a solution to it. I recommend publication, with just a few suggestions that may help readers understand the impact of overestimation and how it should affect our interpretation of previous studies.**

**Main Issues**

**The paper does a good job of describing the problem and solution from a technical standpoint, and anyone who has written feature-tracking algorithms will benefit from reading the paper. However, there are many readers who don't write their own algorithms, but will nonetheless want to understand how overestimation might affect their scientific results. Some work could be done in this paper to better communicate the overall impact of how overestimation impacts long-term studies.**

**Here's a type of analysis that I would find much more insightful than the stats for PIG, Totten, and David Gl that are currently presented in the abstract: I would like to see a figure showing Eulerian grounding line flux calculations as a function of dt, where dt might range from a day to 20 years. This would provide readers with some intuition for a threshold value of dt, beyond which Eulerian measurements produce significantly different estimates of ice flux. It's possible that the percentage reduction in GL flux as a function of dt might vary regionally, and that diversity could be interesting to show as well.**

Response:

We performed an experiment of "GL flux vs. time span" for PIG. We used a baseline velocity map of PIG 2013 (Fig. R2-1a) from ITS_LIVE (Gardner et al., 2019). The flux gate (red line)

is set along the grounding line (GL, black line) and separated into flux nodes every 240 m where the ice flow velocity and ice thickness data are used to calculate ice flux. We calculated L-velocities along GL with time spans of 1-15 years based on the 2013 E-velocity map (Fig. R2-1b). Instead of suggested 20 years, we used 15-year time span mainly because the 20-year tracking distance from GL would run beyond the ice shelf front.

[Figure]

Figure R2-1: (a) Annual ice velocity map of PIG (2013) from ITS_LIVE (Gardner et al., 2019) with the flux gate (red line) set on GL (black line), (b) L-velocities of 1- to 15-year span $V^L_{0-i}$ ($i=1, 2, ... 15$) along GL calculated from the 2003 annual map, (c) GL ice flux (estimated from $V^L_{0-i}$) vs. time span, and (d) L-velocities $\bar{V}^L_{0-i}$ (averaged over the GL portion across main trunk, over 1500 m a$^{-1}$) vs. time span.

The L-velocity along GL increases mainly in the margin areas of the main trunk as the time span increases (Fig. R2-1b). The maximum velocity OE reached ~1300 m a$^{-1}$ for the 15-year span. The annual maximum OE rate started at ~461 m a$^{-1}$ and decreases to ~31 m a$^{-1}$ as the time span increases, because the later part of the trajectory reached the flat part of the ice shelf front (high velocity, but low acceleration).

Consequently, the OE induced GL flux increase (flux overestimation) speeds up quickly at an annual rate of ~2.1 Gt a$^{-1}$ for the first 4 annual spans, by ~6.3 Gt a$^{-1}$ from ~116.7 Gt a$^{-1}$ to ~123.0

Gt a$^{-1}$. Thereafter the annual rate is maintained at 0.4 Gt a$^{-1}$ until the 15-year span, reaching the maximum flux OE of 11.5 Gt a$^{-1}$. This flux OE has the same trend pattern as the average L-velocity over the GL portion across main trunk (Fig. R2-1d), indicating that the main flux OE "came" across GL from the main trunk portion.

Due to the limited time and amount of work related to the responses to all three sets of comments we will investigate the same issue in other diverse types of glacier regions in the future.

**In addition to a figure showing how dt affects GL flux in Eulerian measurements, I'd like some clear guidance in the abstract for when the Eulerian approximation is sufficient or insufficient.**

The above simulation result shows that a GL flux OE of ~11.5 Gt a$^{-1}$ in PIG would be induced by a 15-year span L-velocity map, which is significant in comparison to the flux uncertainty $\sigma_{Flux}$ of 5.8 Gt a$^{-1}$ in PIG given by Rignot et al. (2019). Therefore, assuming $\sigma_{Flux}$ = 5.8 Gt a$^{-1}$, we estimated a "flux-OE-free" time span of ~3 years using the curve in Fig. R2-1c.

At the end of *Experiment 3* we added: "*In addition, based on a 2013 E-velocity map in PIG from ITS_LIVE, L-velocities along GL with time spans of 1-15 years and the associated GL ice flux were computed. The results show that the OE of the 15-year span can reach up to ~1300 m a$^{-1}$ in the GL region. This OE in velocity further caused an overestimation in GL flux, which rapidly increases by ~6.3 Gt a$^{-1}$ within the 4-year span; thereafter it slows down until the 15-year span, resulting in a total flux OE of ~11.5 Gt a$^{-1}$. Consequently, the results indicate that a velocity map of a time span within 3 years would be "flux OE-free", inducing a flux OE less than a threshold $\sigma_{Flux}$. We used $\sigma_{Flux}$=5.8 Gt a$^{-1}$ that is the flux uncertainty in PIG reported by Rignot et al. (2019). Overall, the influence of the OEs on the GL flux appears not very significant, with the OE of a 15-year span map less than 2 $\sigma_{Flux}$.*"

We added a statement in Abstract: "*……Our experiment results in PIG with a 15-year time span showed that the flux overestimation caused by the OE in velocity increases rapidly within the first 4-year span before it slows down and reaches the maximum of ~11.5 Gt a$^{-1}$; the flux OE is negligible within a time span of 3-years……*"

**In the abstract and/or discussion, I suggest flipping the logic/wording around at least once to make it clear that the overestimation of historical velocities could mean that previous papers have underestimated the magnitude of glacier acceleration over the past few decades. It's only a minor change in wording, but I think it's an important take-home message of this paper that should be stated directly.**

Response:

We added a statement in the discussion section: "*…… The implication is that, when using newer*

*velocity maps of short spans along with historical maps of long spans in previous studies for estimation of glacier acceleration over the past few decades, the overestimation of historical velocities could have caused an underestimation in the acceleration magnitude. On the other hand, new efforts in historical velocity mapping at an ice sheet – wide or large regional scale should be made with a full consideration of the OE correction.*"

**Minor comments**

**Abstract: The case studies of PIG, Totten, and David Glacier provide decent testing grounds for the methods presented in this paper, but the details of these studies feel somewhat anecdotal and very specific to the exact images that were used in these particular cases. I recommend generalizing the results in the abstract to give readers a better overview of the problem. Only after discussing the overall impact of the overestimation, then it may be helpful to mention a specific case of PIG, Totten, or David to as a tangible example.**

Response:

Accepted. We revised Abstract accordingly: "*…… In comparison to velocity maps derived from recent satellite images of monthly to weekly time spans, historical maps, from before the 1990s, generally cover longer time spans, e.g., over 10 years, due to the scarce spatial and temporal coverage of earlier satellite image data. We found velocity overestimations (OEs) in such long-term maps that can be mainly attributed to ice flow acceleration, time span of the images used, and glaciers with complex geometry. If used for long-term change studies, these OEs in historical velocities may further affect the estimated trends of ice flow dynamics and mass balance. For example, the OEs can reach from ~69 m $a^{-1}$ (7-year span) in Totten Glacier, East Antarctica, up to ~930 m $a^{-1}$ (10-year span) in Pine Island, West Antarctica……*"

**L29: This line mentions "the input-output method" and some good references are provided for it, but some readers may be unfamiliar with the term. If the term is necessary for some point that's being made, then I recommend briefly describing what is meant by "the input-output method" here. If the term is not important for this paper, then consider removing it.**

Response:

The phrase "using the input–output method" is removed.

**L69: Recommend changing "It is proven that…" to "We show that…" to make it clear that the correction is original work that is presented in this paper.**

Response:

We changed the sentence accordingly.

**L80: I'm not entirely sure what "descending passages" means. Consider rewording.**

Response:

The sentence is revised: "*We describe an acceleration-induced overestimation using a typical scenario in AIS (Fig. 1a) where ice flow accelerates over a long slope from several glaciers originated from the inland interior, running through the main trunk, and discharging to the ocean.*"

**L160: "At each grid…" I think this should be "At each grid cell…" or "At each pixel…"**
Response:
We changed it to "*At each grid cell ……*"

**Figure 5 is very compelling, and I want to make sure I understand it. Unfortunately, the labels and caption are somewhat cryptic, so I'm not sure if I'm even getting the main message right. The caption contains a list of the data labels that are mostly redundant with labels that are presented directly in the figure. What's missing is physical interpretation or any direct take home message. For example, the variables U, U', and V are labeled in the figure and in the caption, but there's no physical definition of what U, U', or V mean. Help readers by providing a sentence or two in the caption that directly states the main point and any secondary point(s) that may be worth noticing. The main point, I assume, is that the black line is consistently higher than the red and blue curves. State that in the caption, in terms of what it means physically. What causes the red and blue curves to cluster together or spread apart from each other? Mention the underlying mechanism in the caption. Most of these points are described in detail on page 11, but most readers will appreciate having the main points stated directly in the figure caption.**
Response:

The caption is revised according to the suggestions: "*Figure 5: Velocities in five areas (rectangles in Fig. 4) of the Totten Glacier are used to validate Premises I and II. (a) Panels 1a–5a show the reconstructed 1-year velocity maps $V_{2013\text{-}2014}$ in the areas; matched points (red triangles) are used to map E-velocity $V_{2013-i}^{E}$ (i=2014, …, 2020) (red lines in Panels 1c-5c); points along the flow line (blue dots) are tracked from the 1-year maps and used to calculate L-velocity $V_{2013-i}^{L}$ (blue lines in Panels 1c-5c). (b) Similarly, Panels 1b–5b illustrate the reconstructed 7-year velocity maps $V_{2013\text{-}2020}$ in the areas with the matched points (red triangles) for E-velocity $V_{2013-i}^{E(7)}$; points along the flow line (black crosses) are tracked from the 7-year span velocity map and used to calculate L-velocity $V_{2013-i}^{L(7)}$ (black lines in Panels 1c-5c). (c) In each area (Panels 1c–5c) the difference between red (straight) and blue (curved) lines increases with time, but is limited within $k\sigma$ at the end (Premise I), except a large k in Area 4 because of effect of a calving event; the black line is above blue line because of the spatial acceleration-induced OE over 7-year span; but they are relatively parallel(Premise II) and thus, a correction can be estimated.*"

---

## Author Response (AR1)

We appreciate the constructive comments from two reviewers and the community. Our manuscript will be much improved by their input. We have made changes to our manuscript. In the following responses, we use "**bold**" text for comments, "non-bold" text for our responses, and "*italic*" for changed text in the manuscript.

**Referee #1**

**General comments**

**The paper by Li et al., 2021 presents an innovative method that aims at correcting glacier velocity overestimation, that are due to accelerations, when using long timespan. The paper is well presented with a clear structure, well written and the Figures are relatively clear.**

**A large share of the paper is dedicated to the description of the method, which is simple in principle, but that could be ambiguous to understand clearly. Consequently, I have few comments that I hope, will help to make the paper more understandable.**

**Among those comments, the definition of the different Premises needs a bit of clarification, and particularly on their area of validity (see below)......**

Response:
In *Premises I and II* now we clarified the correspondence between the short span (months to year) and "OE-free" ($< \sigma$) trajectory segments, as well as that between the longer span (5 to over 10 years) and difference of E- and L-velocities ($< k\,\sigma$) at the end of the entire trajectory.

(Line 152 in the marked-up manuscript) *"**Premise I:** Within a baseline time span (e.g., 1 year or shorter) each segment (from $P_{i-1}$ to $P_i$ in Figure 1b) is relatively short and the E- and L-velocity difference is smaller than $\sigma$. Furthermore, over the map span of n years (e.g., 5-10 years or longer) the accumulated E- and L-distances along the entire trajectory do not deviate significantly from each other, so that the maximum velocity difference in the Lagrangian and Eulerian frameworks (end points of blue and red lines in Fig. 2) is limited within a threshold ($V_{0-n}^L - V_{0-n}^E \leq k\,\sigma$), where k is a constant and $\sigma$ is the velocity mapping uncertainty."*

(Line 169) *"**Premise II:** Within the time span of n years (e.g., over 5-10 years), the velocity field described by $V_{0-1}^E$ and $V_{0-n}^E$ does not change significantly, so that the line between $V_{0-n}^{L(n)}$ and $V_{0-1}^{L(n)}$ (black line in Fig. 2) and that between $V_{0-n}^L$ and $V_{0-1}^L$ (blue line in Fig. 2) are approximately parallel to each other. Accordingly, the difference between their simple averaged accelerations is within a threshold ($\left|\bar{a}(V^{L(n)}) - \bar{a}(V^L)\right| \leq k'\,\frac{\sigma}{\Delta t_n}$), where $k'$ is a constant and $\sigma$ is the velocity mapping uncertainty and*

$$\overline{a}\left(V^{L(n)}\right) = \frac{V_{0-n}^{L(n)} - V_{0-1}^{L(n)}}{\Delta t_n}, \ \ \overline{a}(V^L) = \frac{V_{0-n}^L - V_{0-1}^L}{\Delta t_n}.$$ *(7)"*

Their validity in different types of glaciers is introduced in the Discussion section. Based on the premises we rephrased sentences of OE corrections to formalize a Theorem:

(Line 190) "***Overestimation Correction Theorem***: *Assume that the necessary condition in Premise II is met, spatial acceleration - induced overestimations in long time span velocities $V_{0-n}^E$ can be corrected or reduced using the following Correction term, regardless of temporal acceleration:*

$$V_{0-1}^E = V_{0-n}^E + Correction + \varepsilon, \tag{9}$$

$$Correction = V_{0-n}^E - V_{0-n}^{L(n)}. \tag{10}$$

*If Premise I holds (sufficient conditions are met), Correction $\approx -OE_{0-n}$; otherwise, $|Correction| < |OE_{0-n}|$, preserving the velocity increases induced by temporal accelerations in the residual term $\varepsilon$ (Fig. A2).*"

**...... The discussion also needs to be supplemented with an overview of the method applicability to different glacier types, and specifically fast flowing glaciers (e.g Jakobshavn Isbrae), or with a more complex geometry (e.g Zachariae Isstrøm ice shelf, Getz Ice Shelf or George VI ice shelf). Another interesting point of discussion is the impact of the glacier seasonal variability. Are the corrections significant with respect to the natural variability of glacier flow? While, seasonal signals are not really pronounced in Antarctica, variability can be much greater in Greenland (cf. Joughine et al., 2020).**

**For example, using a 1-year velocity reference for a glacier like Jakobshavn Isbrae, might not be ideal, as the glacier is flowing at more than 15 km/yr (which increases the chances of acceleration along a flowline). Similarly does the premises still holds, for glaciers that are changing directions and not flowing in a straight line (for example the ice shelf of Zachariae Isttrøm before 2000)?**

Response:
(Line 417) The Discussion section is restructured. At the beginning we added an overview statement: "*In this section we discuss the applicability of the proposed method in terms of overestimation-free time span, influence of complex glacier geometry, overestimation in fast flowing glaciers, and comparison with the "Midpoint" method.*"

Then we added three new subsections and restructured one section to make a strong Discussion according to the comments. The four subsection titles are:

(Line 419) "***4.1 Threshold of the overestimation-free time span for trajectory segments***"

(Line 430) "***4.2 Glaciers with complex geometry***"

(Line 443) "***OE correction in fast-flowing glaciers***"

(Line 484) "***Comparison with the "Midpoint" method***"

Please see manuscript for details.
* * *
In the following we present supporting materials for the above Discussion that are not given in the manuscript.

4.1 Threshold of the overestimation-free time span for trajectory segments

The choice of a short time, i.e., baseline or reference, span $\Delta t_{Ref}$ (e.g., a few months to a year) for the "overestimation-free" segments along a trajectory in *Premises I* makes sure that the difference between E-and L-velocities within the span is negligible, or less than σ (velocity mapping uncertainty, σ=20 m a$^{-1}$ in this study). It is also the time span of the initial "OE-free" E-velocity map that is used for ice mass tracking and L-velocity computation in premise validation. Determination of this threshold has an implication on validation of *Premise I*, as well as the integration period of the trajectory segments from $P_i$ to $P_{i+1}$ (Fig. 3b). Estimation of $\Delta t_{Ref}$ can be performed in a systematic way. An area of the highest acceleration in a glacier should be selected. Within the area a multi-span E-velocity series $V_{0-i}^E$ (*i=1, 2, ... n*) can be used to establish a linear relationship between the E-velocity $V^E$ and time span $\Delta t$, $V^E = K \Delta t + b$, by a linear regression (red line in Fig. 2). With the known parameters of *b* and *K*, $\Delta t_{Ref}$ can be calculated as $\Delta t_{Ref} = \frac{\sigma - b}{K}$. For example, *Area 5* in TG in *Experiment 1* has the highest acceleration (Table 1, Fig. 5c). After a regression using the E-velocities in *Area 5*, $\Delta t_{Ref}$ is calculated as 3.2 years ($R^2$=0.96, Fig. R1-1a). Thus, the selected $\Delta t_{Ref}$ of one year for TG in *Experiment 1* is justified.

Alternatively, if the E-velocities are not available, multi-span L-velocities along a profile on the main trunk of a glacier may be established from an available short span E-velocity map (e.g., 1-year map in PIG, Fig. R1-2a). Along the profile the highest acceleration (location "A" in Fig. R1-2a) is localized where a multi-span L-velocity series can be computed (Fig. R1-2b). Using this L-velocity series and the above regression method, the "OE-free" time span $\Delta t_{Ref}$ can also be estimated. For example, given σ=20 m a$^{-1}$ and a series of computed multi-span L-velocities from 1- to 10-years at location "A" in PIG (Fig. R1-2) and "B" in Jakobshavn Isbrae (JI, Fig. R1-6), we estimated $\Delta t_{Ref}$ as ~3.0 months ($R^2$=0.99, Fig. R1-1b) and ~1.4 months ($R^2$=0.86, Fig. R1-1c) for PIG and JI, respectively.

Based on the above analysis results of $\Delta t_{Ref}$ in TG (3.2 years), PIG (3.0 months) and JI (1.4 months), it appears that the threshold of an "OE-free" time span is strongly related to the ice flow dynamics of the glaciers. Given a known σ, shorter spans of $\Delta t_{Ref}$ should be selected for trajectory segments in validation of *Premise I* and L-velocity integration in faster flow

glaciers. We suggest that an analysis of multi-span L-velocities and a regression for $\Delta t_{Ref}$ be performed before extensive historical velocity mapping would be carried out.

[Figure]

Figure R1-1. Linear regression of multi-span velocities vs. time span is performed to estimate the "OE-free" time span. (a) Totten Glacier (TG): 7-years of E-velocities in *Area 5* (Fig. 4), $\Delta t_{Ref}$=3.2 years; (b) Pine Island Glacier (PIG): 10-years of L-velocities at location "A" near grounding line (Fig. R1-2), $\Delta t_{Ref}$=3.0 months; and (c) Jakobshavn Isbrae (JI): 10-years of L-velocities at location "B" along the main trunk profile (Fig. R1-6), $\Delta t_{Ref}$=1.4 months.

[Figure]

Figure R1-2. (a) Velocity, grounding line, and profile position along the main trunk of PIG; (b) E-velocity of 1-year span along the profile from ITS_LIVE, and L-velocities of 2-, 3-, and 10-year spans calculated from the 1-year span E-velocity map; (c) L-acceleration of the corresponding time spans along the profile, and (d) Estimated OEs caused by the L-velocities of different time spans. Please note that the L-velocities are only used for simulation estimation of time span $\Delta t_{Ref}$. In validation of *Premises I* and *II*, we used actual measurements of multi-year span E-velocities.

**4.2 Glaciers with complex geometry**

Furthermore, within a longer time span (e.g., over 5 - 10 years) in *Premise I* the difference between the E- and L-velocities accumulated over all segments along the entire trajectory, i.e. the end-point deviation between the red and blue lines in Fig. 2, is measured with a more tolerable threshold of $k$ times of $\sigma$ ($k$ $\sigma$). Although the OEs of the trajectory segments are controlled by $\Delta t_{Ref}$, ice mass moving along a curved flow line over this long span may result in an additional discrepancy along the entire trajectory.

As suggested, we examined the complex geometry issue in 5 Antarctic glaciers, including George VI, Abbott, Dotson, Crosson and Getz (Fig. R1-3a) where the velocity ranges from ~100 m a$^{-1}$ to 1,000 m a$^{-1}$. We used one-year velocity maps of 2013 of 5 glaciers from ITS_LIVE to derive 20-year span L-velocities along trajectories in the significantly curved sections of the glaciers (Fig. R1-4). The computed straight and curved lengths vs. time span in

the 5 glaciers are illustrated in Fig. R1-3b. The statistics of the corresponding E- and L-velocities are given in Table R1-1.

[Figure]

Figure R1-3. (a) Locations of 5 glaciers with complex geometry, including George VI, Abbott, Dotson, Crosson, and Getz. (b) Computed straight (displacement) and curved (trajectory) lengths vs. time span in 5 glaciers.

[Figure]

Figure R1-4. (a-e) locations of trajectories in 5 glaciers with complex geometry, including

George VI, Abbott, Dotson, Crosson, and Getz. (f-j) 20-year span trajectories in 5 glaciers.

It is shown in Table R1-1 that in these 5 glaciers three glaciers (Abbott, Dotson and Getz) have very small OEs in the 20-year span L-velocities ($< \sigma = 20$ m a$^{-1}$), and their "curvature" induced differences $\Delta V_{20\,y}^{L-E}$ are 0 m a$^{-1}$. The other two glaciers (George VI and Crosson) have OEs of 114 m a$^{-1}$ and 143 m a$^{-1}$, respectively. However, the "curvature"-induced differences are only 11 m a$^{-1}$ and 6 m a$^{-1}$, both of which are smaller than $\sigma$ and thus, negligible. So, the OEs are mainly caused by spatial acceleration here, not the curvature.

Table R1-1. Overestimations and "curvature" induced velocity differences in 5 glaciers

| ID | Name | $V_{1\,y}^{E}$ (m a$^{-1}$) | $V_{20\,y}^{L}$ (m a$^{-1}$) | OE (m a$^{-1}$) | $\Delta t$ (year) | E-dist. (m) | L-dist. (m) | $\Delta V_{20\,y}^{L-E}$ (m a$^{-1}$) |
|----|------|------|------|------|------|------|------|------|
| 1 | George VI | 507 | 621 | -114 | 20 | 12412 | 12633 | 11 |
| 2 | Abbott | 110 | 115 | -6 | 20 | 2306 | 2308 | 0 |
| 3 | Crosson | 1054 | 1197 | -143 | 20 | 23942 | 24064 | 6 |
| 4 | Dotson | 280 | 290 | -10 | 20 | 5803 | 5809 | 0 |
| 5 | Getz | 315 | 324 | -9 | 20 | 6481 | 6482 | 0 |

In the Totten Glacier flow lines are less curved and velocity is higher (up to ~1,400 m a$^{-1}$). The E- and L-velocity differences of the 7-year trajectories in all five areas ($V^L - V^E$ in Table 1) are within $2\sigma$ (40 m a$^{-1}$, $< 2\%$ of their velocities). Thus, the flow line curvature does not cause a significant E- and L-velocity difference, and the conditions in *Premise I* are well met.

We further performed an in-depth experiment in *Area 2* of PIG (in Fig. 3a), which is located in the most curved section along the main trunk of PIG (Fig. R1-5). We tracked positions of point *P* from 2013 (*P*) to 2020 (*P'*) consecutively using 7 Landsat-8 images with a 1-year interval, resulting in 7 trajectory segments. The straight distance *PP'* is 19,720 m and the curved distance is 21,161 m. Accordingly, the E-and L-velocities of the 7-year span are 2,817 m a$^{-1}$ and 3,023 m a$^{-1}$, respectively. That means that the curvature of the 7-year trajectory (a deviation *b* of 1,305 m from the straight line) created a difference of 206 m a$^{-1}$ at the trajectory end, among which 195 m a$^{-1}$ (95%) was corrected by our method (Table 3).

[Figure]

Figure R1-5. Illustration of difference between E- and L-velocities along a 7-year trajectory from *P* to *P'* in the most curved section in PIG. The background is the one-year span velocity map of 2013 from ITS_LIVE.

4.3 OE estimation in fast flowing glaciers

We estimate OE corrections assuming that within a longer time span (e.g., over 5 - 10 years) in *Premise II* the acceleration trend would not change significantly, $\left| \bar{a}(V^{L(n)}) - \bar{a}(V^L) \right| \leq k' \frac{\sigma}{\Delta t_n}$. As shown in *Experiments 1*, this trend change is under $1 \frac{\sigma}{\Delta t_n}$ (the acceleration equivalent of velocity mapping uncertainty σ) for the Totten Glacier, one of the fast-flowing glaciers in East Antarctica. Since the velocity requirements in *Premise I* and acceleration requirement in *Premise II* were met properly, we were able to correct in average 88% of the OEs.

The extremely high ice dynamics exists in fast flowing glaciers in West Antarctica or Greenland due to impact of climate warming. Here we evaluate the influence of such high dynamics on OE corrections in PIG. The acceleration trend differences $\left| \bar{a}(V^{L(n)}) - \bar{a}(V^L) \right|$ in all 5 areas (Table A4) are in average less than $3 \frac{\sigma}{\Delta t_n}$. Correspondingly, the black and blue lines of all 5 areas appear parallel (Panels 1c-5c in Fig. A3), indicating that the acceleration condition in *Premise II* is properly met. Consequently, the proposed method corrected in average 97 m a$^{-1}$ (~40%) of the total OE (245 m a$^{-1}$), leaving the residuals (60%) in the adjusted velocities. The residuals represent the velocity change signature over the time span caused by the continuous basal melting and drastic calving activities in and after 2017 in PIG (*Experiment 3*).

We performed an experiment for Jakobshavn Isbrae (JI), Greenland. We used a baseline velocity map of 2013 (one year span) from ITS_LIVE, based on which we calculated L-velocities of 2-, 3-, and 10-year time spans along the centerline of the main trunk. Subsequently, we estimate the L-accelerations and OEs (Fig. R1-6).

[Figure]

Figure R1-6. (a) Velocity, grounding line, and profile position along the main trunk of Jakobshavn Isbrae; (b) E-velocity of 1-year span along the profile from ITS_LIVE, and L-velocities of 2-, 3-, and 10-year spans calculated from the 1-year span E-velocity map; (c) L-accelerations of the corresponding time spans along the profile, and (d) Estimated OEs caused by the L-velocities of 3 time spans.

Jakobshavn Isbrae is about ~60 km from the inland interior to the marine terminus (Figs. R1-6a), over which the ice mass picks up velocity from ~1000 m a$^{-1}$ to over ~15000 m a$^{-1}$. The maximum velocity is ~3 times higher, while the main trunk is ~4 times shorter than in PIG. Consequently, the ice shelf part can only be covered by the 1-year span E-velocity map (Fig. R1-6b); similarly, only 16 km long inland interior along the profile is covered by the 10-year span L-velocity. This makes it difficult to map velocities in the grounding zone and ice shelf region using images of over 1-year span, i.e., lost opportunities for historical velocity

recovery. The estimated OEs of three spans reached ~1,500 m a$^{-1}$ (up to 38% of the 10-year span L-velocity, Figs. R1-6b and d). This is significantly higher than 19% in PIG (Table 3).

In comparison to the reported velocity changes of ~125 m a$^{-1}$ from 2001-2008 in TG (Li et al., 2015) and ~500 m a$^{-1}$ from 2018-2020 in PIG (Joughin et al., 2021), the estimated average OE corrections of up to ~20 m a$^{-1}$ in TG and DG (Tables 1 and 2), and ~97 m a$^{-1}$ in PIG (Table 3) are not significant with respect to the natural variability of the glacier flow. In addition, the proposed method is applied for longer span velocities (a few years to over 5-10 years), seasonal variations should be averaged out. Therefore, the applicability of this method should not be affected by seasonal velocity changes and natural variability of glacier flow in Antarctica.

The estimated OEs in Jakobshavn Isbrae reached ~1,500 m a$^{-1}$ (up to 38% of the 10-year span L-velocity, Figs. R1-6b and d). The reported seasonal change can go as high as ~5,000 m a$^{-1}$ (50%) in last decade (Joughin et al., 2020). We believe that more comprehensive studies are needed in applicability of our method in Jakobshavn Isbrae and other fast flowing glaciers in Greenland.

Li, X., E. Rignot., M. Morlighem., J. Mouginot., & B. Scheuchl. (2015). Grounding line retreat of Totten Glacier, East Antarctica, 1996 to 2013. Geophysical Research Letters, 42(19), 8049-407. http://doi.org/10.1002/2015GL065701.

Joughin, I., Shean, D. E., Smith, B. E., & Floricioiu, D. (2020). A decade of variability on Jakobshavn Isbræ: ocean temperatures pace speed through influence on mélange rigidity. The Cryosphere, 14(1), 211-227.

Joughin, I., D. Shapero, B. Smith, P. Dutrieux, M. Barham (2021). Ice-shelf retreat drives recent Pine Island Glacier speedup. Sci. Adv. 7, eabg3080.

**Finally, while the authors are discussing the large overestimation error on Pine Island glacier (36%), they are presenting a first application of the method on Totten glacier. Hence, I think that it would increase the paper's logic and readability to keep this example for the application part (Totten could be put in the supplementary material). With such a high overestimation, I expect the results to be spectacular.**

Response:

As suggested, we carried out an experiment in PIG, *Experiment 3*. The results are presented as a new section:

(Line 391) "*3.3 Experiment 3: Velocity overestimation correction at Pine Island Glacier, West Antarctica*"

**Comments**

**L45. This is a citation for the Landsat-8 program. Not appropriate here.**
Response:
(Line 53) Agreed. We replaced Wulder et al. 2019 with Chander et al. (2009) that is more relevant to the historical Landsat programs (Landsat MSS, TM, etc.). It is also added in references.

Chander, G., Markham, B. L., & Helder, D. L. (2009). Summary of current radiometric calibration coefficients for Landsat MSS, TM, ETM+, and EO-1 ALI sensors. Remote sensing of environment, 113(5), 893-903.

**L46. "3 to 15 years", is not accurate. Bindschadler and Scambos., 1991 used a cross-correlation algorithm on two images separated by roughly 1 year. Similarly, Bindschadler et al., 1996 also uses 1 year image-pairs (see Table 1 of their paper). Wulder et al does not contain ice velocity maps prior to the 1990s.**
Response:
(Line 54) We revised the text to "*... ... have been used to create regional velocity maps with a time span ranging from 1 to 23 years (Bindschadler & Scambos, 1991; Bindschadler et al., 1996; Wang et al., 2016; Cheng et al., 2019; Rignot et al., 2019)*". Here we deleted Wulder et al. 2019.
The time spans for the maps in the cited papers are: ~ 1 year in Bindschadler & Scambos (1991), 1 to 7 years in Bindschadler et al. (1996), 2 to 23 years in Wang et al. (2016), 1 to 15 years in Cheng et al. (2019), and 1 to 15 years in Rignot et al. (2019).

**L46. Can you define after which time span the overestimation is significant? (2 yr, 3yr ?). I found the use of images acquired more than 2 years apart quite rare, or limited to few points (large rifts for example).**
Response:
(Line 425) In Discussion we added: "*......Our experiment results show $\Delta t_{Ref}$ as 3.2 years, 3.0 months, and 1.4 months for TG, PIG, and Jakobshavn Isbrae (JI), Greenland, respectively. Thus, the estimated OE-free time spans appear to be related to ice flow dynamics of the glaciers. In Experiments 1 and 2, we used 1 year for TG and 3 months for PIG. We suggest that a linear regression for $\Delta t_{Ref}$ estimation be performed before extensive historical velocity mapping would be carried out.*" The detailed reasoning is given in the responses to General Comments (4.1).

**L49-51. From these lines it is a bit difficult to understand the overestimation issue. Please, extend a bit this description with more details, and split the sentence in two or**

**three parts.**

Response:

(Line 57) We explained it in a more mathematical or physics way. "*For example, at time1 a feature, with an initial velocity $v_0$ at the first location, is taken in the first image. The same feature is tracked in the second image taken at time2 after traveling at the velocity $v_0$ and an acceleration a for a time span of Δt (time2-time1). Thus, the velocity $v=v_0+a\Delta t$ increases along with the time span Δt if acceleration a exists. Given a constant acceleration, the velocity can be overestimated if the time span is long. Or the velocity overestimation is proportional to the time span.*"

**L52. Greene et al., 2020b; the reference list just says (Personal communication, comments on a manuscript), which I found a bit weak for a reference of a concept that is the base of this paper.**

Response:

(Line 691) We understand the concern. Chad Greene is now Referee #2 who volunteered to make the referee information open. We added Greene's Figures in Appendix as Figure A1 and quoted his text.

[Figure]

Figure A1. Velocity map of the Totten Glacier from ITS_LIVE (left) used to explain the concept of velocity overestimation caused by acceleration (right): "*Over 16 years, that parcel of ice travels about 13 km downstream (red path). It begins at a velocity of about 720 m/yr, and in the first 8 months it travels at an average rate very close to 720 m/yr. But then the ice picks up speed as it moves downstream, so in the first 10 years it does not travel just 7200 m—it actually travels about 7900 m, or an average speed of 790 m/yr……*" (Greene, 2020b).

**L53. The overestimation calculation over Pine Island Glacier is derived later in the manuscript, hence remove this part of the sentence.**

Response:

The "Pine Island" part of the sentence is deleted.

**L54. I would like to see a complete comparison of the simple method from Berthier et al., 2003, with the approach proposed here in the discussion section.**

Response:

(Line 64) Here we first present an analytical proof, and then we added a section of experiment results of TG in Discussion.

Assume that a tracked feature flows from A to B over a period of n years, with the middle point of AB denoted as M (arriving in m years); $v_0$ is the initial velocity at A; $a$ is acceleration that is constant both spatially and temporally (not a requirement in this paper). The following figure is a simplified situation (straight flow line, otherwise Lagrangian path and L-velocity have to be used).

[Figure]

Figure R1-10. Analytical description of the "midpoint" method for OE correction.

The one year (or shorter) "true" velocity at A (year 0 - 1) is $V_{0-1} = \frac{D_{0-1}}{1\ year}$; the one year (or shorter) "true" velocity at M (year (m-1) – m) is $V_{(m-1)-m)} = \frac{D_{(m-1)-m}}{1\ year}$; the overestimated velocity at A using the n year segment AB (year 0 - n) is $V_{0-n} = \frac{D_{0-n}}{n\ year}$. The math or physics problem is

$$V_{(m-1)-m)} = \frac{D_{(m-1)-m}}{1\ year} \ \textcolor{red}{=?}\ V_{0-n} = \frac{D_{0-n}}{n\ year} \tag{R1-1}$$

We further calculate the one year "true" velocity at M:

$$V_{(m-1)-m} = \frac{D_{(m-1)-m}}{1\ year} = \frac{V_{(m-1)} \cdot 1\ year + \frac{1}{2} \cdot a \cdot 1\ year^2}{1\ year} = V_{(m-1)} + \frac{1}{2} \cdot a \cdot 1\ year = V_0 + a \cdot$$

$(m-1)\ years + \frac{1}{2} \cdot a \cdot 1\ year = V_0 + a \cdot \left(m - \frac{1}{2}\right)\ years.$

On the other hand, the overestimated velocity is

$$V_{0-n} = \frac{D_{0-n}}{n\ years} = \frac{V_0 \cdot n\ years + \frac{1}{2} \cdot a \cdot (n\ years)^2}{n\ years} = V_0 + \frac{1}{2} \cdot a \cdot n\ years$$

Therefore, if Equation (R1-1) holds we must have

$$\left(m - \frac{1}{2}\right) = \frac{n}{2}; m = \frac{n+1}{2}. \tag{R1-2}$$

However, given that it is a simplified uniformly accelerated motion, the tracked feature arrives at the halfway (M) in m years, which must be greater than half time $\frac{n}{2}$, namely

$$m > \frac{n}{2}. \tag{R1-3}$$

Therefore, we have

$$V_{(m-1)-m} = V_0 + a \cdot \left(m - \frac{1}{2}\right) \, years > V_0 + a \cdot \left(\frac{n}{2} - \frac{1}{2}\right) \, years = V_{0-n} - a\frac{1}{2} \, year.$$

We further have

$$V_{(m-1)-m} > V_{0-n} - a\frac{1}{2} \, year. \tag{R1-4}$$

If a = 0, $V_{(m-1)-m}$ and $V_{0-n}$ are the same. Otherwise, they may be different, depending on acceleration "a". Thus, for glaciers with low to median range acceleration (weak spatial gradient), the overestimation may have been corrected to a good percentage by "assigning velocities at middle points of segments". But if a $\neq$ 0, the velocity at midpoint M is different from $V_{0-n}$.

We added a new section in Discussion:

(Line 484) "*4.4 Comparison with the "Midpoint" method*

*The "Midpoint" method presented in Berthier et al. (2003) compensates overestimations by assigning the overestimated velocities to middle points of the trajectories. We use the velocity measurements in Experiment 1 to compare the performances of these two OE correction methods. Since Area 4 was affected by a calving event during the time span, here we use other four areas (Areas 1, 2, 3, and 5, Fig. 4a). We estimated the E-velocities of 7 years (2013-2020) $V^E_{2013-2020}$ and assigned them to the midpoints of the trajectories in the four areas (Table A4). They were then compared to the one-year span velocity $V^E_{2013-2014}$ at the midpoints to calculate the bias $\varepsilon_M$. Similarly, the same overestimated 7-year span E-velocities $V^E_{2013-2020}$ were corrected using the OE correction method of this paper and assigned to the start points of the trajectories as $V_{corrected}$ (Table A4). They were then compared to the one-year span $V^E_{2013-2014}$ also, but at the start points to calculate another set of bias $\varepsilon_S$. As shown in Table A4, the proposed OE correction method achieved a higher overall accuracy of 4 $\pm$10 m a$^{-1}$, compared to 12 $\pm$14 m a$^{-1}$ of the "midpoint" method."*

**Table A4. Comparison of the proposed OE correction method with the "Midpoint" method in Berthier et al. (2003)**

| Area ID | OE velocity assigned to midpoint | 1-year map at midpoint | Bias | OE-corrected velocity assigned to start point | 1-year map at start point | Bias |
|---|---|---|---|---|---|---|
| | $V^E_{2013-2020}$ (m a$^{-1}$) | $V^E_{2013-2014}$ (m a$^{-1}$) | $\varepsilon_M$ (m a$^{-1}$) | $V_{corrected}$ (m a$^{-1}$) | $V^E_{2013-2014}$ (m a$^{-1}$) | $\varepsilon_S$ (m a$^{-1}$) |
| 1 | 843 | 860 | 17 | 824 | 824 | 0 |
| 2 | 1008 | 1007 | -1 | 999 | 1007 | 8 |
| 3 | 1317 | 1332 | 15 | 1297 | 1280 | -17 |
| 5 | 789 | 807 | 18 | 759 | 754 | -5 |
| MEAN | 989 | 1001 | 12 | 970 | 966 | -4 |

**L108. Please add reference to Figure 1a,b,c to help the reader's understanding of the whole concept.**

Response:

(Line 116) Wherever appropriate, we added references to these figures in the text for more clarity. "*As the time span increases at a fixed rate of 1 year, the traversed straight-line distance $D_{0-i}$ (red lines in Figure 1b), correspondingly E-velocity $V_{0-i}^{E}$, increases rapidly because of the acceleration over the traverse (Fig. 1c). In principle, every $V_{0-i}^{E}$ (i=1, 2, ...n) value represents the velocity at the same point $P_o(x_o, y_o)$ (Figure 1a) in these n velocity maps. In the cases where $Image_i$ were not available and thus the maps $V_{0-i}$ (i=1, ... n-1) were not produced, we only had the map $V_{0-n}$ with the longest span of n years. It is obvious that at $P_o(x_o, y_o)$ its n-year velocity $V_{0-n}^{E}$ is significantly larger than the 1-year velocity $V_{0-1}^{E}$ (Figure 1c).*"

**L110. Here and in the remaining of the manuscript you use 1 year ice velocity as a reference map. But does your method still apply for very fast glaciers? For example Jakobshavn Isbrae (Greenland), or Penguin gl. (Patagonia) are flowing at speeds that are exceeding 10 km/yr, hence there is good chances of acceleration along flowlines within that year. Can you please discuss this point here? And better specify the use of a 1 year ice velocity map as a reference for your method.**

Response:

(Line 123) We agree with you that the time span of the reference (or baseline) velocity map may be different for different glaciers. We changed the sentences: "*Here we use a velocity map of a 1-year span as a baseline ("overestimation free") throughout the paper for simplicity, which can be changed for glacier regions of different ice flow dynamics (spatial acceleration, mainly caused by bed topography and slopes). For example the baseline span is one year for TG in Experiment 1 and three months for PIG in Experiment 3. We require that the overestimation of the baseline map is negligible, or smaller than $\sigma$ (velocity mapping uncertainty).*"

(Line 419) We also added a section in Discussion to introduce an analytical method for determining the threshold of an "OE-free" time span: "*4.1 Threshold of the overestimation-free time span for trajectory segments*"

**L114-115. Does it depends on the speed of the glacier? i.e this assumption still hold for Jakobshavn Isbrae flowing at more than 15 km/yr ? Or Penguin gl. In Patagonia (12 km/yr) ?**

Response:

(Line 123) Yes, it does depend on speed/acceleration of the glaciers. Our new experimental results proved that also. In the earlier part of the paper we added: "*Here we use a velocity map of a 1-year span as a baseline ("overestimation free") throughout the paper for simplicity, which can be changed for glacier regions of different ice flow dynamics (spatial acceleration, mainly caused by bed topography and slopes). For example the baseline span is one year for TG in Experiment 1 and three months for PIG in Experiment 3. We require that the overestimation of the baseline map is negligible, or smaller than $\sigma$ (velocity mapping uncertainty).*"

(Line 425) In Discussion we added: "……*Our experiment results show $\Delta t_{Ref}$ as 3.2 years, 3.0 months, and 1.4 months for TG, PIG, and Jakobshavn Isbrae (JI), Greenland, respectively. Thus, the estimated OE-free time spans appear to be related to ice flow dynamics of the glaciers. In Experiments 1 and 2, we used 1 year for TG and 3 months for PIG, respectively. We suggest that a linear regression for $\Delta t_{Ref}$ estimation be performed before extensive historical velocity mapping would be carried out.*"

**Section 2.2. I am getting lost with the notation, between the U, S, Map, V…. What do you mean by Maps? Maps of ice velocity I guess, than why introducing Maps if you have later V? Why not just using V, and add E and L for Eulerian and Lagrangian as indice (VE and VL).**
Response:
We accepted your suggestion and used $V$ for map, $V^E$ for Eulerian and $V^L$ for Lagrangian velocity throughout the manuscript.

**Figure 2. Please give a more comprehensive caption of Figure2. This one is just not enough to understand what is there. What the difference between the two Lagrangian lines mean? See earlier comment on the writing of equation to simplify the text and improve the understanding the paper. I guess you have Map$_{0-1}$ to specify that the Lagrangian is only calculated with Map$_{0-1}$ ? This should be specified in the caption.**
Response:
(Line 136) The caption of Figure 2 is extended according to your suggestion. "*Figure 2: Derivation of equation for overestimation correction using L-velocity. Eulerian velocities $V^E_{0-i}$ (i=1, 2, … n) are represented as bars. The red line is the average Eulerian velocity $\overline{V}^E_{0-i}$ of $V^E_{0-1}$ and $V^E_{0-n}$ . The blue line is the average Lagrangian velocity $\overline{V}^L_{0-i}$ of $V^L_{0-1}$ and $V^L_{0-n}$ derived from $V^E_{0-1}$. The black line is the average Lagrangian velocity $\overline{V}^{L(n)}_{0-i}$ of $V^{L(n)}_{0-1}$ and $V^{L(n)}_{0-n}$ derived from $V^E_{0-n}$.*"

**L125-155. I think that the choice of hyperscript and subscript in equations could be simplified for the seek of the reader's understanding. First, ice velocity maps are defined as Map$_{0-i}$, V is used for Eulerian ice velocity and U is used for Lagrangian ice velocity. All of these are referring to ice velocities, so I suggest you switch to V for the velocity maps, V$_{E(0-i)}$ for the Eulerian speeds and V$_{L(0-i)}$ for the Lagrangian ones. You could also do V$^L_{0-I}$ and V$^E_{0-I}$, since I don't think that the use of J at line 240 is necessary for**

**understanding (you could just say in the text that you calculate the overestimation of all sub-images).**

Response:

We accepted your suggestion and used $V$ for map, $V^E_{0-i}$ for Eulerian and $V^L_{0-i}$ for Lagrangian velocity throughout the manuscript.

**L 133. What do you define as a "short" lagrangian trajectory? This should depend on the glacier speed (see earlier comments on fast flowing glaciers), hence the distance where this premise holds decreases when the glacier speed increases (which is in part linked to the local bedrock slope). Furthermore, this premise holds if you assume that the point moved on a straight line within this short time span.**

**What do you define as short time span? If I assume this is 1 year, this premise might be true for some ice shelves, but what happens if the flow changes direction? This might happen within 1 year for example for George VI, Abbott, Dotson/Crosson or the Getz ice shelves in Antarctica.**

Response:

(Line 152) We added a section in Discussion to introduce a linear regression method for determine the threshold of an "OE-free" time span $\Delta t_{Ref}$ (see responses to "General Comments" above). The result shows that $\Delta t_{Ref}$ is ~3.2 years ($R^2$=0.96) for TG. Thus, $\Delta t_{Ref}$=1 year would be appropriate for a large number of glaciers in Antarctica, including TG. However, a reduced time span of ~3.0 months ($R^2$=0.99) for PIG and ~1.4 months ($R^2$=0.86) for Jakobshan Isbrae in Greenland should be used. As shown in the proposed regression equation, this threshold changes with how "fast" the ice flows in a glacier.

We changed the text for *Premise I* to: *"**Premise I:** Within a baseline time span (e.g., 1 year or shorter) each segment (from $P_{i-1}$ to $P_i$ in Figure 1b) is relatively short and the E- and L-velocity difference is smaller than σ. Furthermore, over the map span of n years (e.g., 5-10 years or longer) the accumulated E- and L-distances along the entire trajectory do not deviate significantly from each other, so that the maximum velocity difference in the Lagrangian and Eulerian frameworks (end points of blue and red lines in Fig. 2) is limited within a threshold ($V^L_{0-n} - V^E_{0-n} \leq k\ \sigma$), where k is a constant and σ is the velocity mapping uncertainty."*

**L134-136. Here the use of the i=1,2,…,n is confusing. You are describing the case of a short time span, hence why not just using the V$_{0-1}$ and U$_{0-1}$ (as you just said in the previous lines)? Or V$_{i-i+1}$ ? What is a short time span on Figure 2 ? All of this Premise holds in what you define as a "limited time span" and "short L trajectory" (which should be straight). Please clarify these points.**

Response:

(Line 152) Agreed. We now use "$P_{i-1}$ to $P_i$". Thus, we changed *Premise I*. See text above.

**The author choose to make a clear distinction between the theory vs the application,**

**which I think was a good idea, but here, it would help the readers to have some more self-explanatory examples, as it is done in section 2.3.**

Response:

(Lines 129-158) We revised the text of *Premises* and other paragraphs to make it more self-explanatory. For example, figures 1 and 2 are used in the text to explain the concept. That way we link the theory with "application" scenarios for better understanding.

**L 137. This could be reformulated, for the more clarity, to "In reality, the available historical images only allow us to produce eulerian velocity maps with a long timespan, i.e Map$_{0-n}$ which leads to the maximum overestimation value as defined in equation 2"**

Response:

(Line 159) Accepted. The sentence is rewritten accordingly.

**L138. "As we can only use Map$_{0-n}$ , the lagrangian velocity, for a long time span, is defined as follow". Please also add a reference to the line in Figure 2.**

Response:

(Line 162) Yes, the sentence is changed to: "… … *Based on this map V$_{0-n}$ of n-year span, the i-year span L-velocity (black line in Figure 2) is defined as follow:*"

We added the text to the black line in Figure 2: "*Average L-velocity $\overline{V}_{0-i}^{L(n)}$ from $V_{0-n}^{E}$*".

**L 141. "Consequently, the 1 year L-velocity U'$_{0-1}$…"**

Response:

Thanks. It is so changed.

**L145. What do you define again as a limited time span? If you compare Map$_{0-1}$ and Map$_{0-n}$, then you are comparing the smallest and largest time span, hence the use of the term "short timespan" is a bit confusing**

Response:

Now we clearly distinguish the baseline (or segment) span (e.g., 1 year or shorter) with the n year (max., entire trajectory) span (e.g., over 5-10 years) in *Premises I* and *II*.

(Line 152 in the marked-up manuscript) ***Premise I:*** *Within a baseline time span (e.g., 1 year or shorter) each segment (from P$_{i-1}$ to P$_i$ in Figure 1b) is relatively short and the E- and L-velocity difference is smaller than σ. Furthermore, over the map span of n years (e.g., 5-10 years or longer) the accumulated E- and L-distances along the entire trajectory do not deviate significantly from each other, so that the maximum velocity difference in the Lagrangian and Eulerian frameworks (end points of blue and red lines in Fig. 2) is limited within a threshold ($V_{0-n}^{L}$ −$V_{0-n}^{E}$ ≤ k σ), where k is a constant and σ is the velocity mapping uncertainty.*"

(Line 169) ***Premise II:*** *Within the time span of n years (e.g., over 5-10 years), the velocity field described by $V_{0-1}^{E}$ and $V_{0-n}^{E}$ does not change significantly, so that the line between $V_{0-n}^{L(n)}$ and $V_{0-1}^{L(n)}$*

*(black line in Fig. 2) and that between $V_{0-n}^L$ and $V_{0-1}^L$ (blue line in Fig. 2) are approximately parallel to each other. Accordingly, the difference between their simple averaged accelerations is within a threshold ( $\left|\overline{a}\left(V^{L(n)}\right) - \overline{a}(V^L)\right| \leq k' \frac{\sigma}{\Delta t_n}$ ), where $k'$ is a constant and $\sigma$ is the velocity mapping uncertainty and*

$$\overline{a}\left(V^{L(n)}\right) = \frac{V_{0-n}^{L(n)} - V_{0-1}^{L(n)}}{\Delta t_n}, \ \ \overline{a}(V^L) = \frac{V_{0-n}^L - V_{0-1}^L}{\Delta t_n}. \tag{7}"$$

**L145. I guess that the magnitude of the velocity Map$_{0-n}$ should be larger than Map0-1 , but the pattern is similar ? Can you provide a figure example with velocity direction to illustrate this point?**

Response:

We agree that the text along the three lines in Figure 2 is a bit confusing. The blue is the average L-velocity calculated from map $V_{0-1}$, and red line is the average E-velocity calculated from 1 year and n year E-velocities. Generally, the L-velocities are greater than E-velocities (see blue and red examples in Figure 5). We revised text in Figure 2 as follows:

*"Average L-velocity $\overline{V}_{0-i}^{L(n)}$ from $V_{0-n}^E$ (for black line)*

*Average L-velocity $\overline{V}_{0-i}^L$ from $V_{0-1}^E$ (for blue line)*

*Average E-velocity $\overline{V}_{0-i}^E$ from $V_{0-1}^E$ and $V_{0-n}^E$ (for red line)"*

**L148. The first part of the sentence can be removed since it has been described earlier, before Premise II. Then, you can just start with: "Hence, based on Premise II, we have…."**

Response:

Thanks. The text is revised accordingly.

**L150. Again, the U$_{0-n}$=V$_{0-n}$ is based on the fact that you are considering only short timespan. But is that the case if you use "0-n" ? (see earlier comment)**

Response:

If there is only spatial acceleration and $(V_{0-n}^L - V_{0-n}^E \leq k\ \sigma)$ in *Premise I*, $V_{0-n}^L \approx = V_{0-n}^E$ (or $U_{0-n} \approx V_{0-n}$), OE can be corrected; otherwise there is also temporal acceleration, we correct spatial acceleration induced OE, but preserve temporal acceleration – induced OE in residuals. We added the following:

(Line 190) "***Overestimation Correction Theorem****: Assume that the necessary condition in Premise II is met, spatial acceleration - induced overestimations in long time span velocities $V_{0-n}^E$ can be corrected or reduced using the following Correction term, regardless of temporal acceleration:*

$$V_{0-1}^E = V_{0-n}^E + Correction + \varepsilon, \tag{9}$$

$$Correction = V_{0-n}^E - V_{0-n}^{L(n)}. \tag{10}$$

*If Premise I holds, Correction $\approx -OE_{0-n}$; otherwise, $|Correction|<|OE_{0-n}|$, preserving the velocity increases induced by temporal accelerations in the residual term $\varepsilon$ (Fig. A2)."*

**L 151. I would reformulate this sentence to remind the reader about the aim of this paper : "Consequently, using the map with the longest timespan, we can go back to $V_{0-1}$ using a Correction term defined as Correction=$V_{0-n}$-$U'_{0-n}$".**

Response:

(Line 182) Thanks. The sentence is revised accordingly: "*Consequently, using the map with the longest time span, we can go back to $V_{0-1}^E$ using a correction term defined as Correction=$V_{0-n}^E - V_{0-n}^{L(n)}$:* "

**L 164. What does it mean to interpolate the positions to the sub grid-level ? Does it make any sense to interpolate the position at a higher level of resolution than the velocity field?**

Response:

(Line 215) The interpolation is not used to make a new higher resolution velocity map, but to determine the positions of the distance segments for L-distance integration. Thus, the intermediated sub-grid positions are used for a continuous distance integration. The sentence is changed to: "*The sub-grid positions of the monthly segments are interpolated for integration of the overall L-distance.*"

**Figure 3. Please add a general Figure of the entire Pine Island glacier, to check out where the location of your flowline is (similar as Figure 4).**

Response:

It is done. Thanks.

**L179-180. What about orthorectification errors in historical images ?**

Response:

(Line 232) We added it to the sentence: "*Despite the sub-pixel accuracy of the orthorectification of historical images and ……*"

**Acceleration computation: this has already been described L147. I would suggest to move this part earlier (or remove it).**

Response:

(Line 198) This section is now moved to the earlier part.

**Section 3.1. Since section 2.3 shows an example over Pine Island glacier, I don't know why the author didn't continue using this example. Since the overestimation is quite spectacular, I would strongly suggest to use Pine Island instead of Totten here.**

Response:

(Line 391) Yes, we accepted your suggestion and added the PIG results as a new section *Experiment 3.*

**L213. Why did you choose a 7 year trajectory ?**

Response:

(Line 266) We found the earliest available high-quality Landsat 8 images in 2013 and latest ones in 2020 (7 years). We changed the sentence: "*To avoid lower quality historical velocity maps that may influence the effectiveness of the validation, we use the earliest available high-quality Landsat 8 images from 2013 to 2020 to produce velocity maps V$_{2013-i}$ (i=2014, ... 2020)......*"

**L217. See previous comment on the choice of symbols in equations.**

Response:

They are all fixed throughout the manuscript (see responses to previous comments)

**L215. Do you generate the map separately or over the entire glacier directly ?**

Response:

Because we have to generate maps for 7 time spans for validating *Premises I* and *II*, it is a lot of work to produce them all. Therefore, we only mapped areas of 5 trajectories separately, instead of the entire glacier directly.

**L225. I am surprised about the error estimation here. Millan et al., 2019; had some smaller number for 1 year map of ice velocity using Landsat-8. Can you discuss why is that? How does your map compare with available NSIDC data ? What is the difference with recent map assembled from sar interferometry? (see Mouginot et al., 2019)**

Response:

The highest accuracy of less than 1 m/year was achieved by using **InSAR** technique (Mouginot et al., 2019), which used the InSAR phase information and the data requirement is generally high. An accuracy of 10 m/year was reported by Millan et al. (2019) for average annual velocities from **multiple individual velocities** derived from Sentinel-2 (10 m resolution) and Landsat-8 (15 m resolution) and other images. Similarly, Gardner et al. (2018) also achieved an accuracy of 10 m/year of annual velocities by averaging velocities derived from **multiple Lansat-7 and -8 image pairs** (15 m resolution). However, our velocity sub-maps (5 areas) were built from **only one Landsat-8 image pair** (15 m resolution) for each map. Given the accuracy of individual maps as $\sigma^2{}_i$, the accuracy of the averaged velocity is $=$ $\frac{\sqrt{\sum \sigma_i^2}}{n}$. In general, the accuracy of the average annual velocity should be smaller than that of the velocity of an individual pair. Therefore, our accuracy of 20 m/year for 1-year (individual pair) and 3 m/year for 7-year (individual pair) velocities are reasonable values.

References

Mouginot, J., Rignot, E., & Scheuchl, B.(2019). Continent wide, interferometricSAR phase, mapping of Antarctic icevelocity. Geophysical Research Letters,46, 9710–9718.

https://doi.org/10.1029/2019GL083826

Millan, R., Mouginot, J., Rabatel, A., Jeong, S., Cusicanqui, D., Derkacheva, A., & Chekki, M. (2019). Mapping surface flow velocity of glaciers at regional scale using a multiple sensors approach. Remote Sensing, 11(21). https://doi.org/10.3390/rs11212498

Gardner, A. S., Moholdt, G., Scambos, T., Fahnstock, M., Ligtenberg, S., van den Broeke, M., and Nilsson, J.: Increased West Antarctic and unchanged East Antarctic ice discharge over the last 7 years, The Cryosphere, 12, 521–547, https://doi.org/10.5194/tc-12-521-2018, 2018.

**L239. Is it "faster than" or "close" ?**

Response:

(Line 302) We deleted "close to".

**L240. I think that the use of "J" in exponent is adding to much complication (see earlier comment). You can just specify that you do the calculation for all sub-images.**

Response:

We accepted your suggestion and removed "J" throughout the manuscript.

**L257. The "apparent parellelity" is quite subjective I think. Is the premise II validated for case 2c and 2a?**

Response:

(Line 320) Cases 2c and 2a are ok (average difference of 1.2 m $a^{-2}$). Cases 3c and 3a are 2.1 m $a^{-2}$ (max) that is still within the allowable uncertainty of 3 m $a^{-2}$ determined based on the velocity mapping uncertainty of 20 m $a^{-1}$. To make it objective, we changed "*apparent parellelity*" to "*relatively well-maintained parellelity*".

**Figure 5. Add the direction of the flow in the sub-images. You could also consider using different symbols for the 7 yr and 1 yr trajectory and use a color gradient for the position of the points that changes with the year. This "year" color could then also be used in the scatter plots.**

Response:

We revised Fig. 5 accordingly.

**Table 1. Please add more details on the caption of the Table, ie, the content of each column.**

Response:

(Line 326) The caption is revised: "*Table 1. Velocity and acceleration in Eulerian and Lagrangian frameworks used for validation of the overestimation correction method. "Actual E-velocity and OE" lists actually mapped 1-year and 7-year E-velocities and their differences as overestimations in all five areas. "Premise I" contains 7-year L-velocities computed from the 1-year velocity map and corresponding L- and E-velocity differences, which are used for validating Premise I. "Premise II" illustrates averaged L-accelerations computed from the 7-year and 1-year velocity maps, respectively, as well as their differences, which are used for validating Premise II. "Overestimation correction" presents 7-year L-velocities computed from the 7-year map, overestimation corrections, and E-velocities and residuals (or errors) after correction.*"

**L302. The acronym OE has been defined before.**

Response:

We deleted "(OE)".

**Section 4 Discussion. Can you discuss the performance of your method, with the relatively simple approach defined by Berthier et al., 2003? I think that the section is missing some discussion on the applicability of the method to 1) fast glacier, 2) the glacier geometry, which can be much more complex than the glaciers that were used here to validate the method (not straight) (see earlier comment). An additional discussion about the significance of the correction, with respect to the seasonal variation in ice flow velocity of the glacier should also be discuss, if the method is expected to be applicable in Greenland. Specifically, does the magnitude of the correction could exceed the natural variability of the glacier? I guess that the amount of acceleration with a flowline would need to be significant in order to induce a correction that would exceed the variability of the seasonal signal?**

Response:

The comparison results with the "Midpoint" method are presented in the response to comment L54. (Line 484) We also added a section "*4.4 Comparison with the "Midpoint" method*" in Discussion.

The responses related to fast glaciers, complex geometry, seasonal variability, applicability in Greenland etc. are presented in responses to General Comments and in Discussion.

**Figure 5-6. Can you provide a figure of the corrected ice velocity? Maybe a difference map.**

Response:

We produced the map of DG after correction (Fig. R1-7b). It does not appear distinctly different from the map before correction (Fig. R1-7a = Fig. 7b). Thus, we will not include this corrected map in the main text. The map with the OEs (Fig. 7c) is actually the difference map. Similarly, the corrected submaps in TG are also visually not distinct from those before correction. Similarly we will not add the corrected or difference map to Fig. 5.

[Figure]

Figure R1-7. E-velocity in TG: (a) before OE correction, and (b) after OE correction.

We appreciate the constructive comments from two reviewers and the community. Our manuscript will be much improved by their input. We have made changes to our manuscript. In the following responses, we use "**bold**" text for comments, "non-bold" text for our responses, and "*italic*" for changed text in the manuscript.

**Referee #2 (Chad A. Greene):**

**This paper identifies three key shortcomings of a common velocity measurement technique, and provides a solution that addresses all three. At issue are 1. the true location of a feature tracked velocity measurement, 2. the acceleration that a parcel of ice may experience between image acquisition times, and 3. the fact that ice does not always move in a perfectly straight line between image acquisition times. The problem and solution are described well in this paper, and the authors demonstrate that they have a good handle on the data and how velocities are interpreted from a glaciological standpoint.**

**This work will be of value to the community, both to raise awareness of the overestimation issue, and to provide a solution to it. I recommend publication, with just a few suggestions that may help readers understand the impact of overestimation and how it should affect our interpretation of previous studies.**

**Main Issues**

**The paper does a good job of describing the problem and solution from a technical standpoint, and anyone who has written feature-tracking algorithms will benefit from reading the paper. However, there are many readers who don't write their own algorithms, but will nonetheless want to understand how overestimation might affect their scientific results. Some work could be done in this paper to better communicate the overall impact of how overestimation impacts long-term studies.**

**Here's a type of analysis that I would find much more insightful than the stats for PIG, Totten, and David Gl that are currently presented in the abstract: I would like to see a figure showing Eulerian grounding line flux calculations as a function of dt, where dt might range from a day to 20 years. This would provide readers with some intuition for a threshold value of dt, beyond which Eulerian measurements produce significantly different estimates of ice flux. It's possible that the percentage reduction in GL flux as a function of dt might vary regionally, and that diversity could be interesting to show as well.**

Response:

We performed an experiment of "GL flux vs. time span" for PIG. We used a baseline velocity map of PIG 2013 (Fig. R2-1a) from ITS_LIVE (Gardner et al., 2019). The flux gate (red line)

is set along the grounding line (GL, black line) and separated into flux nodes every 240 m where the ice flow velocity and ice thickness data (BedMachine) are used to calculate ice flux. We calculated L-velocities along GL with time spans of 1-15 years based on the 2013 E-velocity map (Fig. R2-1b). Instead of suggested 20 years, we used 15-year time span mainly because the 20-year tracking distance from GL would run beyond the ice shelf front.

[Figure]

Figure R2-1: (a) Annual ice velocity map of PIG (2013) from ITS_LIVE (Gardner et al., 2019) with the flux gate (red line) set on GL (black line), (b) L-velocities of 1- to 15-year span $V^L_{0-i}$ (*i=1, 2, ... 15*) along GL calculated from the 2003 annual map, (c) GL ice flux (estimated from $V^L_{0-i}$) vs. time span, and (d) L-velocities $\bar{V}^L_{0-i}$ (averaged over the GL portion across main trunk, over 1500 m a$^{-1}$) vs. time span.

The L-velocity along GL increases mainly in the margin areas of the main trunk as the time span increases (Fig. R2-1b). The maximum velocity OE reached ~1300 m a$^{-1}$ for the 15-year span (marked "A" in Fig. R2-1b). In this place, the annual maximum OE rate started at ~461 m a$^{-1}$ and decreases to ~31 m a$^{-1}$ as the time span increases, because the later part of the trajectory reached the flat part of the ice shelf front (high velocity, but low acceleration).

Consequently, the OE induced GL flux increase (flux overestimation) speeds up quickly at an annual rate of ~2.1 Gt a$^{-1}$ for the first 4 annual spans, by ~6.3 Gt a$^{-1}$ from ~116.7 Gt a$^{-1}$ to ~123.0

Gt a$^{-1}$. Thereafter the annual rate is maintained at 0.4 Gt a$^{-1}$ until the 15-year span, reaching the maximum flux OE of 11.5 Gt a$^{-1}$. This flux OE has the same trend pattern as the average L-velocity over the GL portion across main trunk (Fig. R2-1d), indicating that the main flux OE "came" across GL from the main trunk portion.

Due to the limited time and amount of work, we will investigate the same issue in other glacier regions (the Totten and David glaciers) in the future. We added the following text in Discussion:

(Line 471 in the marked-up manuscript) "*……In addition, based on an annual E-velocity map of 2013 in PIG from ITS_LIVE, L-velocities along grounding line (GL) with time spans of 1 to 15 years and the associated GL ice flux were computed. The results show that the velocity OEs of the 15-year span can reach up to ~1,300 m a$^{-1}$ in the GL region. Such OEs in velocity can further cause an overestimation in GL flux, which is negligible within a 3-year span ($\leq \sigma_{Flux}$). We used $\sigma_{Flux}$=5.8 Gt a$^{-1}$ based on the flux uncertainty in PIG reported by Rignot et al. (2019). The GL flux OE increases rapidly by ~6.3 Gt a$^{-1}$ within the first 4-year span; thereafter it slows down until a maximum of 11.5 Gt a$^{-1}$ is reached at the 15-year span. Therefore, the influence of the velocity OEs on the GL flux appears to be not very significant (11.5 Gt a$^{-1}$ $\leq 2\ \sigma_{Flux}$).*"

**In addition to a figure showing how dt affects GL flux in Eulerian measurements, I'd like some clear guidance in the abstract for when the Eulerian approximation is sufficient or insufficient.**

The above simulation result shows that a GL flux OE of ~11.5 Gt a$^{-1}$ in PIG would be induced by a 15-year span L-velocity map, which is significant in comparison to the flux uncertainty $\sigma_{Flux}$ of 5.8 Gt a$^{-1}$ in PIG given by Rignot et al. (2019). Therefore, assuming $\sigma_{Flux}$ = 5.8 Gt a$^{-1}$, we estimated a "flux-OE-free" time span of ~3 years using the curve in Fig. R2-1c.

Please see the added text in Discussion (Line 476; above).

(Line 26) We added a statement in Abstract: "*……Our experiment results in PIG show that, if not corrected, the OEs can further cause an overestimated grounding line flux that is negligible within a 3-year span, but reaches the maximum of 11.5 Gt a$^{-1}$ with a 15-year span…….*"

**In the abstract and/or discussion, I suggest flipping the logic/wording around at least once to make it clear that the overestimation of historical velocities could mean that previous papers have underestimated the magnitude of glacier acceleration over the past few decades. It's only a minor change in wording, but I think it's an important take-home message of this paper that should be stated directly.**
Response:
(Line 27) We added a statement in Abstract: "*…… The implication is that, when using newer velocity maps of short spans along with historical maps of long spans produced in previous studies over the past few decades, the overestimation of historical velocities could have caused an underestimation*

*in the long-term acceleration magnitude. We recommend that overestimations of more than the velocity mapping uncertainty (1σ) be corrected……"*

(Line 498) We added a statement in the discussion section: "*……The implication is that, when using newer velocity maps of short spans along with historical maps of long spans produced in previous studies over the past few decades, the overestimation of historical velocities could have caused an underestimation in the long-term acceleration magnitude. On the other hand, new efforts in historical velocity mapping at an ice sheet – wide or large regional scale should be made with a full consideration of OE corrections……"*

**Minor comments**

**Abstract: The case studies of PIG, Totten, and David Glacier provide decent testing grounds for the methods presented in this paper, but the details of these studies feel somewhat anecdotal and very specific to the exact images that were used in these particular cases. I recommend generalizing the results in the abstract to give readers a better overview of the problem. Only after discussing the overall impact of the overestimation, then it may be helpful to mention a specific case of PIG, Totten, or David to as a tangible example.**

Response:

(Line 11) Accepted. We revised Abstract accordingly: "*…… In comparison to velocity maps derived from recent satellite images of monthly to weekly time spans, historical maps, from before the 1990s, generally cover longer time spans, e.g., over 10 years, due to the scarce spatial and temporal coverage of earlier satellite image data. We found velocity overestimations (OEs) in such long-span maps that can be mainly attributed to ice flow acceleration and time span of the images used. If used for long-term change studies, these OEs in historical velocities may further affect the estimated trends of ice flow dynamics and mass balance. For example, the OEs can reach from ~69 m a$^{-1}$ (7-year span) in Totten Glacier (TG), East Antarctica, up to ~930 m a$^{-1}$ (10-year span) in Pine Island Glacier (PIG), West Antarctica……*"

**L29: This line mentions "the input-output method" and some good references are provided for it, but some readers may be unfamiliar with the term. If the term is necessary for some point that's being made, then I recommend briefly describing what is meant by "the input-output method" here. If the term is not important for this paper, then consider removing it.**

Response:

The phrase "using the input–output method" is removed.

**L69: Recommend changing "It is proven that…" to "We show that…" to make it clear that the correction is original work that is presented in this paper.**

Response:

We changed the sentence accordingly.

**L80: I'm not entirely sure what "descending passages" means. Consider rewording.**
Response:

(Line 91) The sentence is revised: "*We describe an acceleration - induced overestimation using a typical scenario in AIS (Fig. 1a) where ice flow accelerates over a long slope from several glaciers originated from the inland interior, running through the main trunk, and discharging to the ocean.*"

**L160: "At each grid…" I think this should be "At each grid cell…" or "At each pixel…"**
Response:

We changed it to "*At each grid cell ……*"

**Figure 5 is very compelling, and I want to make sure I understand it. Unfortunately, the labels and caption are somewhat cryptic, so I'm not sure if I'm even getting the main message right. The caption contains a list of the data labels that are mostly redundant with labels that are presented directly in the figure. What's missing is physical interpretation or any direct take home message. For example, the variables U, U', and V are labeled in the figure and in the caption, but there's no physical definition of what U, U', or V mean. Help readers by providing a sentence or two in the caption that directly states the main point and any secondary point(s) that may be worth noticing. The main point, I assume, is that the black line is consistently higher than the red and blue curves. State that in the caption, in terms of what it means physically. What causes the red and blue curves to cluster together or spread apart from each other? Mention the underlying mechanism in the caption. Most of these points are described in detail on page 11, but most readers will appreciate having the main points stated directly in the figure caption.**
Response:

(Line 292) The caption is revised according to the suggestions: "*Figure 5: Velocities in five areas (rectangles in Fig. 4) of the Totten Glacier are used to validate Premises I and II. (a) Panels 1a–5a show the reconstructed 1-year velocity maps $V_{2013\text{-}2014}$ in the areas; matched points (red triangles) are used to map E-velocity $V^E_{2013-i}$ (i=2014, …, 2020) (red lines in Panels 1c-5c); points along the flow line (blue dots) are tracked from the 1-year maps and used to calculate L-velocity $V^L_{2013-i}$ (blue lines in Panels 1c-5c). (b) Similarly, Panels 1b–5b illustrate the reconstructed 7-year velocity maps $V_{2013\text{-}2020}$ in the areas with the matched points (red triangles) for E-velocity $V^{E(7)}_{2013-i}$; points along the flow line (black crosses) are tracked from the 7-year span velocity map and used to calculate L-velocity $V^{L(7)}_{2013-i}$ (black lines in Panels 1c-5c). (c) In each area (Panels 1c–5c) the difference between red and blue lines increases with time, but is limited within k $\sigma$ at the end (Premise I), except a large k in Area 4 because of effect of a calving event; the black line is above blue line due to the spatial acceleration - induced OE over 7-year span; but they are relatively parallel (Premise II) and thus, a correction can be estimated.*"

We appreciate the constructive comments from two reviewers and the community. Our manuscript will be much improved by their input. We have made changes to our manuscript. In the following responses, we use "**bold**" text for comments, "non-bold" text for our responses, and "*italic*" for changed text in the manuscript.

**Community comment (Massimo Frezzotti):**

**Very interesting paper, the manuscript does not take in account the previous studies on glacier analyzed. To improve the results of their paper I suggest the authors to compare their result with previous ice velocity analysis using satellite image and also with GPS measurements.**

**For David, Reeves, Priestley several papers and measurements are available since 1998:**

**Frezzotti M., Capra A. & Vittuari L. (1998) Comparison between glacier ice velocities inferred from GPS and sequential satellite images. Ann. Glaciology, 27, 54-60,**

**Frezzotti M., I. Tabacco and A. Zirizzotti (2000) Ice discharge of eastern Dome C drainage area, Antarctica, determined from airborne radar survey and satellite image analysis. J. of Glaciology, Vol 46 (153), 253-273, DOI: 10.3189/172756500781832855**

**Danesi, S., Dubbini, M., Morelli, A., Vittuari, L., & Bannister, S. (2008). Joint geophysical observations of ice stream dynamics. In Geodetic and Geophysical Observations in Antarctica (pp. 281-298). Springer, Berlin, Heidelberg.**

**Rignot E, Mouginot J, Scheuchl B (2011) Ice flow of the Antarctic Ice Sheet. Science 333:1427–1430.**

**Stearns, L. A. (2011). Dynamics and mass balance of four large East Antarctic outlet glaciers. Annals of Glaciology, 52(59), 116-126.**

**Mouginot J, Rignot E, Scheuchl B, Millan R (2017) Comprehensive annual ice sheet velocity mapping using Landsat-8, Sentinel-1, and RADARSAT-2 data. Remote Sens 9:364–1370.**

**https://earthdata.nasa.gov/esds/competitive-programs/measures/ice-velocity-mapping-of-the-great-ice-sheets-antarctica**

**Moon, J., Cho, Y., & Lee, H. (2021). Flow Velocity Change of David Glacier, East Antarctica, from 2016 to 2020 Observed by Sentinel-1A SAR Offset Tracking Method. Korean Journal of Remote Sensing, 37(1), 1-11.**

**Your Sincerely**
**Massimo Frezzotti**

Response:

(Line 45 in the marked-up manuscript) Thank you for the comments, well received. We added some of your suggested references in Introduction: "*......, regional velocity maps at a seasonal or monthly scale have been generated from optical and SAR images (e.g., Landsat and Sentinel; Frezzotti et al., 1998, 2000; Nakamura et al., 2010; Zhou et al., 2014; Greene et al., 2017, 2018, 2020a; Moon et al., 2021).*"

Frezzotti M., Capra A. & Vittuari L. (1998). Comparison between glacier ice velocities inferred from GPS and sequential satellite images. Ann. Glaciology, 27, 54-60.

Frezzotti M., I. Tabacco and A. Zirizzotti (2000). Ice discharge of eastern Dome C drainage area, Antarctica, determined from airborne radar survey and satellite image analysis. J. of Glaciology, Vol 46 (153), 253-273, DOI: 10.3189/172756500781832855.

Moon, J., Cho, Y., & Lee, H. (2021). Flow Velocity Change of David Glacier, East Antarctica, from 2016 to 2020 Observed by Sentinel-1A SAR Offset Tracking Method. Korean Journal of Remote Sensing, 37(1), 1-11.

(Line 349) In *Experiment 2* where the David Glacier region is used for demonstration of the proposed method, we added the following sentences to recognize the previous work and comparative coverages of the velocity maps: "*...... Velocities in this region from 1988 to 1992 were mapped by using GPS and image feature tracking techniques (Frezzotti et al., 1998, 2000). A new GPS campaign was carried out in the region during 2005-2006 (Danesi et al., 2008). Velocity changes from 2016 to 2020 were detected using Sentinel-1A SAR images (Moon et al., 2021). In this experiment we produced a velocity map of the region from 64 Landsat images collected from 1972 to 1989 (Table A3)......*"

---

## Referee Report (RR1)

**Second Review of Li et al's Overestimation and Adjustment of Antarctic Ice Flow Velocity Fields Reconstructed from Historical Satellite Imagery**

Chad A. Greene
December 19, 2021

**General comments**

In my first review of this paper, I commended the authors for raising awareness of an issue that could potentially introduce a systematic bias into any study of ice dynamics that compares recent satellite measurements to velocities measured decades ago, when the satellite record was sparse and image pairs were often separated by many years. The paper presented a reasonable approach for dealing with overestimation from a technical standpoint, but I felt the manuscript needed improvement in communicating
1. why the overestimation effect matters,
2. how big of a problem it is, and
3. when it should be taken into account in scientific studies.

On these three points, I don't see that much has changed in this revised manuscript. A few sentences were added in this revision to mention the effects of OE at PIG, but no attempt was made to put the effect into any greater context or generalize the findings beyond the specific case at PIG. Notably, one of the new sentences states that OE is negligible when computing GL flux at PIG. As a reader I am left wondering, if OE is negligible at PIG, does it matter anywhere?

Most of this paper is devoted to specific case studies of PIG, Totten, and David Glaciers, but the results are never generalized beyond the specific image pairs that were analyzed in this study. Readers are tasked with absorbing pages of details about tenth-of-a-meter-per-year level accelerations in some arbitrary pixels from a handful of image pairs, but the findings from these case studies are never synthesized in any way that could be applied beyond the image pairs that are analyzed in this study.

In my previous review, I suggested that readers would benefit from clear guidance on when to consider the effects of OE. The authors responded with a beautiful Figure R2-1 that for some reason is not included in the manuscript. I've recreated panel C of R2-1 below (the code is provided at the end of this review) and I find a similar overall curve for PIG. The GL flux curve for PIG takes an interesting shape—why does the OE effect appear to level off after a couple of years? That phenomenon is worth exploring and discussing in the manuscript.

Is the same effect seen elsewhere around Antarctica? When I synthetically "measure" GL flux as a function of dt for the entire continent[*], it doesn't appear to matter much on the whole. This suggests that while overestimation is in effect at PIG, there is some compensating
* * *
[*] My GL flux values for "the entire continent" are low, because I excluded GL datapoints that would calve into the ocean within 15 years.

*underestimation* elsewhere around Antarctica. Where does this happen? A discussion of the physical phenomena that affect ice velocity measurements would be insightful.

[Figure]

*Summary:* I don't believe there's any particular harm in publishing the manuscript in its current form, but I do believe an opportunity has been missed for the authors to share the insights and intuition they have surely developed over the course of this work.

**Specific comments**

**Line 394** alludes to experiments that were performed on Jakobshavn Isbrae and introduces the acronym JI for the glacier, but the experiments at Jakobshavn are not described anywhere in the paper, and the acronym is never used. Am I missing something?

**Line 399** starts a new section with the word "Furthermore,…" as if a thought is being continued from a previous section. Section breaks are often where readers pause to grab a cup of coffee, so as a general rule I recommend starting each section in such a way that readers can pick up where they left off, without having to re-read the previous section.

**Line 339** says GL flux was computed. How? What grounding line and ice thickness products were used?

**Line 440** says "OEs in velocity can further cause an overestimation in GL flux, which is negligible within a 3-year span" but no evidence or rationale for this statement is presented in the manuscript.

**Line 443** says "Therefore, the influence of the velocity OEs on the GL flux appears to be not very significant." Does this mean the main thesis of the paper is more theoretical than practical?

**Lines 445-449** offer concluding remarks on the Jakobshavn experiments that are not described in this paper. This paragraph can be removed.

**Matlab code**

```matlab
% Load velocity data:
[vx,x,y] = itslive_data('vx');
vy = itslive_data('vy');

vx = double(vx);
vy = double(vy);

[X,Y] = meshgrid(x,y);

% Fill data gaps:
isn = isnan(vx);
vx(isn) = measures_interp('vx',X(isn),Y(isn));
vy(isn) = measures_interp('vy',X(isn),Y(isn));

ground = ismember(bedmachine_interp('mask',X,Y),[1 2 4]);

% mask2outline is in the Climate Data Toolbox
[glx,gly] = mask2outline(x,y,imfill(isfinite(vx) & ground,'holes'),...
    'buffer',-1000,'regions',1);

glx = glx';
gly = gly';

% Load thickness data:
H = bedmachine_interp('thickness',glx,gly);

dt = 0.5; % time step for advection
t = (0:dt:15)'; % time (years)

% Passively advect the GL points downstream, one dt at a time:
for k=2:length(t)

   % Grow the array with a new column for each dt:
   glx = [glx;glx(k-1,:)+dt*interp2(x,y,vx,glx(k-1,:),gly(k-1,:))];
   gly = [gly;gly(k-1,:)+dt*interp2(x,y,vy,glx(k-1,:),gly(k-1,:))];

end

% Measured velocity:
vxi = (glx-glx(1,:))./t;
vyi = (gly-gly(1,:))./t;

% fix the 0/0 problem for time zero:
vxi(1,:) = interp2(x,y,vx,glx(1,:),gly(1,:));
vyi(1,:) = interp2(x,y,vy,glx(1,:),gly(1,:));
vi = hypot(vxi,vyi);

% "Measure" flux each year:
flux = [];
for k = 1:length(t)
   flux = [flux;(gradient(glx(1,:)).*H.*vyi(k,:) -
gradient(gly(1,:)).*H.*vxi(k,:))*917*1e-12];
end
```

```matlab
% Indices of data points that are good for the entire record:
% (Caveat: Most Thwaites and many Antarctic Peninsula GL points end up in
% the ocean within 15 years, so they aren't counted in Antarctica's total.)
allgood = all(isfinite(flux));

% Indices of PIG points:
pig = inbasin(glx(1,:),gly(1,:),'imbie refined','pine island');

figure
subplot(1,2,1)
plot(t(3:end),sum(flux(3:end,allgood&pig),2))
box off
axis tight
xlabel 'years dt'
ylabel 'gl flux (Gt/yr)'
title PIG

subplot(1,2,2)
plot(t(3:end),sum(flux(3:end,allgood),2))
box off
axis tight
xlabel 'years dt'
ylabel 'gl flux (Gt/yr)'
title Antarctica
```

---

## Author Response (AR2)

We appreciate the constructive comments and suggestions from the editor and reviewer. Our manuscript will be much improved by their input. We have made changes to our manuscript. In the following responses, we use "**bold**" text for comments, "non-bold" text for our responses, and "*italic*" for changed text in the manuscript.

**Editor:**

**I fully agree with the reviewer. The key now for you is to put your results into context and make sure the readers/users are aware of the validity domain of the correction you propose. This will help our colleagues to fully understand the importance of your contribution.**

Response:

Thanks. The manuscript is revised accordingly to explain the validity domain of the correction and importance of the contribution (mainly in the first response and then in detailed responses in the following text).

**Referee #2 (Chad A. Greene):**

**Main Issues**

**In my first review of this paper, I commended the authors for raising awareness of an issue that could potentially introduce a systematic bias into any study of ice dynamics that compares recent satellite measurements to velocities measured decades ago, when the satellite record was sparse and image pairs were often separated by many years. The paper presented a reasonable approach for dealing with overestimation from a technical standpoint, but I felt the manuscript needed improvement in communicating**

    **1. why the overestimation effect matters,**

    **2. how big of a problem it is, and**

    **3. when it should be taken into account in scientific studies.**

**On these three points, I don't see that much has changed in this revised manuscript. A few sentences were added in this revision to mention the effects of OE at PIG, but no attempt was made to put the effect into any greater context or generalize**

**the findings beyond the specific case at PIG… …**

Response:

(Line 387 in the marked-up manuscript) After presenting three experiments in Totten, Davis and PIG in Section 3 *Results*, we added a paragraph to address these three points: "*In summary, OEs exist in historical long span velocity maps of Antarctica, from smaller glaciers, such as David Glacier, to the fast-flowing glacier of PIG, where spatial accelerations are produced by, e.g., bed topography and slopes. In general, they are less significant in slow-flowing grounded regions with low spatial accelerations. Instead, they take effects in places of high ice dynamics, for example, near grounding lines and often in ice shelf fronts. Velocities in these areas are important for estimating ice sheet mass balance and contribution to global sea level changes, and for analyzing ice shelf instability. For instance, in the David Glacier the large OE corrections (up to 36 m $a^{-1}$) occur on the ice shelf (Fig. 6). The OEs of a 7-year span, up to 69 m $a^{-1}$ (Table 1), would be about ~50% of the velocity increase detected during 1989 – 2015 in the Totten Glacier (Li et al., 2016). The PIG experiment showed an extreme case in Antarctica where the OEs of a 7-year span can go as large as 626 m $a^{-1}$ (~20%) near the grounding zone (Table 3); furthermore, the OEs of a 15-year span can reach up to 1,300 m $a^{-1}$ along the grounding line and cause an overestimated GL flux of 11.5 Gt $a^{-1}$ if not corrected (Fig. A4). Therefore, the magnitude of the OEs contained in the long span historical velocity maps is significant. When overestimated historical maps of 1960s – 1980s are used alongside recent maps of 1990s – 2010s for assessment of the long-term global climate change impact on the Antarctic ice sheet and for forecast modeling, overestimated historical states may lead to underestimated long-term changes. Furthermore, compromised forecasting may be resulted. Thus, the OEs in the long span historical maps must be seriously examined and corrected.*"

(Lines 17 and 525) Accordingly we made changes in Abstract and Conclusions.

**… … Notably, one of the new sentences states that OE is negligible when computing GL flux at PIG. As a reader I am left wondering, if OE is negligible at**

**PIG, does it matter anywhere?**

Response:

(Line 465) We deleted the sentence "…… "

**Most of this paper is devoted to specific case studies of PIG, Totten, and David Glaciers, but the results are never generalized beyond the specific image pairs that were analyzed in this study. Readers are tasked with absorbing pages of details about tenth-of-a-meter-per-year level accelerations in some arbitrary pixels from a handful of image pairs, but the findings from these case studies are never synthesized in any way that could be applied beyond the image pairs that are analyzed in this study.**

Response:

We synthesized the findings and generalized the results. The text is included in the new paragraph in *Results* (Line 384, see the response above to the first comment).

**In my previous review, I suggested that readers would benefit from clear guidance on when to consider the effects of OE. The authors responded with a beautiful Figure R2-1 that for some reason is not included in the manuscript. I've recreated panel C of R2-1 below (the code is provided at the end of this review) and I find a similar overall curve for PIG. The GL flux curve for PIG takes an interesting shape—why does the OE effect appear to level off after a couple of years? That phenomenon is worth exploring and discussing in the manuscript.**

Response:

(Lines 460 and 723) We give a guidance on when to consider the effects of OE based on the GL flux trend in PIG, and added the suggested figure in Appendix as Figure A4. In *Discussion* we added the following text: "…… *Such OEs in velocity can further cause an overestimation in ice discharge cross flux gates upstream the GL, which increases rapidly by ~6.3 Gt $a^{-1}$ within the first 4-year span (Fig. A4c); thereafter the flux overestimation slows down until a maximum of 11.5 Gt $a^{-1}$ is reached at the 15-year*

*span. This suggests that in fast-flowing glaciers like PIG, OEs in velocity maps with a time span of greater than 0.5 – 2 years should be corrected. The fact that the OE effect in PIG appears to level off after a couple of years is mainly because the velocity in the majority of the ice shelf is approximately leveled at ~4000 km a-1 and the acceleration is thus very small. This makes the average L-velocities along the GL to increase in a much-reduced pace where an integral of E-velocities is performed in the leveled velocity area a few years after crossing the GL (Fig. A4d). Consequently, the flux OE indicated in Fig. A4c showed a similar trend."*

**Is the same effect seen elsewhere around Antarctica? When I synthetically "measure" GL flux as a function of dt for the entire continent\* , it doesn't appear to matter much on the whole. This suggests that while overestimation is in effect at PIG, there is some compensating underestimation elsewhere around Antarctica. Where does this happen? A discussion of the physical phenomena that affect ice velocity measurements would be insightful.**

**\* My GL flux values for "the entire continent" are low, because I excluded GL datapoints that would calve into the ocean within 15 years.**

[Figure]

**Figure 1. GL flux in PIG and Antarctica (Chad A. Greene)**

Response:

Thanks for your code. We used it to calculate GL flux of a set of selected glaciers. The results of four glaciers are illustrated below.

[Figure]

Figure 2. Velocity map from ITS_LIVE (Gardner et al., 2019), flow lines (black), grounding line (red), and GL flux based on 1- to 15-year span L-velocities (blue curve on the right side of each map) in the PIG (a), Byrd (b), Totten (c), and David (d) glaciers.

[Figure]

Figure 3. Trend of long span L-velocity directions – approaching the ice shelf margin orientation, and its impact on flux computation.

For a GL point in a given glacier (Fig. 3) the actual GL flux estimated from a non-biased $V_{0-0}^E$ map is assumed to be $Flux_0$. $Flux_i$ is calculated from the overestimated i-year span L-velocity $V_{0-i}^L$:

$$Flux_i = \overrightarrow{Gate} \times \overrightarrow{V_{0-i}^L} \cdot Thickness \cdot \rho_{Ice} \cdot dt = \left|\overrightarrow{Gate}\right| \left|\overrightarrow{V_{0-i}^L}\right| \sin\theta_{0-i} \cdot Thickness \cdot \rho_{Ice} \cdot dt \qquad (1)$$

"$\times$" is cross product. As time span $i$ increases from 1 to 15 years, $Flux_i$ varies according to both $\left|\overrightarrow{V_{0-i}^L}\right|$ and $\theta_{0-i}$ (angle between $\overrightarrow{Gate}$ and $\overrightarrow{V_{0-i}^L}$, see Fig. 3). The GL flux gate has a fixed direction for all years, while the $i$-year span L-velocity $(\overrightarrow{V_{0-i}^L})$ ($i=1, ...n$) direction **changes** each year. Given a velocity range in a glacier, $\theta_{0-i}$ controls the flux magnitude. A large $\theta_{0-i}$ angle allows more ice mass to cross GL (e.g.

at a perpendicular angle), otherwise the flux decreases. The extreme case is flux = 0 when $\overrightarrow{V_{0-i}^L}$ is parallel to GL ($\theta_{0-i} = 0$). Changing of $\theta_{0-i}$ over a long span (e.g., from 1 to 15 years) and its impact on flux computation is the focus here.

In PIG (Fig. 2a) majority of flow lines cut GL at large angles (approximately perpendicular to GL or large $\theta_{0-i}$) that are mostly not changed significantly over 15 years. Thus, $Flux_i$ increases for all 15 years (a total flux OE of 11.5 Gt a⁻¹) due to continued large $\theta_{0-i}$ and velocity OE in $\left|\overrightarrow{V_{0-i}^L}\right|$. The Byrd glacier (Fig. 2b) shows a similar case with a small total positive value of flux OE (< 1 Gt a⁻¹). In contrast, in the Totten glacier (Fig. 2c) there are some flow lines that have large $\theta_{0-i}$ angles and make direct contributions to ice flux cross GL of the main trunk and a tributary glacier. However, flow lines cut most of the GL (e.g., along shear margins) at smaller $\theta_{0-i}$ angles (e.g., enlarged areas "A' and "B"), which continuously decrease over the 15 years (see Table 1), making a decreased trend in flux (a total flux underestimation of ~3.7 Gt a⁻¹). The David glacier (Fig. 2d) has a similar situation with a total flux underestimation of ~1.8 Gt a⁻¹ for the 15-year span.

Table 1. Values of $\theta_{0-i}$ angle in areas A and B over 15 years.

| Time (Year) | $\theta_{0-i}$ in Box A (°) | $\theta_{0-i}$ in Box B (°) |
|---|---|---|
| 1 | 45.8 | 25.8 |
| 2 | 45.2 | 25.4 |
| 3 | 44.3 | 24.1 |
| 4 | 43.1 | 23.1 |
| 5 | 41.8 | 21.9 |
| 6 | 40.4 | 20.5 |
| 7 | 38.9 | 19.0 |
| 8 | 37.3 | 17.8 |
| 9 | 35.6 | 16.6 |
| 10 | 33.9 | 15.9 |
| 11 | 32.0 | 14.9 |
| 12 | 29.9 | 14.0 |
| 13 | 27.8 | 13.8 |
| 14 | 25.7 | 13.1 |

| 15 | 23.6 | 12.5 |
|---|---|---|

Extension of this exercise to the entire continent involves a complex situation, including deceleration caused by ice rises, mapping errors near shear margins, GL data points that would calve into the ocean within 15 years, and other potential factors. Given the effects of the $\theta_{0-i}$ angle on the ice flux that is amplified by the long span of 15 years (as indicated in the code provided, Equation (1) and results in Fig. 2), the "Antarctica" GL flux curve in Fig. 1 mainly reflect the fact that at the continent scale, the GL flux OEs accumulated in glaciers like PIG are compensated by underestimations in glaciers such as Totten and David. These underestimations may be small in localized regions, but they accumulate along the large percentage of the entire Antarctic GL and takes a significant effect at the continent scale (Fig. 1).

(Line 470) We believe that a systematic and in-depth investigation is needed to reach a conclusive result. We added the following text in *Discussion*: "*...... Extension of this exercise to the entire continent involves a complex situation, including deceleration caused by ice rises, mapping errors near ice shelf shear margins, GL data points that would calve into the ocean within n-year span, and other potential factors. It should be noted that given a velocity range in flux computation, the ice flow angle between a flux gate and ice flow controls the flux magnitude. In principle, at the same gate this angle changes with velocities of different time spans and flow line patters, resulting in an overestimation or underestimation in flux for individual glaciers. A systematic and in-depth investigation should be carried out to handle the overestimation issue in GL flux of the entire Antarctica.*"

**Summary: I don't believe there's any particular harm in publishing the manuscript in its current form, but I do believe an opportunity has been missed for the authors to share the insights and intuition they have surely developed over the course of this work.**

Response:

Thanks for the suggestions. We added those we felt comfortable into this paper and left few for our future work.

**Specific comments**

**Line 394 alludes to experiments that were performed on Jakobshavn Isbrae and introduces the acronym JI for the glacier, but the experiments at Jakobshavn are not described anywhere in the paper, and the acronym is never used. Am I missing something?**

Response:

(Line 412) It was suggested by Reviewer 1 to compare Jakobshavn Isbrae (JI) as well. Now we removed everything related JI in the revised manuscript.

**Line 399 starts a new section with the word "Furthermore,..." as if a thought is being continued from a previous section. Section breaks are often where readers pause to grab a cup of coffee, so as a general rule I recommend starting each section in such a way that readers can pick up where they left off, without having to re-read the previous section.**

Response:

(Line 417) We deleted "Furthermore" and start the section directly with "*Within a long time span ......*"

**Line 339 says GL flux was computed. How? What grounding line and ice thickness products were used?**

Response:

(Line 456) We added the details: "*In addition, based on an annual E-velocity map of 2013 in PIG from ITS_LIVE (Fig. A4a), L-velocities along grounding line (GL; Gardner et al., 2018) with time spans of 1 to 15 years (Fig. A4b) and the associated ice discharge was computed. The actual flux gates were set with nodes separated every 240 m, which were located up to 13 km upstream the GL to reduce the uncertainty of ice thickness data (Gardner et al., 2019) from BedMachine (Morlighem et al., 2020)......*"

**Line 440 says "OEs in velocity can further cause an overestimation in GL flux, which is negligible within a 3-year span" but no evidence or rationale for this statement is presented in the manuscript.**

Response:

(Line 461) We deleted that part of the sentence: "…… "

**Line 443 says "Therefore, the influence of the velocity OEs on the GL flux appears to be not very significant." Does this mean the main thesis of the paper is more theoretical than practical?**

Response:

(Line 465) We deleted the sentence ""

**Lines 445-449 offer concluding remarks on the Jakobshavn experiments that are not described in this paper. This paragraph can be removed.**

Response:

(Line 477) Now we removed this paragraph and everything related JI in the paper.

---

## Author Response (AR3)

We appreciate the constructive comments and suggestions from the editor.

**Editor:**

**Dear Authors,**

**Thanks a lot for revising the manuscript and taking into account the final comments of the referee.**

**However, I think that your abstract needs some work to really capture the attention of the readers and not loose them in too many numbers. The structure of the abstract is also not very clear. See suggestions below.**

**Next round, I will just check the abstract.**

**Best regards,**

**Etienne Berthier**

**The abstract is now quite long. If you could streamlined and shortened it, it would have more impact on the reader. I provide some suggestions below but I think it is better to really rewrite it.**

Response:

Thanks. The abstract is revised accordingly.

*Abstract. Antarctic ice velocity maps describe the ice flow dynamics of the ice sheet and are one of the primary components used to estimate the Antarctic mass balance and contribution to global sea level changes. In comparison to velocity maps derived from recent satellite images of monthly to weekly time spans, historical maps, from before the 1990s, generally cover longer time spans, e.g., over 10 years, due to the scarce spatial and temporal coverage of earlier satellite image data. We found velocity overestimations (OEs) in such long-span maps that can be mainly attributed to velocity gradients and time span of the images used. In general, they are less significant in slow-flowing grounded regions with low spatial accelerations. Instead, they take effects in places of high ice dynamics, for example, near grounding lines and often in ice shelf fronts. Velocities in these areas are important for estimating ice sheet mass balance and analyzing ice shelf instability. We propose an innovative Lagrangian velocity-based method for OE correction without the use of field observations or additional image data. The method is validated by using a set of "ground truth" velocity maps for the Totten Glacier and Pine Island Glacier which are produced from high-quality Landsat 8 images from 2013 to 2020. Subsequently, the validated method is applied to a historical velocity map of the David Glacier region from images from 1972–1989 acquired during Landsat 1, 4 and 5 satellite missions. It is demonstrated that velocity*

*overestimations of up to 39 m a⁻¹ for David Glacier and 195 m a⁻¹ for Pine Island Glacier can be effectively corrected. Furthermore, temporal acceleration information, e.g., on basal melting and calving activities, is preserved in the corrected velocity maps and can be used for long-term ice flow dynamics analysis. Our experiment results in PIG show that OEs of a 15-year span can reach up to 1,300 m a⁻¹ along the grounding line and cause an overestimated GL flux of 11.5 Gt a⁻¹ if not corrected. The magnitudes of the OEs contained in both velocity and mass balance estimates are significant. When used alongside recent velocity maps of 1990s – 2010s, they may lead to underestimated long-term changes for assessment and forecast modeling of the global climate change impact on the Antarctic ice sheet. Therefore, the OEs in the long span historical maps must be seriously examined and corrected. We recommend that overestimations of more than the velocity mapping uncertainty (1σ) be corrected. This velocity overestimation correction method can be applied to the production of regional and ice sheet-wide historical velocity maps from long-term satellite images.*

**L14: here the reader could think of acceleration through time. Maybe "velocity gradients" or "ice flow acceleration (in space)" are better**

Response:

We revised it, using "velocity gradients".

**L15-16. I am note sure you need to include these examples here because they are other numbers now further down in the abstract to illustrate the magnitude of the OEs.**

Response:

The examples with numbers are removed.

**L28-31. Could you try to simplify these sentences. "Heavy" writing style.**

Response:

They are simplified: "*The magnitudes of the OEs contained in both velocity and mass balance estimates are significant. When used alongside recent velocity maps of 1990s – 2010s, they may lead to underestimated long-term changes for assessment and forecast modeling of the global climate change impact on the Antarctic ice sheet. Therefore, ……*"